# AN IMPROVED ANALYSIS OF PER-SAMPLE AND PER-UPDATE CLIPPING IN FEDERATED LEARNING

**Bo Li**[*†]
DTU[‡]
blia@dtu.dk

**Xiaowen Jiang**[*]
CISPA[§]
xiaowen.jiang@cispa.de

**Mikkel N. Schmidt**
DTU[‡]
mnsc@dtu.dk

**Tommy S. Alstrøm**
DTU[‡]
tsal@dtu.dk

**Sebastian U. Stich**
CISPA[§]
stich@cispa.de

## ABSTRACT

Gradient clipping is key mechanism that is essential to differentially private training techniques in Federated learning. Two popular strategies are per-sample clipping, which clips the mini-batch gradient, and per-update clipping, which clips each user's model update. However, there has not been a thorough theoretical analysis of these two clipping methods. In this work, we rigorously analyze the impact of these two clipping techniques on the convergence of a popular federated learning algorithm FedAvg under standard stochastic noise and gradient dissimilarity assumptions. We provide a convergence guarantee given any arbitrary clipping threshold. Specifically, we show that per-sample clipping is guaranteed to converge to the neighborhood of the stationary point, with the size dependent on the stochastic noise, gradient dissimilarity, and clipping threshold. In contrast, the convergence to the stationary point can be guaranteed with a sufficiently small stepsize in per-update clipping at the cost of more communication rounds. We further provide insights into understanding the impact of the improved convergence analysis in the differentially private setting.

## 1 INTRODUCTION

Federated learning (FL) is an essential distributed learning scheme where workers collaboratively train a model without sharing their data (Acar et al., 2021; Kairouz et al., 2019; Karimireddy et al., 2019; McMahan et al., 2016). One of the popular federated optimization methods is FedAvg (McMahan et al., 2016), where locally trained models are averaged on a central server in a series of rounds. However, existing works have shown that vanilla FedAvg is vulnerable to inference attacks as sensitive information can be extracted from the learned model parameters (Carlini et al., 2018; Fredrikson et al., 2015; Nasr et al., 2018; Melis et al., 2018). Therefore, ensuring the model parameters do not reveal input data distribution has attracted considerable interest (Andrew et al., 2021; McMahan et al., 2018; Choudhury et al., 2019; Wei et al., 2019; Geyer et al., 2017).

Since each training example can leave a footprint on the gradients (Nasr et al., 2018), one way to mitigate the risk of information leakage is to bound the gradients (Andrew et al., 2021; McMahan et al., 2018) via gradient clipping (Zhang et al., 2022; Koloskova et al., 2023). Two popular clipping strategies in FL are per-sample clipping (Liu et al., 2022) and per-update clipping (McMahan et al., 2018; Andrew et al., 2021; Geyer et al., 2017; Wei et al., 2019). In each local iteration, per-sample clipping limits the norm of the mini-batch gradient. In contrast, per-update clipping limits the overall local update in each round, which is the product of the stepsize and the sum of the mini-batch gradients. Per-sample (when the batch size is 1) and per-update clipping are commonly used to

---

[*]Equal contribution
[†]Work done while at CISPA
[‡]Technical University of Denmark
[§]CISPA Helmholtz Center for Information Security

achieve example-level (Abadi et al., 2016; Choudhury et al., 2019) and user-level (McMahan et al., 2018; Geyer et al., 2017) differential privacy.

While tremendous efforts have been made to understand the influence of clipping on the convergence of optimization algorithms (Zhang et al., 2020b; Koloskova et al., 2023; Zhang et al., 2020a; Mai & Johansson, 2021; Zhang et al., 2022; Liu et al., 2022), existing methods often imply a specific choice of $c$ to guarantee convergence (Zhang et al., 2020b). To address this issue, Koloskova et al. (2023) studied the convergence behavior for clipped mini-batch SGD under *any* arbitrary clipping threshold on stochastic non-convex functions and established a tight convergence guarantee. However, the impact on federated learning is not yet known. Besides, prior theoretical results usually rely on uniformly bounded stochastic noise (Zhang et al., 2020b;a; Crawshaw et al., 2023; Liu et al., 2022) and bounded gradients (Zhang et al., 2022) assumptions, which do not necessarily hold in practice, especially in the training of deep neural networks (Gorbunov et al., 2020; Simsekli et al., 2019; Krizhevsky et al., 2012). Moreover, a formal theoretical analysis comparing the convergence behavior of these two commonly used clipping operations in federated learning remains unexplored.

In this work, we precisely characterize the impact of the two commonly used clipping techniques: per-sample and per-update clipping, on the convergence of a popular federated optimization algorithm FedAvg (McMahan et al., 2016), under the standard bounded variance $\sigma^2$ (Koloskova et al., 2023) and gradient dissimilarity $\zeta^2$ (Woodworth et al., 2020; Reddi et al., 2020) assumptions to include heavy-tailed noise (Simsekli et al., 2019) and data heterogeneity scenarios (Kairouz et al., 2019).

We prove that per-sample clipping is guaranteed to converge to a neighborhood with the size dependent on the stochastic noise $\sigma$, gradient dissimilarity $\zeta$ and the clipping threshold $c$. Specifically, the assurance of attaining any desired level of accuracy can only be achieved by selecting a sufficiently large value for the clipping threshold. This result is consistent and is an extension of the work from Koloskova et al. (2023), which studied the convergence behavior of clipped mini-batch SGD in a single worker. Conversely, given any arbitrary clipping threshold, per-update clipping can converge to any accuracy level by choosing the appropriate stepsizes. We show that the main reason for the different behaviors is the incorporation of an inner stepsize in per-update clipping. Consequently, one can develop a better per-sampling clipping algorithm based on this insight.

**Contributions:** we summarize our main contributions as below:

- We rigorously analyze the impact of two popular clipping strategies: per-sample and per-update clipping, on the convergence of FedAvg, under standard bounded variance and gradient dissimilarity assumptions.
- We precisely characterize the impact of stochastic noise $\sigma^2$ and data heterogeneity $\zeta^2$ on the clipped FedAvg. Specifically, we show that per-sample clipping is guaranteed to converge to a neighborhood of the stationary point, with the size being $\mathcal{O}\big(\min(\sigma + \zeta, \frac{\sigma^2 + \zeta^2}{c})\big)$. We provide examples to show the tightness of the neighborhood size. On the other hand, given arbitrary clipping threshold $c$, per-update clipping can converge to any accuracy by picking the inner stepsize to be smaller than $\mathcal{O}\left(\frac{c}{\sqrt{\tau}\sigma + \tau\zeta}\right)$ where $\tau$ is the number of local steps.
- We extend our theoretical results into the differentially private setting where a stochastic noise is injected into each step. We provide insights into understanding the impact of the $\sigma$ and $\zeta$ for determining the appropriate stepsize and clipping thresholds to achieve better privacy-utility trade-off.
- We experimentally validate our theoretical statements under different levels of data heterogeneity.

## 1.1 RELATED WORK

**Clipping in centralized learning:** Clipping is a popular technique that can enhance the stability of the optimization process (Dosovitskiy et al., 2020; Mai & Johansson, 2021; You et al., 2017; Pascanu et al., 2012b;a; Steiner et al., 2021). Recently, numerous efforts have been made to understand clipping theoretically. Inspired by the practical observations from DNN training, Zhang et al. (2020b) proposed a new $(L_0, L_1)$-smoothness assumption and justified that gradient clipping can accelerate the convergence of SGD. Zhang et al. (2020a) and Mai & Johansson (2021) then extended the analysis to incorporate momentum methods for non-smooth functions. However, the above theoretical results rely on a strong assumption, namely every stochastic gradient falls into a ball

Table 1: Convergence rate of for non-convex function $f : \mathbb{R}^d \to \mathbb{R}$, with $L$ described the smoothness, given $c$ as the clipping threshold, $\eta$ as the stepsize in per-sample clipping, $\eta_l$ and $\eta_g$ as the inner and outer stepsize in per-update clipping, $F_0 := f(\mathbf{x}_0) - f(\mathbf{x}^\star)$, $\tau$ as the number of local steps, $n$ as the number of workers, $R$ as the number of communication rounds, and $T := R \cdot \tau$ as the total number of iterations.

| Algorithm | Convergence | Assumptions |
|---|---|---|
| **Per-sample:** $\min_{t \in [1,T]} \mathbb{E}[\|\nabla f(\bar{\mathbf{x}}_t)\|]$ | | |
| SGD Koloskova et al. (2023) (single worker) | $\mathcal{O}\left(\sqrt{\frac{F_0}{\eta T}} + \sqrt{\eta L}\frac{\sigma}{\sqrt{B}}\right)$ | - |
| Clipped-SGD Koloskova et al. (2023) (single worker) | $\mathcal{O}\left(\sqrt{\frac{F_0}{\eta T}} + \frac{F_0}{\eta T c} + \sqrt{\eta(L_0+cL_1)}\frac{\sigma}{\sqrt{B}} + \min(\sigma, \frac{\sigma^2}{c})\right)$ | Assumption 1 |
| LocalSGD Koloskova et al. (2020) ($n$ workers) | $\mathcal{O}\left(\sqrt{\frac{F_0}{\eta T}} + \sqrt{\frac{\eta L}{n}}\sigma + \eta L\sqrt{\tau^2\zeta^2 + \tau\sigma^2}\right)$ | Assumption 1,2 |
| Clipped LocalSGD (homogeneous Liu et al. (2022)) | $\mathcal{O}\left(\sqrt{\frac{F_0}{\eta T}} + \sqrt{\frac{\eta L_0}{n}}\sigma + \eta L_0 \tau\sigma\right)$ | Assumptions 3, 4, 5 |
| FAT clipping-PI ($\beta$-moment Yang et al. (2022)) | $\mathcal{O}\left(\sqrt{\frac{F_0}{\eta_l\eta_g T}} + G^\beta c^{1-\beta} + L\eta_l\tau G^{\frac{\beta}{2}}c^{1-\frac{\beta}{2}} + \sqrt{\frac{L\eta_l\eta_g}{n}}G^{\frac{\beta}{2}}c^{1-\frac{\beta}{2}}\right), \quad \beta \in (1,2]$ | Assumption 7 |
| Ours (per-sample) | $\mathcal{O}\left(\sqrt{\frac{F_0}{\eta T}} + \frac{F_0}{\eta cT} + \sqrt{\frac{\eta(L_0+cL_1)(\sigma^2+\zeta^2)}{n}} + (L_0+cL_1)\eta\min\left(\tau\sqrt{\sigma^2+\zeta^2}, c\right) + \min\left(\sigma+\zeta, \frac{\sigma^2+\zeta^2}{c}\right)\right)$ | Assumption 1-3 |
| **Per-update:** $\min_{r \in [1,R]} \mathbb{E}[\|\nabla f(\mathbf{x}_r)\|]$ | | |
| FedAvg Karimireddy et al. (2019) ($n$ workers) | $\mathcal{O}\left(\sqrt{\frac{F_0}{\eta_l\eta_g\tau R}} + \sqrt{\frac{\eta_l\eta_g L\sigma^2}{n}} + \eta_l\tau L\zeta\right)$ | Assumption 1,2 |
| Clipped-FedAvg Zhang et al. (2022) (per-update) | $\mathcal{O}\left(\sqrt{\frac{F_0}{\eta_l\eta_g\tau R}} + \eta_l\sqrt{L\tau(\sigma^2+\tau\zeta^2)} + \sqrt{\frac{\eta_l\eta_g L\sigma^2}{n}} + \sqrt{G^2\frac{1}{R}\sum_{r=1}^R \mathbb{E}[\frac{1}{n}\sum_{i=1}^n(|\alpha_i^r-\tilde{\alpha}_i^r|+|\tilde{\alpha}_i^r-\bar{\alpha}_i^r|)]} + \sqrt{\eta_g\eta_l L\tau G^2 \frac{1}{R}\sum_{r=1}^R \mathbb{E}[\frac{1}{n}\sum_{i=1}^n(|\alpha_i^r-\tilde{\alpha}_i^r|+|\tilde{\alpha}_i^r-\bar{\alpha}_i^r|)]}\right)$ | Assumption 1, 4, 6 |
| FAT clipping-PR ($\beta$-moment Yang et al. (2022)) | $\mathcal{O}\left(\sqrt{\frac{F_0}{\eta_l\eta_g\tau R}} + L\eta_l\tau G + \tau G^\beta c^{-(\beta-1)} + \sqrt{L\eta_l\tau^2 G^{1+\beta}c^{1-\beta}} + \sqrt{\frac{L\eta_g\eta_l}{n}}(\tau G^\beta c^{2-\beta})\right), \quad \beta \in (1,2]$ | Assumption 7 |
| Ours (per-update) | $\mathcal{O}\left(\mathbb{1}_{c \geq \mathcal{O}(\eta_l\sqrt{\tau}\sigma+\eta_l\tau\zeta)}\left\{\sqrt{\frac{F_0}{\eta_l\eta_g\tau R}} + \sqrt{\frac{\eta_l\eta_g\tau L}{n}}(\frac{\sigma}{\sqrt{\tau}}+\zeta) + \eta_l\tau L\zeta + \frac{\eta_l\tau}{c}(\frac{\sigma^2}{\tau}+\zeta^2)\right\} + \frac{F_0}{c\eta_g R} + \mathbb{1}_{c \leq \mathcal{O}(\eta_l\sqrt{\tau}\sigma+\eta_l\tau\zeta)}\left\{\frac{\sigma}{\sqrt{\tau}}+\zeta\right\}\right)$ | Assumption 1-3 |

[4] Bounded gradient assumes $\|\nabla F_i(\mathbf{x})\| \leq G$ for all $i \in [n]$ and $\mathbf{x} \in \mathbb{R}^d$.
[5] Uniformly bounded noise assumes $\|\nabla F_i(\mathbf{x}) - \nabla f_i(\mathbf{x})\| \leq \sigma^2$ for all $i \in [n]$ and $\mathbf{x} \in \mathbb{R}^d$.
[6] Bounded dissimilarity assumes $\|\nabla f_i(\mathbf{x}) - \nabla f(\mathbf{x})\| \leq \zeta^2$ for all $i \in [n]$ and $\mathbf{x} \in \mathbb{R}^d$.
[7] $\beta$-moment assumes $\mathbb{E}[\|\nabla F_i(\mathbf{x})\|^\beta] \leq G^\beta$ for all $i \in [n]$ and $\mathbf{x} \in \mathbb{R}^d$, and it implies $\|\nabla f(\mathbf{x})\| \leq G$.
[8] $\alpha_i^r := \frac{c}{\max(c,\eta_l\|\sum_{k=1}^\tau \nabla F_i(\mathbf{y}_{i,k}^r)\|)}, \tilde{\alpha}_i^r := \frac{c}{\max(c,\eta_l\|\mathbb{E}[\sum_{k=1}^\tau \nabla F_i(\mathbf{y}_{i,k}^r)]\|)}$ and $\bar{\alpha}^r := \frac{1}{n}\sum_{i=1}^n \tilde{\alpha}_i^r$.

with a radius of $\sigma$ to the true (expected) gradients, which can hardly hold in many practical DNN-based applications (Simsekli et al., 2019; Zhang et al., 2020c; Krizhevsky et al., 2012; Vaswani et al., 2017). Therefore, Gorbunov et al. (2020) relaxed this by introducing bounded variance in expectation to include heavy-tailed noise. Nevertheless, the convergence guarantee above usually implies and requires a specific value of the clipping threshold $c$, meaning $c$ must be tuned carefully to achieve the suggested convergence. Tuning clipping threshold, however, may raise other privacy concerns (Andrew et al., 2021). Therefore, Koloskova et al. (2023) provided a convergence guarantee of using clipped SGD given *any* arbitrary clipping threshold to address this issue. In a stochastic setting, they demonstrated that the *clipping bias* can hamper the convergence to the true optimum.

**Clipping for privacy protection in FL:** Recent works have shown that vanilla FL algorithms are vulnerable to adversary attacks as it is possible to extract information from the participating users by looking at the parameters of a trained model (Fredrikson et al., 2015; Carlini et al., 2018; Melis et al., 2018). Therefore, providing a certain level of privacy guarantee is crucial Wei et al. (2019); Abadi et al. (2016); McMahan et al. (2018). To make such promises, each data point (user) 's maximum contribution needs to be bounded (Bassily et al., 2019; Wang et al., 2018; Das et al., 2022; Amin et al., 2019), which is usually achieved by projecting larger updates back to a ball of norm $c$ using clipping. Two popular clipping operations are per-sample clipping and per-update clipping. Per-sample clipping clips the gradient of each data point to limit its influence on the model parameters, (Liu et al., 2022), which can protect example-level privacy (Abadi et al., 2016). Per-update clipping restricts the local model update (Geyer et al., 2017), which is the product between the sum of mini-batch gradient and the stepsize (McMahan et al., 2016; Karimireddy et al., 2019; Acar et al., 2021). Per-update clipping

is an essential tool for preserving user-level privacy (Geyer et al., 2017; Zhang et al., 2022; McMahan et al., 2018), which provides stronger privacy guarantee than example-level privacy.

To theoretically understand the effect of clipping in the federated optimization, Zhang et al. (2022) and Liu et al. (2022) presented the convergence guarantee of per-update and per-sample clipping, requiring the bounded gradient assumption. Despite the success of these algorithms, the convergence behavior of clipping in federated learning, especially the differences between different clipping methods, has yet to be well explored. We aim to fill this gap in this paper.

## 2 PROBLEM FORMULATION

We formalize the problem as minimizing a sum of stochastic functions with only access to stochastic samples:

$$f^\star := \min_{\mathbf{x} \in \mathbb{R}^d} \left[ f(\mathbf{x}) := \frac{1}{n} \sum_{i=1}^n f_i(\mathbf{x}) \right], \quad f_i(\mathbf{x}) := \mathbb{E}_{\xi \sim \mathcal{D}_i} F_i(\mathbf{x}, \xi),$$

where $f_i : \mathbb{R}^d \to \mathbb{R}$ are distributed among $n$ workers and $\mathcal{D}_i$ is the distribution of data $\xi$ on worker $i$. For our theoretical results, we make the following assumptions.

We first assume bounded variance in expectation following Koloskova et al. (2023) to incorporate heavy-tailed stochastic noise (Gorbunov et al., 2020) in Assumption 1. As data heterogeneity in unavoidable in FL, we assume gradient dissimilarity following Woodworth et al. (2020); Reddi et al. (2020); Yuan et al. (2016) in Assumption 2. If all the objective functions are identical (homogeneous workers), $f_i = f_j, \forall i, j$, then the gradient dissimilarity is 0.

**Assumption 1** (bounded variance)**.** *We assume that there exists a $\sigma^2$ such that $\forall \mathbf{x} \in \mathbb{R}^d$, we have:*

$$\mathbb{E}_\xi ||\nabla F_i(\mathbf{x}, \xi) - \nabla f_i(\mathbf{x})||^2 \le \sigma^2.$$

**Assumption 2** (gradient dissimilarity)**.** *We assume there exist a $\zeta^2$, such that, $\forall \mathbf{x} \in \mathbb{R}^d$, we have:*

$$\mathbb{E}_i ||\nabla f_i(\mathbf{x}) - \nabla f(\mathbf{x})||^2 \le \zeta^2.$$

We propose an adaptation of the $(L_0, L_1)$-smoothness assumption from Zhang et al. (2020b); Koloskova et al. (2023); Zhang et al. (2020a) that is tailored to distributed setup as our smoothness assumption can account for data heterogeneity and facilitate proof:

**Assumption 3** (distributed $(L_0, L_1)$-smoothness)**.** *We assume there exists $L_0$ and $L_1$ such that $\{f_i\}$ is $(L_0, L_1)$-smooth $\forall \mathbf{x}, \mathbf{y} \in \mathbb{R}^d$ with $||\mathbf{x} - \mathbf{y}|| \le \frac{1}{L_1}$:*

$$||\nabla f_i(\mathbf{x}) - \nabla f_i(\mathbf{y})|| \le (L_0 + L_1 ||\nabla f(\mathbf{x})||)||\mathbf{x} - \mathbf{y}||. \tag{1}$$

This assumption introduces a coupling of heterogeneity and smoothness of the function. Under standard individual $(L_0, L_1)$-smoothness (Zhang et al., 2020b;a) and gradient dissimilarity, this assumption is always satisfied. Similar to the non-distributed setting, Assumption 3 recovers the standard L-smoothness from Nesterov (2018) when $L_1 = 0$. When $L_1 > 0$, Assumption 3 is weaker than the standard L-smoothness assumption as it can include a group of simple and important functions that do not necessarily satisfy global L-smoothness under any $L$, *e.g.* polynomial function $f(x) = x^4$. See Appendix A.1.1 for a discussion on the implication of Assumption 3.

## 3 ALGORITHMS AND CONVERGENCE RESULTS

In this section, we first describe the FedAvg algorithm (McMahan et al., 2016). FedAvg mainly has two steps: local model updating on each worker and model aggregation on the server. FedAvg initializes the server with a server model $\mathbf{x}$. Then each participating worker receives a copy of the server model $\mathbf{x}$ and performs $\tau$ steps of (stochastic) gradient descent (SGD). The updated local models are then communicated to the server for aggregation, finishing one communication round. This process is repeated for $R$ rounds or until we have reached the target accuracy. However, it has been observed that adversarial servers can extract sensitive information from the learned model parameters (Carlini et al., 2018; Zhang et al., 2022; Fredrikson et al., 2015; Melis et al., 2018).

**Algorithm 1** Per-sample clipping

1: **procedure** PER-SAMPLE CLIPPING
2:     Initialize stepsize $\eta$
3:     **for** $r = 1, \ldots, R$ **do**
4:         Send server model $\mathbf{x}$ to all clients
5:         **for** client $i = 1, \ldots, n$ **in parallel do**
6:             initialize local model $\mathbf{y}_i \leftarrow \mathbf{x}$
7:             **for** $k = 1, \ldots, \tau$ **do**
8:                 $\mathbf{g}_i \leftarrow \min \left(1, \frac{c}{||\nabla F_i(\mathbf{y}_i)||}\right) \nabla F_i(\mathbf{y}_i)$
9:                 $\mathbf{y}_i \leftarrow \mathbf{y}_i - \eta \mathbf{g}_i$
10:             **end for**
11:             Communicate $\mathbf{y}_i$ to the server
12:         **end for**
13:         $\mathbf{x} \leftarrow \mathbf{x} + \frac{1}{n} \sum_i (\mathbf{y}_i - \mathbf{x})$
14:     **end for**
15: **end procedure**

**Algorithm 2** Per-update clipping

1: **procedure** PER-UPDATE CLIPPING
2:     Initialize local and global stepsize $\eta_l, \eta_g$
3:     **for** $r = 1, \ldots, R$ **do**
4:         Send server model $\mathbf{x}$ to all clients
5:         **for** client $i = 1, \ldots, n$ **in parallel do**
6:             initialize local model $\mathbf{y}_i \leftarrow \mathbf{x}$
7:             **for** $k = 1, \ldots, \tau$ **do**
8:                 $\mathbf{y}_i \leftarrow \mathbf{y}_i - \eta_l \nabla F_i(\mathbf{y}_i)$
9:             **end for**
10:             $\Delta_i \leftarrow \mathbf{y}_i - \mathbf{x}$
11:             $\Delta_i \leftarrow \min \left(1, \frac{c}{||\Delta_i||}\right) \Delta_i$
12:             Communicate $\Delta_i$ to the server
13:         **end for**
14:         $\mathbf{x} \leftarrow \mathbf{x} + \eta_g \frac{1}{n} \sum_i \Delta_i$
15:     **end for**
16: **end procedure**

Therefore, we next describe two popular clipping methods, which are critical to regularize each user's contribution and thus facilitate the analysis of differential privacy.

### 3.1 PER-SAMPLE CLIPPING

Following Liu et al. (2022), we clip the mini-batch gradient at every local iteration (see Algorithm 1). Throughout this section, we denote $M := \max_t\{||\nabla f(\bar{\mathbf{x}}_t)||\}$ and adopt the virtual sequence definition $\bar{\mathbf{x}}_t := \frac{1}{n} \sum_{i=1}^{n} \mathbf{y}_{i,t}$ with $\bar{\mathbf{x}}_t = \mathbf{x}_t$ when $t \mod (\tau + 1) = 0$ from Stich (2019). See Appendix A.1.1 for the discussion of the $M$ parameter and Appendix D for the proof.

**Theorem I** (per-sample clipping). *Suppose functions $\{f_i\}$ satisfy Assumption 1 to 3. If we run Algorithm 1 for $T := R \cdot \tau$ steps with $R$ communication rounds, $\tau$ local steps, clipping threshold $c$, and stepsize $\eta \leq \frac{1}{14L\tau}$ with $L := L_0 + \min(c, M)L_1$ and $M := \max_t\{||\nabla f(\bar{\mathbf{x}}_t)||\}$, then it holds that:*

$$\min_{t \in [1,T]} \mathbb{E}||\nabla f(\bar{\mathbf{x}}_t)|| \leq \mathcal{O}\left( \frac{F_0}{\eta c T} + \sqrt{\frac{F_0}{\eta T}} + \sqrt{\frac{\eta L(\sigma^2 + \zeta^2)}{n}} \right.$$
$$\left. + \eta L \min\left(\tau \sqrt{\sigma^2 + \zeta^2}, c\right) + \min\left(\sigma + \zeta, \frac{\sigma^2 + \zeta^2}{c}\right) \right), \tag{2}$$

*where $F_0 := f(\mathbf{x}_0) - f^\star$*

The convergence criterion on the left-hand side, the minimum gradient norm, could also be replaced with the average of the expected gradient norm of the virtual iterates. When there is no stochastic noise and heterogeneity ($\sigma^2 = 0, \zeta^2 = 0$), the first two terms control the convergence and can decrease with the increasing number of iterations $T$. The third term indicates a linear speedup in the number of workers (take square for both sides). The last bias term can decrease with a larger clipping threshold assuming the $\sigma^2$ and $\zeta^2$ are fixed and are larger than zero. We provide concrete examples similarly to Koloskova et al. (2023) considering $\tau = 1$ to illustrate that the neighborhood size is tight, i.e., $\mathbb{E}||\nabla f(\bar{\mathbf{x}}_t)|| = \Omega\left(\min\left(\sigma + \zeta, \frac{\sigma^2 + \zeta^2}{c}\right)\right)$, $\forall t \geq 1$ together with the proof in Appendix B.

**Comparison to the unclipped FedAvg:** Under Assumptions 1–2 and standard L-smoothness assumption, the unclipped FedAvg has the following convergence rate $\min_{t \in [1,T]} \mathbb{E}||\nabla f(\bar{\mathbf{x}}_t)|| \leq \mathcal{O}\left(\sqrt{\frac{F_0}{\eta T}} + \sqrt{\frac{\eta L \sigma^2}{n}} + L\eta \sqrt{\tau \sigma^2 + \tau^2 \zeta^2}\right)$. Comparably, the most notable difference is the bias term $\mathcal{O}\left(\min\left(\sigma + \zeta, (\sigma^2 + \zeta^2)/c\right)\right)$, which can decrease with increasing $c$. When $c \to \infty$, the radius is 0 as no gradients get clipped. (See Theorem IV in Appendix D for recovering FedAvg when $c$ is large).

**Corollary I.** *Suppose functions $\{f_i\}$ satisfy Assumption 1 to 3, if we add random Gaussian noise $\mathbf{z}_{i,t}$ to $\mathbf{g}_{i,t}$ in Algorithm 1, such that $\mathbf{z}_{i,t} \sim \mathcal{N}\left(0, \sigma_{DP}^2/d\mathbf{I}_d\right)$, and run it for $T := R \cdot \tau$ steps with stepsize*

$\eta \le \frac{1}{14L\tau}$ *and* $L := L_0 + ML_1$, *and define* $F_0 := f(\mathbf{x}_0) - f^\star$, *then we have:*

$$\min_{t \in [1,T]} \mathbb{E}||\nabla f(\bar{\mathbf{x}}_t)|| \le \mathcal{O}\left( \frac{F_0}{\eta cT} + \sqrt{\frac{F_0}{\eta T}} + \sqrt{\frac{\eta L(\sigma^2 + \zeta^2)}{n}} + L\eta \min\left(\tau\sqrt{\sigma^2 + \zeta^2}, c\right) \right.$$

$$\left. + \min\left(\sigma + \zeta, \frac{\sigma^2 + \zeta^2}{c}\right) + \frac{\eta L \sigma_{DP}^2}{nc} + \sqrt{\frac{\eta L}{n}} \sigma_{DP} \right). \tag{3}$$

**Extension to privacy-preserving FedAvg:** We extend our algorithm to make it suitable for preserving privacy for each individual data point in some scenarios. Specifically, we update the local model following $\mathbf{y}_i \leftarrow \mathbf{y}_i - \eta(\mathbf{g}_i + \mathbf{z}_i)$ where $\mathbf{z}_i \sim \mathcal{N}(0, \sigma_{\mathrm{DP}}^2/d\mathbf{I}_d)$ and $d$ is the dimensionality of the parameters. When batch size is 1, this updating rule protects example-level privacy for each client (Abadi et al. (2016)) and achieves local differential privacy. The updated convergence result is shown in Corollary I. Compared to the convergence result from Theorem I, we have extra terms that are dependent on the variance of the added Gaussian noise and are decreasing with a larger number of clients $n$. See the privacy discussion in section 3.3 for the connection to centralized DP-SGD.

**Privacy discussion** We consider the scenario where the server might be malicious. Following Abadi et al. (2016), the variance of the added noise $\sigma_{\mathrm{DP}}^2$ needs to be $\Omega\left(\frac{c^2 d \log(1/\delta)T}{N^2 \varepsilon_{\mathrm{DP}}^2}\right)$ to achieve example-level $(\varepsilon_{\mathrm{DP}}, \delta)$-local differential privacy, where $N := \min_i\{|\mathcal{D}_i|\}$ is the minimum number of data points among all the clients and mini-batch size is one on each client. We require the definition of $N$ as the *record* that differs between the neighboring dataset is one example in the training data for each client (Abadi et al., 2016). When $\Theta(\sigma^2 + \zeta^2)$ is small, the optimal utility after $T = \Theta\left(\frac{\tau n N^2 \varepsilon_{\mathrm{DP}}^2}{d \log(1/\delta)}\right)$ iteration is $\mathcal{O}\left(\frac{d \log(1/\delta)}{n N^2 \varepsilon_{\mathrm{DP}}^2}\right)$ by choosing $c = \Theta\left(\sqrt{\frac{d \log(1/\delta)}{n N^2 \varepsilon_{\mathrm{DP}}^2}}\right)$ and $\eta = \Theta(1/(\tau L))$. We obtain optimal iteration complexity $T$ when $\tau = 1$, which shows no benefits in doing multiple local steps from the analysis. When $\Theta(\sigma^2 + \zeta^2)$ dominates the convergence, we need to pick larger $c$ and smaller $\eta$ to reach the optimal utility. See Appendix D.5 for a more detailed discussion and the proof.

## 3.2 Per-update clipping

Per-update clipping aims to bound the influence of any user in FedAvg by clipping the model update $\Delta_i$ The complete theorem with client sampling and its proof can be found in Appendix C.3. Throughout this section, we assume $L_1 = 0$ and let $L := L_0$ (using the standard smoothness assumption) for clarity of presentation. A complete version can be found in Theorem III in Appendix C.1.

**Theorem II** (per-update clipping). *Under Assumptions 1 to 3, consider Algorithm 2 with stepsizes* $\eta_l \le \frac{1}{32\tau L}$ *and* $\eta_l \eta_g \tau L \le \frac{1}{10}$. *If* $c < \mathcal{O}(\eta_l \sqrt{\tau}\sigma + \eta_l \tau \zeta)$, *then after* $R$ *rounds, it holds that:*

$$\min_{r \in [1,R]} \mathbb{E}[||\nabla f(\mathbf{x}_r)||] \le \mathcal{O}\left( \frac{F_0}{c\eta_g R} + \frac{\sigma}{\sqrt{\tau}} + \zeta \right), \tag{4}$$

*If* $c \ge \Theta(\eta_l \sqrt{\tau}\sigma + \eta_l \tau \zeta)$, *let* $\eta_g \ge 2\sqrt{n}$ *and then it holds that:*

$$\min_{r \in [1,R]} \mathbb{E}[||\nabla f(\mathbf{x}_r)||] \le \mathcal{O}\left( \underbrace{\sqrt{\frac{F_0}{\eta_l \eta_g \tau R}} + \tau L\eta_l \zeta + \sqrt{\frac{\eta_l \eta_g \tau L}{n}}\left(\frac{\sigma}{\sqrt{\tau}} + \zeta\right)}_{\text{FedAvg term}} + \underbrace{\frac{F_0}{c\eta_g R} + \frac{\eta_l \tau}{c}\left(\frac{\sigma^2}{\tau} + \zeta^2\right)}_{\text{clipping term}} \right).$$
$$\tag{5}$$

*where* $F_0 := f(\mathbf{x}_0) - f^\star$.

When the clipping threshold $c$ is larger than $\Theta(\eta_l \sqrt{\tau}\sigma + \eta_l \tau \zeta)$ (Eq.(5)), the exact convergence to any accuracy $\varepsilon$ can be achieved by choosing a sufficiently small $\eta_l$ to reduce the clipping bias, which is in contrast to Algorithm 1 where the convergence is only guaranteed with a sufficiently large clipping threshold. However, when $\eta_l > \Theta\left(\frac{c}{\sqrt{\tau}\sigma + \tau\zeta}\right)$ (Eq.(4)), Algorithm 2 is only guaranteed to converge to a neighborhood of a stationary point, with size $\Omega(\frac{\sigma}{\sqrt{\tau}} + \zeta)$. We illustrate that this neighborhood size is tight with an example following Koloskova et al. (2023) given $\tau = 1$ in Appendix B.

**Comparison to the unclipped FedAvg.** Apart from the additional clipping terms, Eq. 5 contains the standard FedAvg terms (Karimireddy et al., 2019, Theorem V), except for $\sqrt{\frac{\eta_l \eta_g \tau L}{n}}\zeta$. We prove that

it disappears as long as $c$ is large enough such that $c \geq \Theta(\eta_l \sqrt{\tau}\sigma + \eta_l \tau \sqrt{n}\zeta)$ (see Appendix C.3 for more details). Therefore, the convergence rates of unclipped FedAvg can be recovered as $c \to \infty$.

**Extension to differentially private FedAvg.** We here extend the convergence result to differentially-private FedAvg (DP-FedAvg) that protects user-level local privacy(McMahan et al., 2018; Geyer et al., 2017; Zhang et al., 2022). Before sending $\Delta_i$ to the server, we add additional noise such that $\Delta_i = \Delta_i + \mathbf{z}_i$, where $\mathbf{z}_i \sim \mathcal{N}(0, \sigma_{\mathrm{DP}}^2/d\mathbf{I}_d)$. We summarize the convergence rate of DP-FedAvg in Corollary II. Compared with Theorem II, DP-FedAvg has two additional terms that depend on the privacy noise $\sigma_{\mathrm{DP}}^2$ and are decreasing with larger number of clients.

**Privacy discussion:** Corollary III protects user-level privacy, where the *record* that differs between neighboring dataset is the combination of all the training examples from a single client (Ponomareva et al., 2023). Following Abadi et al. (2016), to achieve $(\varepsilon_{\mathrm{DP}}, \delta)$-differential privacy, the variance of the added noise $\sigma_{\mathrm{DP}}^2$ needs to be $\Omega(c^2 d \log(1/\delta) R/\varepsilon_{\mathrm{DP}}^2)$ (full client participation). Compared with Corollary I, $\sigma_{\mathrm{DP}}^2$ does not depend on $\tau$ as clipping is performed every $\tau$ local steps. When $\Theta(\sigma^2/\tau + \zeta^2)$ is small with relatively low privacy-budget, we obtain the optimal privacy-utility bound $\mathcal{O}\left(\frac{d\log(1/\delta)}{n\varepsilon_{\mathrm{DP}}^2}\right)$ after running $R = \Theta\left(\frac{n\varepsilon_{\mathrm{DP}}^2}{d\log(1/\delta)}\right)$ rounds by choosing $c = \Theta\left(\sqrt{\frac{d\log(1/\delta)}{n\varepsilon_{\mathrm{DP}}^2}}\right)$, $\eta_l = \Theta(\frac{1}{\sqrt{n\tau}L})$ and $\eta_g = \Theta(\sqrt{n})$. Due to the more practical bounded variance assumption, our optimal utility bound is the square of the standard local DP utility $\mathcal{O}(\sqrt{\frac{d\log(1/\delta)}{n\varepsilon_{\mathrm{DP}}^2}})$ which assumes bounded gradient. Increasing $n$ can linearly reduce the reached optimal error. Differently from Corollary I, increasing local steps $\tau$ does not impact the number of rounds $R$ to reach the optimal utility trade-off. When $\Theta(\sigma^2/\tau + \zeta^2)$ is large, the utility error increases as $\sigma$ and $\zeta$ increases. See Appendix C.6 for detailed discussions and proofs. While Algorithm 2 can converge to any accuracy, it has to pay the price for larger DP noise as $\sigma_{\mathrm{DP}}$ is proportional to $R$. Consequently, the overall privacy-utility trade-off is no better than Algorithm 1 ($N^2$ does not appear in the denominator due to a stronger privacy notion).

**Corollary II.** *Under Assumptions 1 to 3, consider DP-FedAvg with stepsizes $\eta_l \leq \frac{1}{32\tau L}$, $\eta_l \eta_g \tau L \leq \frac{1}{10}$. Let $F_0 := f(\mathbf{x}_0) - f^\star$. If $c < \mathcal{O}(\eta_l \sqrt{\tau}\sigma + \eta_l \tau \zeta)$, then after $R$ rounds, it holds that:*

$$\min_{r \in [1,R]} \mathbb{E}[||\nabla f(\mathbf{x}_r)||] \leq \mathcal{O}\left(\frac{F_0}{c\eta_g R} + \frac{L\eta_g}{cn}\sigma_{DP}^2 + \frac{\sigma}{\sqrt{\tau}} + \zeta\right). \tag{6}$$

*Suppose $c \geq \Theta(\eta_l \sqrt{\tau}\sigma + \eta_l \tau \zeta)$. Let $\eta_g \geq 2\sqrt{n}$. Then it holds that:*

$$\min_{r \in [1,R]} \mathbb{E}[||\nabla f(\mathbf{x}_r)||] \leq \mathcal{O}\Bigg(\underbrace{\sqrt{\frac{F_0}{\eta_l \eta_g \tau R}} + \tau L\eta_l \zeta + \sqrt{\frac{\eta_l \eta_g \tau L}{n}}\left(\frac{\sigma}{\sqrt{\tau}} + \zeta\right)}_{\text{FedAvg term}}$$
$$+ \underbrace{\frac{F_0}{c\eta_g R} + \frac{\eta_l \tau}{c}\left(\frac{\sigma^2}{\tau} + \zeta^2\right)}_{\text{clipping term}} + \underbrace{\sqrt{\frac{L\eta_g}{n\eta_l \tau}\sigma_{DP}^2} + \frac{L\eta_g}{cn}\sigma_{DP}^2}_{\text{DP noise term}}\Bigg). \tag{7}$$

### 3.3 COMPARISON BETWEEN PER-SAMPLE AND PER-UPDATE CLIPPING

While per-update and per-sample clipping are designed to protect different levels of privacy, they share some similarities from the pure algorithmic point of view as discussed below.

**Special case $\tau = 1$: Accuracy:** when $\tau = 1$, $\eta = \eta_l \eta_g$, and $c_{\mathrm{per\_sample}} = c_{\mathrm{per\_update}}/\eta_l$ given $c_{\mathrm{per\_sample}}$ and $c_{\mathrm{per\_update}}$ as the clipping thresholds, the two algorithms are the same and the clipping bias for both algorithm can be simplified as $\mathcal{O}\left(\min\left(\sigma + \zeta, \frac{\eta_l(\sigma+\zeta)}{c_{\mathrm{per\_update}}}\right)\right)$ (see Appendix E for the proof and the simplification of the convergence rate). Therefore, in this special case, we can adjust $c_{\mathrm{per\_sample}}$ in per-sample clipping to reach any accuracy, which is essentially equivalent to the strategy of adjusting inner stepsize $\eta_l$ (adjusting the magnitude of the local updates before clipping) to reach any accuracy in per-update clipping (Appendix E). **Per-sample clipping recovers clipped mini-batch SGD:** When $\tau = 1$, $\zeta = 0$ and the mini-batch size is one on each client, Theorem I recovers Theorem 3.3 from Koloskova et al. (2023) by discussing the relation between $c$ and $\sigma$. See Appendix E for

the proof. **Privacy-utility trade-off**: When the mini-batch size is one on each client and $\zeta = 0$, Algorithm 1 and 2 with DP-noise are equivalent to the centralized DP-SGD Koloskova et al. (2023) algorithmically (except that one algorithm uses two stepsizes). Note that the DP notion is slightly different from before. See Appendix C.6 and D.5 for more details. In this case, we prove that both algorithms achieve the same privacy-utility bound $\mathcal{O}\left(\frac{d\log(1/\delta)}{N_{\text{total}}^2 \varepsilon_{\text{DP}}^2}\right)$ with $T = \Theta\left(\frac{N_{\text{total}}^2 \varepsilon_{\text{DP}}^2}{Ld\log(1/\delta)}\right)$ iterations, where $N_{\text{total}}$ being the total number of data points. Due to the use of the bounded variance assumption 1, our utility bound is slightly worse (higher order) than existing works (Kifer et al., 2012; Wang et al., 2018) which assume bounded gradient and prove the optimal utility bound as $\mathcal{O}\left(\frac{\sqrt{d\log(1/\delta)}}{N_{\text{total}}\varepsilon_{\text{DP}}}\right)$.

*Arbitrary* **choices of** $c$: We can always adjust the inner stepsize $\eta_l$ in per-update clipping to change the norm of the model update $\Delta_i := -\eta_l \sum \nabla F_i(\mathbf{y}_i)$ such that it might not get clipped and can reach any accuracy. However, we cannot adjust the norm of the mini-batch gradient in per-sample clipping and thus have a chance of converging to the neighborhood (see Appendix B and E for examples).

**Stepsize:** When $L_1 > 0$, we can use a larger stepsize $\eta \leq \mathcal{O}\left(1/((L_0 + cL_1)\tau)\right)$ when $c < M$ in Theorem I than in Theorem III as Theorem III requires the maximum norm of the gradient due to the use of Assumption 3. More detailed discussions can be found in Appendix A.1.1.

**More local steps** $\tau$: Compared to Algorithm 1, larger local step $\tau$ can reduce the influence of the stochastic noise $\sigma^2$ by a factor of $\frac{1}{\tau}$ in Algorithm 2.

### 3.4 PRACTICAL IMPLICATIONS

In practice, Algorithm 1 and 2 are parallel from privacy perspective. Algorithm 1 limits the contribution of (individual) training example on the client model update, whereas Algorithm 2 controls the contribution of all the training examples from a client on the server model update. Therefore, depending on the specific definition of neighboring dataset Ponomareva et al. (2023), we should select different algorithm. We next discuss the practical implications based on our results. We refer McMahan et al. (2018); Zhang et al. (2022); Liu et al. (2022); Yang et al. (2022) for an extensive experimental study of the similar algorithms as we study in this paper.

**Accuracy perspective:** Suppose there is no DP noise. Theorem II suggests that a higher accuracy can be achieved by picking a sufficiently small inner stepsize $\eta_l \leq \mathcal{O}(\frac{c}{\sqrt{\tau}\sigma + \tau\zeta})$. As this can slow down the training, one can pick $\eta_g$ to be as large as $\Theta(1/(\eta_l \tau L))$ to accelerate the training process. For per-sample clipping, we need to set $c \geq \Theta\left(\frac{\sigma^2 + \zeta^2}{\varepsilon}\right)$ to guarantee the convergence to $\varepsilon$-accuracy. We refer to Appendix C.5 and D.4 where more explicit formulas of the stepsizes can be found.

**Privacy perspective:** We provide insights in understanding the impact of the stochastic noise $\sigma^2$ and $\zeta^2$ on the choices of stepsize and clipping threshold for obtaining optimal privacy-utility bound in Appendix D.5 and C.6. While per-update clipping can converge to any accuracy by adjusting the inner stepsize, it is no better than per-sample clipping in terms of the optimal privacy-utility trade-off seeing that the added noise needs to proportional to the number of communication rounds.

**Limitation** The observations in Theorem II suggest that one can improve per-sample clipping by incorporating both inner and outer stepsizes into Algorithm 1 so that the clipping bias can be reduced by decreasing the inner stepsize Addtionally, both algorithms with DP noise can require the maximum norm of the gradient for determining the stepsize, which can be less practical.

## 4 EXPERIMENTAL RESULTS

As the effect of stochastic noise has been thoroughly evaluated experimentally Koloskova et al. (2023), we here mainly focus on demonstrating the impact of the data heterogeneity $\zeta^2$ on the convergence. Our main findings are: 1) when the data heterogeneity is low, per-sample and per-update clipping have similar convergence behaviour2) when the data heterogeneity is high, per-sample clipping can converge to a neighborhood of the target accuracy. However, this neighbourhood size is reducing as clipping threshold $c$ increases 3) Per-update clipping can reach the target accuracy at the cost of

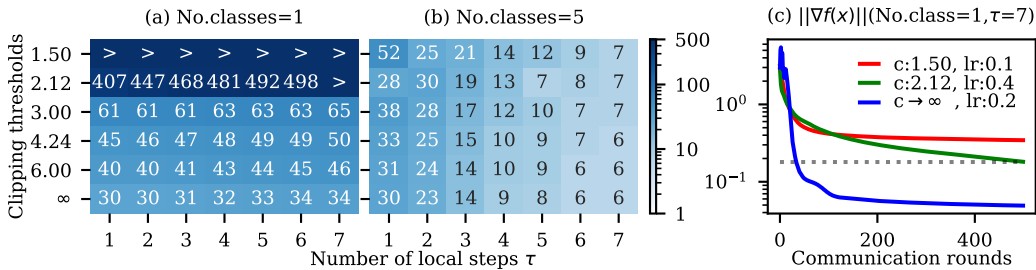

Figure 1: The number of communication rounds to reach target accuracy $||\nabla f(\mathbf{x}_t)|| = 0.18$ (dotted line in (c)) using per-sample clipping. No.classes represents the number of classes in each worker. When the data heterogeneity is high (a), we can hardly reach $\varepsilon$ when $c = 1.5$. However, the neighborhood size $\mathcal{O}\left(\zeta^2/c\right)$ gets smaller as we increase the clipping threshold and we can reach $\varepsilon$ with larger clipping threshold. (c) shows that the gap between when $c \to \infty$ and $c = 1.5$ does not decrease with larger $R$.

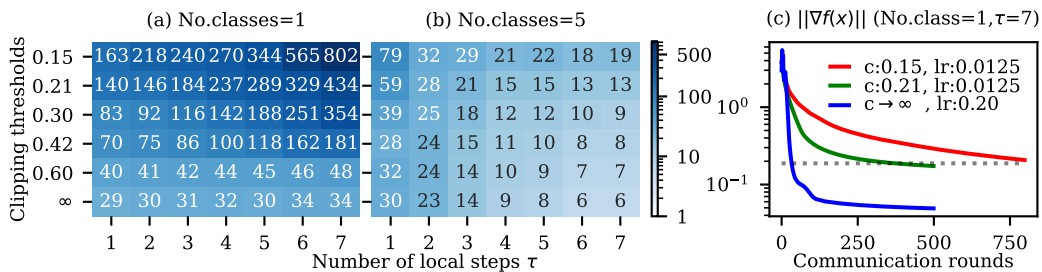

Figure 2: The number of communication rounds to reach target accuracy $||\nabla f(\mathbf{x}_t)|| = 0.18$ (dotted line in (c)) using per-update clipping. Given fixed clipping thresholds and high data heterogeneity (a), we require more rounds to reach the target accuracy as the number of local steps increase. (c) shows that when the clipping threshold is large, we can slowly arrive at the target accuracy eventually.

more communication rounds. See Appendix G for experimental setup and Appendix I for a more complicated NN experiment on CIFAR10 dataset.

**Experimental results:** We tune the stepsize for all the experiments to reach the desired target accuracy $\varepsilon := ||\nabla f(\mathbf{x}_t)||$ with the fewest rounds. We show the required number of communication rounds to reach the target accuracy $\varepsilon = 0.18$ using per-sample clipping in Fig. 1 and per-update clipping in Fig. 2. We observe that when the data heterogeneity is low (Fig. 1 (b) and Fig. 2 (b)), both clipping operations can easily reach the $\varepsilon$ accuracy. However, when the data heterogeneity is high such that each worker only has images from a single class, comparably, we can hardly reach $\varepsilon$ with per-sample clipping when the clipping threshold is 1.50. (Fig. 1 (a)). The gap between when $c = 1.5$ and $c \to \infty$ is clearly non-decreasing as shown in Fig. 1 (c). However, we can reach the target accuracy with a larger clipping threshold. When we use per-update clipping, even when the clipping threshold is small, we can slowly but eventually converge to the required $\varepsilon$ accuracy (Fig. 2(a)).

## 5 CONCLUSION

In this paper, we have rigorously analyzed the impact of two popular clipping strategies: per-sample and per-update clipping, on the convergence of FedAvg under any clipping threshold. Comparably, as per-update clipping interacts with the stepsize, adjusting stepsize accordingly can lead to convergence eventually, but at the cost of more communication rounds. We note that the conclusions drawn in this paper are tied to the specific specifications of the procedures. It is possible that by introducing an additional stepsize in per-sample clipping, a different trade-off between accuracy and convergence rate can be achieved. Our work provide a better theoretical understanding of two clipping strategies in federated learning. We have established a precise impact of data heterogeneity on the convergence of clipping-based FedAvg. As preserving user privacy becomes more important, our work can facilitate the analysis of differential privacy in federated learning and open up future directions for tackling the unfavourable influence of data heterogeneity in the field of privacy-friendly federated learning.

## ACKNOWLEDGEMENT

Bo Li, Mikkel N. Schmidt, and Tommy S. Alstrøm thank for financial support from the European Union's Horizon 2020 research and innovation programme under grant agreement no. 883390 (H2020-SU-SECU-2019 SERSing Project) and Pioneer Centre for AI, DNRF grant number P1. Sebastian Stich thanks for partial financial support from a Meta Privacy Enhancing Technologies Research Award 2022. The authors thank Anastasia Koloskova for the discussion.

## REPRODUCIBILITY STATEMENT

We refer to Appendix G for the implementation of the two clipping methods as well as the hyperparameter choices and software. We refer to Appendix C and D for the detailed proof.

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

# Appendix

## Table of Contents

# A  TECHNICALITIES

## A.1  ASSUMPTIONS

**Assumption A-1** (individual gradient dissimilarity). *If Assumption 2 holds, we assume there exist $\zeta_i^2$ such that $\forall \mathbf{x} \in \mathbb{R}^d$:*

$$||\nabla f_i(\mathbf{x}) - \nabla f(\mathbf{x})||^2 \leq \zeta_i^2, \quad \zeta_{\max}^2 := \max_i \zeta_i^2 \leq n\zeta^2 \ .$$

*Proof.* According to Assumption 2, the maximum of gradient dissimilarity on a single worker is $n\zeta^2$ when we consider the gradient dissimilarity on the rest of $n-1$ workers as 0. Therefore, $\zeta_{\max}^2 \leq n\zeta^2$  □

Note that we use Assumption 2 throughout the proof. We only use Assumption A-1 when we demonstrate that we can recover the convergence rate of the unclipped FedAvg when the clipping threshold is large.

**Proposition 1** (Relation to standard $(L_0, L_1)$-smoothness Zhang et al. (2020b;a)). *If Assumption 2 and the standard $(L_0, L_1)$-smoothness assumption holds, such that $||\nabla f_i(\mathbf{x}) - \nabla f_i(\mathbf{y})|| \leq (L_0 + L_1||\nabla f_i(\mathbf{x})||)||\mathbf{x} - \mathbf{y}||$ for all $\mathbf{x} \in \mathbb{R}^d$ and $||\mathbf{x} - \mathbf{y}|| \leq \frac{1}{L_1}$, then we have:*

$$||\nabla f_i(\mathbf{x}) - \nabla f_i(\mathbf{y})|| \leq \underbrace{(L_0 + L_1||\nabla f_i(\mathbf{x})||)||\mathbf{x} - \mathbf{y}||}_{\text{standard}(L_0,L_1)-\text{smoothness}} \leq (L_0 + L_1\zeta_{\max} + L_1||\nabla f(\mathbf{x})||)||\mathbf{x} - \mathbf{y}|| \ . \quad \text{(A.1)}$$

*Proof.* According to Assumption 2, we have:

$$||\nabla f_i(\mathbf{x})|| \leq ||\nabla f_i(\mathbf{x}) - \nabla f(\mathbf{x})|| + ||\nabla f(\mathbf{x})|| \leq \zeta_{\max} + ||\nabla f(\mathbf{x})|| \ .$$

The first inequality uses triangle inequality. The second inequality uses Assumption A-1. Considering the standard smoothness assumption from Koloskova et al. (2023); Zhang et al. (2020b) for every worker $i$:

$$\begin{aligned}||\nabla f_i(\mathbf{x}) - \nabla f_i(\mathbf{y})|| &\leq (L_0 + L_1||\nabla f_i(\mathbf{x})||)||\mathbf{x} - \mathbf{y}|| \\ &\leq (L_0 + L_1\zeta_{\max} + L_1||\nabla f(\mathbf{x})||)||\mathbf{x} - \mathbf{y}|| \ .\end{aligned}$$

□

**Proposition 2** (Implication of the distributed $(L_0, L_1)$-smoothness). *If Assumption 3 holds, then for any $\mathbf{x}, \mathbf{y} \in \mathbb{R}^d$ with $||\mathbf{x} - \mathbf{y}|| \leq \frac{1}{L_1}$:*

$$||\nabla f_i(\mathbf{x}) - \nabla f_i(\mathbf{y})|| \leq (L_0 + L_1||\nabla f(\mathbf{x})||)||\mathbf{x} - \mathbf{y}||$$

*If the above assumption holds, then:*

$$f_i(\mathbf{y}) \leq f_i(\mathbf{x}) + \langle \nabla f_i(\mathbf{x}), \mathbf{y} - \mathbf{x} \rangle + \frac{L_0 + L_1||\nabla f(\mathbf{x})||}{2}||\mathbf{x} - \mathbf{y}||^2 \ .$$

### A.1.1  DISCUSSION ON THE IMPLICATIONS OF THE DISTRIBUTED $(L_0, L_1)$-SMOOTHNESS

Note that the standard $(L_0, L_1)$-smooth assumption (Zhang et al., 2020b;a) requires $||\mathbf{x} - \mathbf{y}|| \leq \frac{1}{L_1}$ for any $\mathbf{x}, \mathbf{y} \in \mathbb{R}^d$. Consequently, if we run the standard gradient descent **without clipping**: $\mathbf{x}_{t+1} = \mathbf{x}_t - \eta\nabla f(\mathbf{x}_t)$, then it must hold that $\eta = \frac{||\mathbf{x}_{t+1} - \mathbf{x}_t||}{||\nabla f(\mathbf{x}_t)||} \leq \frac{1}{L_1||\nabla f(\mathbf{x}_t)||}$ for any $t \geq 0$. Therefore, to guarantee convergence, the stepsize has to satisfy

$$\eta \leq \frac{1}{L_0 + L_1 \max_t\{||\nabla f(\mathbf{x}_t)||\}} \ .$$

Therefore, setting the stepsize that depends on the maximum gradient norm is, in fact, *necessary* in *some scenarios*. See also the discussion regarding (Zhang et al., 2020b, Assumption 4).

## A.2  SOME TECHNICAL LEMMAS

**Lemma 1.** *For any $\alpha > 0$ and $\mathbf{u} \in \mathbb{R}^d$, the following holds:*

$$-\nabla f(\mathbf{x})^T \mathbf{u} = -\frac{\alpha}{2}||\nabla f(\mathbf{x})||^2 - \frac{1}{2\alpha}||\mathbf{u}||^2 + \frac{1}{2\alpha}||\mathbf{u} - \alpha\nabla f(\mathbf{x})||^2 \ . \quad \text{(A.2)}$$

**Lemma 2** (triangle inequality). *For arbitrary set of $n$ vectors $\{\boldsymbol{v}_i\}_{i=1}^N$ with $\boldsymbol{v}_i \in \mathbb{R}^d$, the following holds:*

$$|| \sum_{i=1}^N \boldsymbol{v}_i ||^2 \leq N \sum_{i=1}^N ||\boldsymbol{v}_i||^2. \tag{A.3}$$

$$||\boldsymbol{v}_i + \boldsymbol{v}_j||^2 \leq (1+\alpha)||\boldsymbol{v}_i||^2 + (1+\alpha^{-1})||\boldsymbol{v}_j||^2, \quad \forall \alpha > 0. \tag{A.4}$$

**Lemma 3** (projection lemma Koloskova et al. (2023)). *Given $c$ as a constant and $\text{clip}_c(\mathbf{x}) := \min\left(1, \frac{c}{||\mathbf{x}||}\right)\mathbf{x}$, we have:*

$$||\text{clip}_c(\mathbf{x}) - \text{clip}_c(\mathbf{y})||^2 \leq ||\mathbf{x} - \mathbf{y}||^2. \tag{A.5}$$

*Proof.* clipping is a projection onto a convex set (ball of radius of $c$), and thus is Lipschitz operation with Lipschitz constant with 1. $\qquad\square$

**Lemma 4** (Implication of Assumption 3). *If each function $f_i$ is $(L_0, L_1)$-smooth, then function $f$ is also $(L_0, L_1)$-smooth such that, $\forall \mathbf{x}, \mathbf{y} \in \mathbb{R}^d$ with $||\mathbf{x} - \mathbf{y}|| \leq \frac{1}{L_1}$, we have:*

$$f(\mathbf{y}) \leq f(\mathbf{x}) + \langle \nabla f(\mathbf{x}), \mathbf{y} - \mathbf{x} \rangle + \frac{L_0 + L_1||\nabla f(\mathbf{x})||}{2}||\mathbf{x} - \mathbf{y}||^2. \tag{A.6}$$

*Proof.* If each function $f_i$ is $(L_0, L_1)$-smooth, we have:

$$f_i(\mathbf{y}) \leq f_i(\mathbf{x}) + \langle \nabla f_i(\mathbf{x}), \mathbf{y} - \mathbf{x} \rangle + \frac{L_0 + L_1||\nabla f(\mathbf{x})||}{2}||\mathbf{x} - \mathbf{y}||^2.$$

Take the expectation w.r.t $i$, we have:

$$\mathbb{E}[f_i(\mathbf{y})] \leq \mathbb{E}[f_i(\mathbf{x})] + \langle \mathbb{E}[\nabla f_i(\mathbf{x})], \mathbf{y} - \mathbf{x} \rangle + \frac{L_0 + L_1||\nabla f(\mathbf{x})||}{2}||\mathbf{x} - \mathbf{y}||^2$$

$$\leq f(\mathbf{x}) + \langle \nabla f(\mathbf{x}), \mathbf{y} - \mathbf{x} \rangle + \frac{L_0 + L_1||\nabla f(\mathbf{x})||}{2}||\mathbf{x} - \mathbf{y}||^2.$$

$\qquad\square$

# B EXAMPLES FOR ILLUSTRATING THE NEIGHBORHOOD SIZE IS TIGHT

We present examples with considering the gradient dissimilarity $\zeta$ given $\tau = 1$ to illustrate the neighborhood size is tight similarly as Koloskova et al. (2023) in the following sections.

## B.1 PER-SAMPLE CLIPPING NEIGHBORHOOD SIZE

We here first provide an example that illustrates the neighborhood size is tight for per-sample clipping, i.e., $\mathbb{E}||\nabla f(\bar{\mathbf{x}}_t)|| = \Omega\left(\min\left(\sigma + \zeta, \frac{\sigma^2 + \zeta^2}{c}\right)\right), \forall t \geq 1$ and show the proof.

**Example 1** (similar to Koloskova et al. (2023)). **Scenario 1** ($\zeta > 0, \sigma = 0, \tau = 1$): `Case 1:` For any $\zeta > 0$, $c < \frac{5}{4}\zeta$, there exists a function $f(\mathbf{x})$ with heterogeneity $\zeta^2$ such that there exists a fixed point $\mathbf{x}^\star$ [1] of Algorithm 1 with $||\nabla f(\mathbf{x}^\star)|| \geq \frac{1}{20}\zeta$. For example, considering $f(\mathbf{x}) = \frac{1}{3}\sum_{i=1}^3 f_i(\mathbf{x})$, where $f_1(\mathbf{x}) = \frac{1}{2}\mathbf{x}^2$, $f_2(\mathbf{x}) = \frac{1}{2}\mathbf{x}^2$, $f_3(\mathbf{x}) = \frac{1}{2}(\mathbf{x} + 3a)^2$ and $a > 0$. We note that $\mathbf{x}^\star = -\frac{c}{2}$ is a fixed point of Algorithm 1 but $\nabla f(\mathbf{x}^\star) > \frac{1}{10}a > \frac{1}{20}\zeta$. `Case 2:` For any $\zeta > 0$, $c \geq \frac{5}{4}\zeta$, there exists a function $f(\mathbf{x})$ with heterogeneity $\zeta^2$ such that there exists a fixed point $\mathbf{x}^\star$ of Algorithm 1 with $||\nabla f(\mathbf{x}^\star)|| \geq \frac{9\zeta^2}{16c}$. For example, considering $f(\mathbf{x}) = \frac{1}{n}\sum_{i=1}^n f_i(\mathbf{x})$, where $f_i(\mathbf{x}) = \frac{1}{2}\mathbf{x}^2$ for $i \leq n-1$ and $f_n(\mathbf{x}) = \frac{1}{2}(\mathbf{x} + na)^2$ with $a > 0$ and $n - 1 = 10\lceil\frac{c^2}{\zeta^2}\rceil$. We note that $\mathbf{x}^\star = -\frac{c}{n-1}$ is a fixed point of Algorithm 1 but $\nabla f(\mathbf{x}^\star) > \frac{\zeta^2}{50c}$. **Scenario 2** ($\zeta = 0, \sigma > 0, \tau = 1$): We can derive the corresponding neighborhood size following Theorem 3.1 and 3.2 from Koloskova et al. (2023).

**Example 1**, **Scenario 1** ($\zeta > 0, \sigma = 0, \tau = 1$), `Case 1:`:

---

[1] By *fixed point*, we mean the algorithm does not move away from $\mathbf{x}^\star$ if it is initialized at $\mathbf{x}^\star$

*Proof.* According to the definition of $\zeta^2$ (Assumption 2), we have $\zeta^2 = \frac{1}{3}\sum_{i=1}^{3}||\nabla f_i(\mathbf{x}) - \nabla f(\mathbf{x})||^2 = \frac{a^2 + a^2 + 4a^2}{3} = 2a^2$. Since we assume $c < \frac{5}{4}\zeta$, we get $c < \frac{5\sqrt{2}}{4}a$. Let $\mathbf{x}^\star = -\frac{c}{2}$. We obtain

$$||\nabla f_2(\mathbf{x}^\star)|| = ||\nabla f_1(\mathbf{x}^\star)|| = ||\mathbf{x}^\star|| = ||-\frac{c}{2}|| < c .$$

and

$$||\nabla f_3(\mathbf{x}^\star)|| = ||\mathbf{x}^\star + 3a|| = ||-\frac{c}{2} + 3a|| > -\frac{c}{2} + \frac{3}{\frac{5}{4}\sqrt{2}}c > c .$$

Therefore, $\nabla f_1(\mathbf{x}^\star)$ and $\nabla f_2(\mathbf{x}^\star)$ are not clipped while $\nabla f_3(\mathbf{x}^\star)$ gets clipped. This implies that

$$\mathbb{E}_i[\text{clip}_c(\nabla f_i(\mathbf{x}^\star))] = -\frac{c}{2} - \frac{c}{2} + c = 0 .$$

Hence, for any stepsize $\eta > 0$, we have $\eta\,\mathbb{E}_i[\text{clip}_c(\nabla f_i(\mathbf{x}^\star))] = 0$ and thus $\mathbf{x}^\star$ is a fixed point of Algorithm 1 (when Algorithm 1 is initialized at $\mathbf{x}^\star$, it will not move). However, the error is as large as the following:

$$||\nabla f(\mathbf{x}^\star)|| = ||\mathbf{x}^\star + a|| = ||-\frac{c}{2} + a|| > -\frac{\frac{5}{4}\sqrt{2}}{2}a + a > \frac{1}{10}a > \frac{1}{20}\zeta .$$

$\square$

**Example 1**, **Scenario 1** ($\zeta > 0, \sigma = 0, \tau = 1$), `Case 2`::

*Proof.* According to the definition of $\zeta^2$ (Assumption 2), we have $\zeta^2 = \frac{1}{n}\sum_{i=1}^{n}||\nabla f_i(\mathbf{x}) - \nabla f(\mathbf{x})||^2 = \frac{(n-1)a^2 + (n-1)^2 a^2}{n}$ and thus $\zeta = \sqrt{n-1}a$. Since we assume that $c \geq \frac{5}{4}\zeta$ and that $n - 1 = 10\lceil\frac{c^2}{\zeta^2}\rceil$. we get $n > 10$. Let $\mathbf{x}^\star = -\frac{c}{n-1}$. We obtain that

$$||\nabla f_i(\mathbf{x}^\star)|| = ||\mathbf{x}^\star|| = ||-\frac{c}{n-1}|| < c, \quad \forall i \leq n-1 .$$

and that

$$||\nabla f_n(\mathbf{x}^\star)|| = ||-\frac{c}{n-1} + na|| .$$

Using our assumption that $n - 1 \geq \frac{c^2}{\zeta^2}$, it holds that

$$\sqrt{n-1} \geq \frac{c}{\zeta} \Rightarrow (n-1)a \geq c \Rightarrow -\frac{c}{n-1} + na \geq c .$$

Equivalently,

$$||\nabla f_n(\mathbf{x}^\star)|| \geq c .$$

Therefore, for any $i \leq n-1$, $\nabla f_i(\mathbf{x}^\star)$ is not clipped while $\nabla f_n(\mathbf{x}^\star)$ gets clipped. This implies that

$$\mathbb{E}_i[\text{clip}_c(\nabla f_i(\mathbf{x}^\star))] = -(n-1)\frac{c}{n-1} + c = 0 .$$

Hence, $\mathbf{x}^\star$ is a fixed point of Algorithm 1. But note that the error can be as large as

$$\begin{aligned}
||\nabla f(x^\star)|| &= -\frac{c}{n-1} + a \\
&= -\frac{c}{n-1} + \frac{\zeta}{\sqrt{n-1}} \\
&= -\frac{c}{10\lceil\frac{c^2}{\zeta^2}\rceil} + \frac{\zeta}{\sqrt{10\lceil\frac{c^2}{\zeta^2}\rceil}} \\
&\geq -\frac{\zeta^2}{10\frac{1}{2}c} + \frac{\zeta^2}{\sqrt{10*2}c} \\
&\geq \frac{\zeta^2}{50c} .
\end{aligned}$$

$\square$

**Example 1**, **Scenario 2** ($\zeta = 0, \sigma > 0, \tau = 1$):

*Proof.* Let us assume for simplicity $f_i = f = F(x, \xi)$ for any $i \in [n]$. By picking the functions indicated in Theorem 3.1 and 3.2 from Koloskova et al. (2023). We end up with the same order of the neighborhood size. $\square$

## B.2 Per-update clipping neighborhood size

We here provide an example to highlight the bias term from Corollary II (Eq. 4) is tight for when $\eta_l \geq \mathcal{O}\left(\frac{c}{\sqrt{\tau}\sigma + \tau\zeta}\right)$

**Example 2** (similar to Koloskova et al. (2023)). **Scenario 1** Let $\tau = 1$ and $\sigma = 0$. For any $\zeta > 0$, $c < \frac{5}{4}\eta_l\zeta$, there exists a function $f(\mathbf{x})$ with heterogeneity $\zeta^2$ such that there exists a fixed point $\mathbf{x}^\star$ of Algorithm 2 with $\|\nabla f(\mathbf{x}^\star)\| \geq \frac{1}{20}\zeta$. For example, considering $f(\mathbf{x}) = \frac{1}{3}\sum_{i=1}^{3} f_i(\mathbf{x})$ where $f_1(\mathbf{x}) = \frac{1}{2}\mathbf{x}^2$, $f_2(\mathbf{x}) = \frac{1}{2}\mathbf{x}^2$, $f_3(\mathbf{x}) = \frac{1}{2}(\mathbf{x} + 3a)^2$ and $a > 0$. We note that $\mathbf{x}^\star = -\frac{c}{2\eta_l}$ is a fixed point of Algorithm 2 but $\nabla f(\mathbf{x}^\star) > \frac{1}{10}a > \frac{1}{20}\zeta$. **Scenario 2** When $\tau = 1$ and $\zeta = 0$, a similar argument can be made with $f_1(\mathbf{x}) = f_2(\mathbf{x}) = f(\mathbf{x})$ where $f(\mathbf{x})$ is defined as above.

**Example 2**, **Scenario 1** $(\zeta > 0, \sigma = 0, \tau = 1)$

*Proof.* According to the definition of $\zeta^2$ (Assumption 2), we have $\zeta^2 = \frac{1}{3}\sum_{i=1}^{3}\|\nabla f_i(\mathbf{x}) - \nabla f(\mathbf{x})\|^2 = \frac{a^2 + a^2 + 4a^2}{3} = 2a^2$. Since we assume $c < \frac{5}{4}\eta_l\zeta$, we get $c < \frac{5\sqrt{2}}{4}a\eta_l$. Let $\mathbf{x}^\star = -\frac{c}{2\eta_l}$. We obtain

$$\|\eta_l\nabla f_2(\mathbf{x}^\star)\| = \|\eta_l\nabla f_1(\mathbf{x}^\star)\| = \|\eta_l\mathbf{x}^\star\| = \|-\frac{c}{2}\| < c .$$

and

$$\|\eta_l\nabla f_3(\mathbf{x}^\star)\| = \|\eta_l\mathbf{x}^\star + 3a\eta_l\| = \|-\frac{c}{2} + 3a\eta_l\| > -\frac{c}{2} + \frac{3}{\frac{5}{4}\sqrt{2}}c > c .$$

Therefore, $\eta_l\nabla f_1(\mathbf{x}^\star)$ and $\eta_l\nabla f_2(\mathbf{x}^\star)$ are not clipped while $\eta_l\nabla f_3(\mathbf{x}^\star)$ gets clipped. This implies that

$$\mathbb{E}_i[\eta_l\text{clip}_c(\nabla f_i(\mathbf{x}^\star))] = -\frac{c}{2} - \frac{c}{2} + c = 0 .$$

Hence, for any outer stepsize $\eta_g > 0$, we have $\eta_g\mathbb{E}_i[\eta_l\text{clip}_c(\nabla f_i(\mathbf{x}^\star))] = 0$ and thus $\mathbf{x}^\star$ is a fixed point of Algorithm 2 (when Algorithm 2 is initialized at $\mathbf{x}^\star$, it will not move). However, the error is as large as the following:

$$\|\nabla f(\mathbf{x}^\star)\| = \|\mathbf{x}^\star + a\| = \|-\frac{c}{2\eta_l} + a\| > -\frac{\frac{5}{4}\sqrt{2}}{2}a + a > \frac{1}{10}a > \frac{1}{20}\zeta .$$

$\square$

**Example 2**, **Scenario 2** $(\zeta = 0, \sigma > 0, \tau = 1)$

*Proof.* Let $f = \frac{1}{2}\sum_{i=1}^{2} f_i(\mathbf{x})$ and $f_i = f$. Further let $f$ be the same function defined in **Example 2**, **Scenario 1**. The same argument can be made with $\zeta$ replaced with $\sigma$. $\square$

## C  PER-UPDATE CLIPPING

In this section, we consider DP-FedAvg algorithm with the following update rule.

$$
\begin{cases}
\mathbf{y}_{i,0}^{r-1} = \mathbf{x}^{r-1}, \mathbf{y}_{i,k}^{r-1} = \mathbf{y}_{i,k-1}^{r-1} - \eta_l \nabla F_i(\mathbf{y}_{i,k-1}^{r-1}) \\
\mathbf{x}^r = \mathbf{x}^{r-1} + \frac{\eta_g}{S} \sum_{i \in S^{r-1}} \left( \mathrm{clip}_c(\mathbf{y}_{i,\tau}^{r-1} - \mathbf{y}_{i,0}^{r-1}) + \mathbf{z}_i^{r-1} \right).
\end{cases}
\tag{C.1}
$$

where $\mathbf{z}_i^{r-1} \sim \mathcal{N}(0, \frac{\sigma_{\mathrm{DP}}^2}{d} \mathbf{I})$ for all $i \in [n]$ and $r \in \mathbb{N}^+$.

In round $r$, we sample $S^{r-1} \subseteq [n]$ clients with $|S^{r-1}| = S$ and then perform $\tau$ local steps starting from the shared model $\mathbf{x}^{r-1}$. Then we clip each local model update and add DP noise. After that, we aggregate the updates from each client and compute the new global parameter $\mathbf{x}^r$.

### C.1  FULL STATEMENT OF THE CONVERGENCE RESULT

In this section, we state our main theorem for DP-FedAvg (C.1) (per-update clipping). Compared with the theorem provided in the main paper, Theorem III includes client sampling as well as the discussion on the case of the large clipping threshold with $c \geq \Theta(\eta_l \sqrt{\tau} \sigma + \eta_l \tau \zeta_{\max})$.

**Theorem III** (Per-update clipping). *Under Assumption 1, 2 and 3, consider DP-FedAvg with update rule* (C.1) *and step sizes* $\eta_l \leq \frac{1}{32\tau L}$, $\eta_l \eta_g \tau L \leq \frac{1}{10}$ *and* $\eta_g \leq \frac{4}{5} \frac{1}{\max_{i,r}\{||\mathbf{z}_i^r||L_1\}}$ *where* $L := L_0 + L_1 \max_{i,k,r}\{||\nabla F_i(\mathbf{y}_{i,k}^{r-1})||, ||\nabla f(\mathbf{x}^{r-1})||\}$. *If* $c < \mathcal{O}(\eta_l \sqrt{\tau} \sigma + \eta_l \tau \zeta)$, *then after $R$ rounds, it holds that:*

$$
\min_{r \in [0,R]} \mathbb{E}[||\nabla f(\mathbf{x}^r)||] \leq \mathcal{O}\left( \frac{f(\mathbf{x}^0) - f^*}{c\eta_g(R+1)} + \frac{L\eta_g}{cS}\sigma_{DP}^2 + \frac{\sigma}{\sqrt{\tau}} + \zeta \right).
\tag{C.2}
$$

*If* $c \geq \Theta(\eta_l \sqrt{\tau} \sigma + \eta_l \tau \zeta)$, *then it holds that:*

$$
\min_{r \in [0,R]} \mathbb{E}[||\nabla f(\mathbf{x}^r)||] \leq \mathcal{O}\left( \frac{f(\mathbf{x}^0) - f^\star}{c\eta_g(R+1)} + \frac{L\eta_g}{cS}\sigma_{DP}^2 + \sqrt{\frac{f(\mathbf{x}^0) - f^\star}{\eta_l \eta_g \tau(R+1)}} + \sqrt{\frac{L\eta_g}{S\eta_l\tau}\sigma_{DP}^2} \right.
$$
$$
\left. \sqrt{\frac{\eta_l \eta_g \tau L}{S}}\left( \frac{\sigma}{\sqrt{\tau}} + \sqrt{\ell}\zeta \right) + \tau L\eta_l\left( \frac{\sigma}{\sqrt{\tau}} + \zeta \right) + \frac{\eta_l \tau}{c}\left( \frac{\sigma^2}{\tau} + \zeta^2 \right) \right),
\tag{C.3}
$$

*where* $\ell = \begin{cases} 1 - \frac{S}{n} & \text{if } c \geq \Theta(\eta_l \sqrt{\tau}\sigma + \eta_l \tau \zeta_{\max}) \\ 1 & \text{otherwise} \end{cases}$.

**Corollary III.** *Under the setting of Theorem III, if* $c \geq \mathcal{O}(\eta_l \sqrt{\tau}\sigma + \eta_l \tau\zeta)$ *and* $\frac{1}{\max_{i,r}\{||\mathbf{z}_i^r||L_1\}} \geq 2\sqrt{S}$, *then with global step size* $\eta_g \geq 2\sqrt{S}$, *it holds that:*

$$
\min_{r \in [0,R]} \mathbb{E}[||\nabla f(\mathbf{x}^r)||] \leq \mathcal{O}\left( \frac{f(\mathbf{x}^0) - f^\star}{c\eta_g(R+1)} + \frac{L\eta_g}{cS}\sigma_{DP}^2 + \sqrt{\frac{f(\mathbf{x}^0) - f^\star}{\eta_l \eta_g \tau(R+1)}} + \sqrt{\frac{L\eta_g}{S\eta_l\tau}\sigma_{DP}^2} \right.
$$
$$
\left. \sqrt{\frac{\eta_l \eta_g \tau L}{S}}\left( \frac{\sigma}{\sqrt{\tau}} + \sqrt{\ell}\zeta \right) + \tau L\eta_l\zeta + \frac{\eta_l \tau}{c}\left( \frac{\sigma^2}{\tau} + \zeta^2 \right) \right).
\tag{C.4}
$$

In Corollary III, if we let $c \to \infty$ and $\sigma_{\mathrm{DP}}^2 = 0$, then $\ell = 1 - \frac{S}{n}$ and thus (C.4) recovers the standard convergence result of FedAvg Karimireddy et al. (2019).

**Discussion on the $L$ parameter**: When $L_1 \neq 0$, $L$ depends on the largest gradient norm. This seems cannot be avoided since Algorithm 2 requires multiple local steps before getting clipped. To ensure a sufficient decrease, the stepsize has to be dependent on the largest gradient norm. See section A.1.1 for detailed explanations.

### C.2  PROOF SKETCH

In section C.3, following Karimireddy et al. (2019), we first provide a lemma (Lemma 5) that quantifies the progress of DP-FedAvg (C.1) for each round where the discussion on different cases for the clipping threshold is also included. Based on this decrease lemma, we give the proof for Theorem III and Corollary III. In section C.4, we provide useful lemmas that we frequently use for the proof of Lemma 5. The main challenge appears in the case when $c$ is large. Note that the local update $\eta_l \sum_{k=1}^{\tau} \nabla F_i(\mathbf{y}_{i,k}^r)$ always has a chance to be clipped since we only assume $\mathbb{E}||\nabla F_i(\mathbf{x}, \xi) - \nabla f_i(\mathbf{x})||^2 \leq \sigma^2$. But $\eta_l \sum_{k=1}^{\tau} \nabla f_i(\mathbf{y}_{i,k}^r)$ can be smaller than a large $c$ almost surely due to the finite sum structure. Hence, one needs to carefully disentangle the effect caused by $\sigma$ and $\zeta$ when estimating the variance term. The key lemma for recovering unclipped FedAvg is provided in Lemma 10.

## C.3 PROOF OF CONVERGENCE

From here on, we use $\mathbb{E}_{r-1}$ to denote the expectation conditioned on all the randomness generated before the round $r$.

**Lemma 5** (one round progress). *Under Assumption 1, 2, and 3, the updates of Algorithm (C.1) with step sizes $\eta_l \leq \frac{1}{32\tau L}$, $\eta_l \eta_g \tau L \leq \frac{1}{10}$ and $\eta_g \leq \frac{4}{5} \frac{1}{\max_{i,r}\{||\mathbf{z}_i^r||L_1\}}$ where $L := L_0 + L_1 \max_{i,k,r}\{||\nabla F_i(\mathbf{y}_{i,k}^{r-1})||, ||\nabla f(\mathbf{x}^{r-1})||\}$ satisfy:*
*case1: $c \leq ||\eta_l \tau \nabla f(\mathbf{x}^{r-1})||$, $||\eta_l \tau \nabla f(\mathbf{x}^{r-1})|| \geq 64\eta_l \sqrt{\tau}\sigma + 64\eta_l \tau \zeta$:*

$$\mathbb{E}[||\nabla f(\mathbf{x}^{r-1})||] \leq \frac{20(\mathbb{E}[f(\mathbf{x}^{r-1})] - \mathbb{E}[f(\mathbf{x}^r)])}{c\eta_g} + \frac{10L\eta_g}{cS}\sigma_{DP}^2 \ . \tag{C.5}$$

*case2: $\frac{c}{2} \leq ||\eta_l \tau \nabla f(\mathbf{x}^{r-1})|| \leq c$ and $c \geq 64\eta_l \sqrt{\tau}\sigma + 64\eta_l \tau \zeta$:*

$$\mathbb{E}[||\nabla f(\mathbf{x}^{r-1})||^2] \leq \frac{4(\mathbb{E}[f(\mathbf{x}^{r-1})] - \mathbb{E}[f(\mathbf{x}^r)])}{\eta_g \eta_l \tau} + \frac{2L\eta_g}{S\eta_l \tau}\sigma_{DP}^2 \ . \tag{C.6}$$

*case3: $||\eta_l \tau \nabla f(\mathbf{x}^{r-1})|| \leq \frac{c}{2}$ and $c \geq 64\eta_l \sqrt{\tau}\sigma + 64\eta_l \tau \zeta$:*

$$\begin{aligned}
\mathbb{E}[||\nabla f(\mathbf{x}^{r-1})||^2] \leq & \frac{4(\mathbb{E}[f(\mathbf{x}^{r-1})] - \mathbb{E}[f(\mathbf{x}^r)])}{\eta_g \eta_l \tau} + \frac{2L\eta_g}{S\eta_l \tau}\sigma_{DP}^2 + (192\tau^2 L^2 \eta_l^2)(\frac{\sigma^2}{\tau} + \zeta^2) \\
& + \frac{16\eta_l \eta_g \tau L}{S}(\frac{\sigma^2}{\tau} + \ell\zeta^2) + \frac{2720\eta_l^2 \tau^2}{c^2}(\frac{\sigma^4}{\tau^2} + \zeta^4) \ .
\end{aligned} \tag{C.7}$$

*where $\ell = \begin{cases} 1 - \frac{S}{n} & \text{if } c \geq 64\eta_l \sqrt{\tau}\sigma + 64\eta_l \tau \zeta_{\max} \\ 1 & \text{otherwise} \end{cases}$.*

*Proof.* Let $\mathbf{u}_i^{r-1} := \text{clip}_c(\eta_l \sum_{k=1}^{\tau} \nabla F_i(\mathbf{y}_{i,k-1}^{r-1}))$. Using the upper bound of $\eta_l$ and $\eta_g$, we have $||\frac{1}{S}\sum_{i\in[S]}\mathbf{u}_i^{r-1}|| \leq \frac{1}{S}\sum_{i\in[S]}||\eta_l \sum_{k=1}^{\tau}\nabla F_i(\mathbf{y}_{i,k-1}^{r-1})|| \leq \frac{1}{32L_1}$ and $||\eta_g \frac{1}{S}\sum_{i\in[S]}\mathbf{z}_i^{r-1}|| \leq \frac{4}{5L_1}$. It follows that: $\eta_g||\frac{1}{S}\sum_{i\in[S]}(\mathbf{u}_i^{r-1} + \mathbf{z}_i^{r-1})|| \leq \frac{1}{L_1}$. Plugging the update rule (C.1) into the definition of $(L_0, L_1)$-smoothness 3 and let $L^{r-1} := L_0 + L_1||\nabla f(\mathbf{x}^{r-1})||$, we have:

$$f(\mathbf{x}^r) \leq f(\mathbf{x}^{r-1}) - \eta_g \langle \nabla f(\mathbf{x}^{r-1}), \frac{1}{S}\sum_{i\in[S]}(\mathbf{u}_i^{r-1} + \mathbf{z}_i^{r-1})\rangle + \eta_g^2 \frac{L^{r-1}}{2}||\frac{1}{S}\sum_{i\in[S]}(\mathbf{u}_i^{r-1} + \mathbf{z}_i^{r-1})||^2 \ . \tag{C.8}$$

Using the fact that $\mathbf{z}_i^{r-1}$ is Gaussian noise with mean zero and variance $\sigma_{\text{DP}}^2$, we can take expectation on both sides conditioned on filtration $r-1$ and use the fact that $L \geq L^{r-1}$ to obtain:

$$\begin{aligned}
\mathbb{E}_{r-1}[f(\mathbf{x}^r)] \leq f(\mathbf{x}^{r-1}) &- \eta_g \mathbb{E}_{r-1}[\langle \nabla f(\mathbf{x}^{r-1}), \frac{1}{S}\sum_{i\in[S]}\mathbf{u}_i^{r-1}\rangle] + \\
& \frac{L}{2}\eta_g^2 \mathbb{E}_{r-1}[||\frac{1}{S}\sum_{i\in[S]}\mathbf{u}_i^{r-1}||^2] + \frac{L}{2S}\eta_g^2 \sigma_{\text{DP}}^2 \ .
\end{aligned} \tag{C.9}$$

**Case 1:** $c \leq ||\eta_l \tau \nabla f(\mathbf{x}^{r-1})||$ and $||\eta_l \tau \nabla f(\mathbf{x}^{r-1})|| \geq 64\eta_l \sqrt{\tau}\sigma + 64\eta_l \tau \zeta$. We start by using Lemma 7 to obtain:

$$\mathbb{E}_{r-1}[-\langle \eta_l \tau \nabla f(\mathbf{x}^{r-1}), \frac{1}{S}\sum_{i\in[S]}\mathbf{u}_i^{r-1}\rangle] = \mathbb{E}_{r-1}[-\langle \eta_l \tau \nabla f(\mathbf{x}^{r-1}), \mathbf{u}_i^{r-1}\rangle] \leq -\frac{1}{10}c||\eta_l \tau \nabla f(\mathbf{x}^{r-1})|| \ . \tag{C.10}$$

Since $c \leq \eta_l \tau ||\nabla f(\mathbf{x}^{r-1})||$, it follows that:

$$\mathbb{E}_{r-1}[||\frac{1}{S}\sum_{i\in[S]}\mathbf{u}_i^{r-1}||^2] \leq \frac{1}{S}\sum_{i\in[S]}\mathbb{E}_{r-1}[||\mathbf{u}_i^{r-1}||^2] \leq c^2 \leq c||\eta_l \tau \nabla f(\mathbf{x}^{r-1})|| \ . \tag{C.11}$$

We conclude by plugging (C.10) and (C.11) into (C.9), rearranging and taking expectation.

**Case 2:** $\frac{c}{2} \leq ||\eta_l \tau \nabla f(\mathbf{x}^{r-1})|| \leq c$ and $c \geq 64\eta_l \sqrt{\tau}\sigma + 64\eta_l \tau\zeta$. We use Lemma 1 with $\alpha = 1$ on $-\mathbb{E}_{r-1}[\langle \eta_l \tau \nabla f(\mathbf{x}^{r-1}), \frac{1}{S}\sum_{i\in[S]} \mathbf{u}_i^{r-1}\rangle]$ and this leads to:

$$
-\mathbb{E}_{r-1}[\langle \eta_l \tau \nabla f(\mathbf{x}^{r-1}), \frac{1}{S}\sum_{i\in[S]} \mathbf{u}_i^{r-1}\rangle]
$$

$$
= -\frac{1}{2}||\eta_l \tau \nabla f(\mathbf{x}^{r-1})||^2 - \frac{1}{2}\mathbb{E}_{r-1}[||\frac{1}{S}\sum_{i\in[S]} \mathbf{u}_i^{r-1}||^2] + \frac{1}{2}\mathbb{E}_{r-1}[||\eta_l \tau \nabla f(\mathbf{x}^{r-1}) - \frac{1}{S}\sum_{i\in[S]} \mathbf{u}_i^{r-1}||^2]
$$

$$
\leq -\frac{1}{2}||\eta_l \tau \nabla f(\mathbf{x}^{r-1})||^2 - \frac{1}{2}\mathbb{E}_{r-1}[||\frac{1}{S}\sum_{i\in[S]} \mathbf{u}_i^{r-1}||^2] + \frac{1}{2}\mathbb{E}_{r-1}[||\eta_l \tau \nabla f(\mathbf{x}^{r-1}) - \mathbf{u}_i^{r-1}||^2] .
$$

$$(C.12)$$

We now use Lemma 8 to bound the last term:

$$
\mathbb{E}_{r-1}[||\eta_l \tau \nabla f(\mathbf{x}^{r-1}) - \mathbf{u}_i^{r-1}||^2]
$$

$$
\leq 16\tau^2 L^2 \eta_l^2 ||\eta_l \tau \nabla f(\mathbf{x}^{r-1})||^2 + (16\tau^4 L^2 \eta_l^4 + 2\tau^2 \eta_l^2)\zeta^2 + (8\tau^3 L^2 \eta_l^4 + \tau\eta_l^2)\sigma^2
$$

$$
\leq \frac{1}{64}||\eta_l \tau \nabla f(\mathbf{x}^{r-1})||^2 + \eta_l^2 \tau^2\Big((16\eta_l^2 \tau^2 L^2 + 2)\zeta^2 + (\frac{8\tau^2 L^2 \eta_l^2}{\tau} + \frac{1}{\tau})\sigma^2\Big)
$$

$$
\leq \frac{1}{64}||\eta_l \tau \nabla f(\mathbf{x}^{r-1})||^2 + \eta_l^2 \tau^2(\frac{9}{4}\zeta^2 + \frac{9}{8\tau}\sigma^2)
$$

$$
\leq \frac{1}{64}||\eta_l \tau \nabla f(\mathbf{x}^{r-1})||^2 + \eta_l^2 \tau^2 2\frac{c^2}{64^2 \eta_l^2 \tau^2}
$$

$$
\leq \frac{1}{64}||\eta_l \tau \nabla f(\mathbf{x}^{r-1})||^2 + \eta_l^2 \tau^2 2\frac{4\eta_l^2 \tau^2 ||\nabla f(\mathbf{x}^{r-1})||^2}{64^2 \eta_l^2 \tau^2}
$$

$$
\leq \frac{1}{2}||\eta_l \tau \nabla f(\mathbf{x}^{r-1})||^2 ,
$$

$$(C.13)$$

where we recursively use the assumption that $\eta_l \tau L \leq \frac{1}{32}, c \leq 2||\eta_l \tau \nabla f(\mathbf{x}^{r-1})||$ and $c \geq 64\eta_l \sqrt{\tau}\sigma + 64\eta_l \tau\zeta$.
Plug (C.13), (C.12) into (C.9), we have:

$$
\mathbb{E}_{r-1}[f(\mathbf{x}^r)] \leq f(\mathbf{x}^{r-1}) - \frac{\eta_g \eta_l \tau}{4}||\nabla f(\mathbf{x}^{r-1})||^2 + (\frac{L}{2}\eta_g^2 - \frac{\eta_g}{2\eta_l \tau})\mathbb{E}_{r-1}[||\frac{1}{S}\sum_{i\in[S]} \mathbf{u}_i^{r-1}||^2] + \frac{L}{2S}\eta_g^2 \sigma_{\text{DP}}^2
$$

$$
\leq f(\mathbf{x}^{r-1}) - \frac{\eta_g \eta_l \tau}{4}||\nabla f(\mathbf{x}^{r-1})||^2 + \frac{L}{2S}\eta_g^2 \sigma_{\text{DP}}^2 ,
$$

$$(C.14)$$

where we use the assumption that $\eta_g \eta_l L \tau \leq 1$. Take expectation on both sides and rearrange gives the result.

**Case 3:** $||\eta_l \tau \nabla f(\mathbf{x}^{r-1})|| \leq \frac{c}{2}$ and $c \geq 64\eta_l \sqrt{\tau}\sigma + 64\eta_l \tau\zeta$. We use Lemma 1 with $\alpha = 1$ on $-\langle \eta_l \tau \nabla f(\mathbf{x}^{r-1}), \mathbb{E}_{r-1}[\frac{1}{S}\sum_{i\in[S]} \mathbf{u}_i^{r-1}]\rangle$ and this leads to:

$$
-\mathbb{E}_{r-1}[\langle \eta_l \tau \nabla f(\mathbf{x}^{r-1}), \frac{1}{S}\sum_{i\in[S]} \mathbf{u}_i^{r-1}\rangle]
$$

$$
= -\frac{1}{2}||\eta_l \tau \nabla f(\mathbf{x}^{r-1})||^2 - \frac{1}{2}||\mathbb{E}_{r-1}[\frac{1}{S}\sum_{i\in[S]} \mathbf{u}_i^{r-1}]||^2 + \frac{1}{2}||\eta_l \tau \nabla f(\mathbf{x}^{r-1}) - \mathbb{E}_{r-1}[\frac{1}{S}\sum_{i\in[S]} \mathbf{u}_i^{r-1}]||^2
$$

$$
= -\frac{1}{2}||\eta_l \tau \nabla f(\mathbf{x}^{r-1})||^2 - \frac{1}{2}||\mathbb{E}_{r-1}[\mathbf{u}_i^{r-1}]||^2 + \frac{1}{2}||\eta_l \tau \nabla f(\mathbf{x}^{r-1}) - \mathbb{E}_{r-1}[\mathbf{u}_i^{r-1}]||^2 .
$$

$$(C.15)$$

Besides, we separate the mean and variance of the last term:

$$
\mathbb{E}_{r-1}[||\frac{1}{S}\sum_{i\in[S]} \mathbf{u}_i^{r-1}||^2] = ||\mathbb{E}_{r-1}[\mathbf{u}_i^{r-1}]||^2 + \mathbb{E}_{r-1}[||\frac{1}{S}\sum_{i\in[S]} \mathbf{u}_i^{r-1} - \mathbb{E}_{r-1}[\mathbf{u}_i^{r-1}]||^2] .
$$

$$(C.16)$$

Apply Lemma 10 and 9 and plug (C.15) and (C.16) into (C.9) and use the assumption that $\eta_l \eta_g \tau L \leq \frac{1}{10} \leq \min(\frac{S}{10}, 1)$, we get:

$$
\mathbb{E}_{r-1}[f(\mathbf{x}^r)] - f(\mathbf{x}^{r-1})
$$

$$
\leq -(\frac{\eta_g}{2\eta_l \tau} - \frac{5L\eta_g^2}{2S}||\eta_l \tau \nabla f(\mathbf{x}^{r-1})||^2 + (\frac{L}{2}\eta_g^2 - \frac{\eta_g}{2\eta_l \tau})||\mathbb{E}_{r-1}[\mathbf{u}_i^{r-1}]||^2 + \text{noise}
$$

$$
\leq -\frac{\eta_g}{4\eta_l \tau}||\eta_l \tau \nabla f(\mathbf{x}^{r-1})||^2 + \text{noise} .
$$

$$(C.17)$$

where noise $= A\frac{\sigma^2}{\tau} + B\zeta^2 + C\mathcal{H} + D\sigma_{\mathrm{DP}}^2$ and

$$
\begin{cases}
A = \eta_l\eta_g\tau(48\eta_l^2\tau^2L^2 + 4\frac{\eta_l\eta_g L\tau}{S}) \\
B = \eta_l\eta_g\tau(48\eta_l^2\tau^2L^2 + 4\frac{\ell\eta_l\eta_g L\tau}{S}) \\
C = \eta_l^2\tau^2\left[\frac{\eta_g}{\eta_l\tau} + \frac{L}{S}\eta_g^2\right] = \eta_l\eta_g\tau(1 + \frac{L\eta_l\eta_g\tau}{S}) \le 2\eta_l\eta_g\tau \\
D = \frac{L}{2S}\eta_g^2 = \eta_l\eta_g\tau(\frac{L\eta_g}{2S\eta_l\tau}) \\
\mathcal{H} = \frac{340\eta_l^2\tau^2}{c^2}(\frac{\sigma^4}{\tau^2} + \zeta^4)\,.
\end{cases}
\tag{C.18}
$$

We use the assumption $\eta_l\tau L \le \frac{1}{32}$ in above inequalities. Rearranging and taking expectation gives the result.

$\square$

*Proof of Theorem III.* We first prove the case where $c < \mathcal{O}(\eta_l\sqrt{\tau}\sigma + \eta_l\tau\zeta)$. If $\min_{r\in[0,R]}\|\eta_l\tau\nabla f(\mathbf{x}^r)\| \le 64\eta_l\sqrt{\tau}\sigma + 64\eta_l\tau\zeta$, then the inequality (C.2) trivially holds. Otherwise, we can use (C.5) in Lemma 5. Average over $1 \le r \le R+1$ gives:

$$
\frac{1}{R+1}\sum_{r=1}^{R+1}\mathbb{E}[\|\nabla f(\mathbf{x}^{r-1})\|] \le \frac{20(f(\mathbf{x}^0) - f^*)}{c\eta_g(R+1)} + \frac{10L\eta_g}{cS}\sigma_{\mathrm{DP}}^2\,.
\tag{C.19}
$$

We next prove the case where $c \ge \mathcal{O}(\eta_l\sqrt{\tau}\sigma + \eta_l\tau\zeta)$. Since the upper bound (C.6) in the second case is always better than the third case (C.7), we consider only two situations henceforth. Define $\mathcal{J}_1$ to be the set of indices with $\|\eta_l\tau\nabla f(\mathbf{x}^r)\| > c$ and $\mathcal{J}_2$ to be the set of indices with $\|\eta_l\tau\nabla f(\mathbf{x}^r)\| \le c$.

$$
\begin{aligned}
&\frac{1}{R+1}\Big(\sum_{(r-1)\in\mathcal{J}_1} c\,\mathbb{E}[\|\nabla f(\mathbf{x}^{r-1})\|] + \sum_{(r-1)\in\mathcal{J}_2}\eta_l\tau\,\mathbb{E}[\|\nabla f(\mathbf{x}^{r-1})\|^2]\Big) \\
&\le \frac{20(f(\mathbf{x}^0) - f^*)}{\eta_g(R+1)} + \frac{12L\eta_g}{S}\sigma_{\mathrm{DP}}^2 + \eta_l\tau(\frac{16\eta_l\eta_g\tau L}{S})(\ell\zeta^2 + \frac{\sigma^2}{\tau}) \\
&\quad + \eta_l\tau(192\tau^2L^2\eta_l^2)(\zeta^2 + \frac{\sigma^2}{\tau}) + \eta_l\tau\frac{2720\eta_l^2\tau^2}{c^2}(\frac{\sigma^4}{\tau^2} + \zeta^4)\,.
\end{aligned}
\tag{C.20}
$$

This implies that:

$$
\begin{aligned}
&\frac{1}{R+1}\sum_{(r-1)\in\mathcal{J}_1}\mathbb{E}[\|\nabla f(\mathbf{x}^{r-1})\|] \\
&\le \frac{20(f(\mathbf{x}^0) - f^*)}{c\eta_g(R+1)} + \frac{12L\eta_g}{cS}\sigma_{\mathrm{DP}}^2 + \frac{\eta_l\tau}{c}(\frac{16\eta_l\eta_g\tau L}{S})(\ell\zeta^2 + \frac{\sigma^2}{\tau}) \\
&\quad + \frac{\eta_l\tau}{c}(192\tau^2L^2\eta_l^2)(\zeta^2 + \frac{\sigma^2}{\tau}) + \frac{\eta_l\tau}{c}\frac{2720\eta_l^2\tau^2}{c^2}(\frac{\sigma^4}{\tau^2} + \zeta^4)\,,
\end{aligned}
\tag{C.21}
$$

and that:

$$
\begin{aligned}
&\frac{1}{R+1}\sum_{(r-1)\in\mathcal{J}_2}\mathbb{E}[\|\nabla f(\mathbf{x}^{r-1})\|^2] \\
&\le \frac{20(f(\mathbf{x}^0) - f^*)}{\eta_l\eta_g\tau(R+1)} + \frac{12L\eta_g}{S\eta_l\tau}\sigma_{\mathrm{DP}}^2 + \frac{16\eta_l\eta_g\tau L}{S}(\ell\zeta^2 + \frac{\sigma^2}{\tau}) \\
&\quad + 192\tau^2L^2\eta_l^2(\zeta^2 + \frac{\sigma^2}{\tau}) + \frac{2720\eta_l^2\tau^2}{c^2}(\frac{\sigma^4}{\tau^2} + \zeta^4)\,.
\end{aligned}
\tag{C.22}
$$

Denote right hand side of (C.22) to be $\mathcal{Q}$. We simplify (C.22) as following:

$$
\begin{aligned}
\frac{|\mathcal{J}_2|}{R+1}\Big(\frac{1}{|\mathcal{J}_2|}\sum_{(r-1)\in\mathcal{J}_2}\sqrt{\mathbb{E}[\|\nabla f(\mathbf{x}^{r-1})\|^2]}\Big)^2 &\le \frac{|\mathcal{J}_2|}{R+1}\frac{1}{|\mathcal{J}_2|}\sum_{(r-1)\in\mathcal{J}_2}\mathbb{E}[\|\nabla f(\mathbf{x}^{r-1})\|^2] \le \mathcal{Q} \\
\frac{1}{|\mathcal{J}_2|}\sum_{(r-1)\in\mathcal{J}_2}\mathbb{E}[\|\nabla f(\mathbf{x}^{r-1})\|] &\le \frac{1}{|\mathcal{J}_2|}\sum_{(r-1)\in\mathcal{J}_2}\sqrt{\mathbb{E}[\|\nabla f(\mathbf{x}^{r-1})\|^2]} \le \sqrt{\mathcal{Q}(R+1)/|\mathcal{J}_2|} \\
\sum_{(r-1)\in\mathcal{J}_2}\mathbb{E}[\|\nabla f(\mathbf{x}^{r-1})\|] &\le \sqrt{\mathcal{Q}(R+1)|\mathcal{J}_2|} \le \sqrt{\mathcal{Q}}(R+1)\,.
\end{aligned}
\tag{C.23}
$$

Combine (C.23) and (C.21), we arrive at:

$$\frac{1}{R+1}\sum_{r=1}^{R+1}\mathbb{E}[||\nabla f(\mathbf{x}^{r-1})|||]$$

$$\leq \frac{20(f(\mathbf{x}^0)-f^\star)}{c\eta_g(R+1)} + \frac{12L\eta_g}{cS}\sigma_{\mathrm{DP}}^2 + \sqrt{\frac{20(f(\mathbf{x}^0)-f^\star)}{\eta_l\eta_g\tau(R+1)}} + \sqrt{\frac{12L\eta_g}{S\eta_l\tau}\sigma_{\mathrm{DP}}^2}$$

$$+ \frac{\eta_l\tau}{c}\frac{16\eta_l\eta_g\tau L}{S}(\frac{\sigma^2}{\tau}+\ell\zeta^2) + \sqrt{\frac{16\eta_l\eta_g\tau L}{S}}(\frac{\sigma}{\sqrt{\tau}}+\sqrt{\ell}\zeta) + \frac{\eta_l\tau}{c}\frac{2720\eta_l^2\tau^2}{c^2}(\frac{\sigma^4}{\tau^2}+\zeta^4)$$

$$+ \frac{\eta_l\tau}{c}192\tau^2L^2\eta_l^2(\frac{\sigma^2}{\tau}+\zeta^2) + 20\tau L\eta_l(\frac{\sigma}{\sqrt{\tau}}+\zeta) + \frac{60\eta_l\tau}{c}(\frac{\sigma^2}{\tau}+\zeta^2) \tag{C.24}$$

$$\leq \frac{20(f(\mathbf{x}^0)-f^\star)}{c\eta_g(R+1)} + \frac{12L\eta_g}{cS}\sigma_{\mathrm{DP}}^2 + \sqrt{\frac{20(f(\mathbf{x}^0)-f^\star)}{\eta_l\eta_g\tau(R+1)}} + \sqrt{\frac{12L\eta_g}{S\eta_l\tau}\sigma_{\mathrm{DP}}^2}$$

$$+ \sqrt{\frac{32\eta_l\eta_g\tau L}{S}}(\frac{\sigma}{\sqrt{\tau}}+\sqrt{\ell}\zeta) + 24\tau L\eta_l(\frac{\sigma}{\sqrt{\tau}}+\zeta) + \frac{100\eta_l\tau}{c}(\frac{\sigma^2}{\tau}+\zeta^2)\,,$$

where we use the assumption that $c \geq 64\eta_l\sqrt{\tau}\sigma + 64\eta_l\tau\zeta$ and $\eta_l \leq \frac{1}{32\tau L}$, which implies $\frac{\eta_l\tau}{c}(\zeta^2+\frac{\sigma^2}{\tau}) \leq \frac{1}{64}(\zeta+\frac{\sigma}{\sqrt{\tau}})$, $\frac{2720\eta_l^2\tau^2}{c^2}(\frac{\sigma^4}{\tau^2}+\zeta^4) \leq (\frac{\sigma^2}{\tau}+\zeta^2)$.

$\square$

*Proof of Corollary III.* It can be shown that $24\tau L\eta_l \leq \sqrt{\frac{32\eta_l\eta_g\tau L}{S}}$ in equation (C.24). $\square$

## C.4 USEFUL LEMMAS

In this section, we provide useful lemmas that we frequently use for the proof of Theorem III.

**Lemma 6** (bounded drift (Karimireddy et al. (2019))). *Under Assumption 1, 2, and 3, for any $r \geq 1$ and $\tau \geq 1$, the updates* (C.1) *with step size $\eta_l \leq \frac{1}{2\tau L}$ where $L := L_0 + L_1\max_{i,k}\{||\nabla F_i(\mathbf{y}_{i,k}^{r-1})||, ||\nabla f(\mathbf{x}^{r-1})||\}$ satisfy:*

$$\sum_{k=1}^{\tau}\mathbb{E}_{r-1}[||\mathbf{y}_{i,k-1}^{r-1}-\mathbf{x}^{r-1}||^2] \leq 8\tau^3\eta_l^2||\nabla f(\mathbf{x}^{r-1})||^2 + 8\tau^3\eta_l^2\zeta^2 + 4\tau^2\eta_l^2\sigma^2\,. \tag{C.25}$$

*Proof.* If $\tau = 1$, then the lemma trivially holds since $\mathbf{y}_{i,0}^{r-1} = \mathbf{x}^{r-1}$ for all $i \in [n]$. Assume $\tau \geq 2$. Recall that $\mathbf{y}_{i,k}^{r-1} = \mathbf{y}_{i,k-1}^{r-1} - \eta_l\nabla F_i(\mathbf{y}_{i,k-1}^{r-1})$, we obtain:

$$\mathbb{E}_{r-1}[||\mathbf{y}_{i,k}^{r-1}-\mathbf{x}^{r-1}||^2]$$

$$= \mathbb{E}_{r-1}[||\mathbf{y}_{i,k-1}^{r-1}-\mathbf{x}^{r-1}-\eta_l\nabla F_i(\mathbf{y}_{i,k-1}^{r-1})||^2]$$

$$\overset{(1)}{\leq} \mathbb{E}_{r-1}[||\mathbf{y}_{i,k-1}^{r-1}-\mathbf{x}^{r-1}-\eta_l\nabla f_i(\mathbf{y}_{i,k-1}^{r-1})||^2] + \eta_l^2\sigma^2$$

$$\overset{(A.4)}{\leq} (1+\frac{1}{\tau-1})\mathbb{E}_{r-1}[||\mathbf{y}_{i,k-1}^{r-1}-\mathbf{x}^{r-1}||^2] + \tau\eta_l^2\mathbb{E}_{r-1}[||\nabla f_i(\mathbf{y}_{i,k-1}^{r-1})||^2] + \eta_l^2\sigma^2$$

$$\leq (1+\frac{1}{\tau-1})\mathbb{E}_{r-1}[||\mathbf{y}_{i,\tau-1}^{r-1}-\mathbf{x}^{r-1}||^2] + 2\tau\eta_l^2\mathbb{E}_{r-1}[||\nabla f_i(\mathbf{y}_{i,k-1}^{r-1})-\nabla f_i(\mathbf{x}^{r-1})||^2]$$

$$+ 2\tau\eta_l^2\mathbb{E}_{r-1}[||\nabla f_i(\mathbf{x}^{r-1})||^2] + \eta_l^2\sigma^2$$

$$\overset{(3)}{\leq} (1+\frac{1}{\tau-1})\mathbb{E}_{r-1}[||\mathbf{y}_{i,k-1}^{r-1}-\mathbf{x}^{r-1}||^2] + 2\tau\eta_l^2(L_0+L_1||\nabla f(\mathbf{x}^{r-1})||)^2\mathbb{E}_{r-1}[||\mathbf{y}_{i,k-1}^{r-1}-\mathbf{x}^{r-1}||^2]$$

$$+ 2\tau\eta_l^2\mathbb{E}_{r-1}[||\nabla f_i(\mathbf{x}^{r-1})||^2] + \eta_l^2\sigma^2$$

$$\overset{(2)}{\leq} (1+\frac{2}{\tau-1})\mathbb{E}_{r-1}[||\mathbf{y}_{i,k-1}^{r-1}-\mathbf{x}^{r-1}||^2] + 2\tau\eta_l^2(||\nabla f(\mathbf{x}^{r-1})||^2+\zeta^2) + \eta_l^2\sigma^2\,, \tag{C.26}$$

where in the second inequality we separate the mean and the variance, in the third inequality, we used Lemma 2, in the fifth inequality, we used Assumption 3 and in the last inequality, we used the upper bound of $\eta_l$ and

Assumption 2. Unrolling the recursion gives:

$$\mathbb{E}_{r-1}[||\mathbf{y}_{i,k}^{r-1} - \mathbf{x}^{r-1}||^2] \leq \sum_{q=0}^{k-1} \left( (2\tau\eta_l^2(||\nabla f(\mathbf{x}^{r-1})||^2 + \zeta^2) + \eta_l^2\sigma^2\right)(1 + \frac{2}{\tau-1})^q \tag{C.27}$$

$$\leq 8\tau^2\eta_l^2||\nabla f(\mathbf{x}^{r-1})||^2 + 8\tau^2\eta_l^2\zeta^2 + 4\tau\eta_l^2\sigma^2 ,$$

where in the last inequality, we use the fact that: $\sum_{q=0}^{k-1}(1 + \frac{2}{\tau-1})^q = \frac{(1+\frac{2}{\tau-1})^k - 1}{2/(\tau-1)} \leq 4\tau$ for all $k \leq \tau - 1$. Summing over $k$ gives:

$$\sum_{k=1}^{\tau} \mathbb{E}_{r-1}[||\mathbf{y}_{i,k-1}^{r-1} - \mathbf{x}^{r-1}||^2] \leq 8\tau^3\eta_l^2||\nabla f(\mathbf{x}^{r-1})||^2 + 8\tau^3\eta_l^2\zeta^2 + 4\tau^2\eta_l^2\sigma^2 . \tag{C.28}$$

$\square$

**Lemma 7.** *Under Assumption 1, 2, and 3 and suppose that* $c \leq ||\eta_l\tau\nabla f(\mathbf{x}^{r-1})||$ *and that* $||\eta_l\tau\nabla f(\mathbf{x}^{r-1})|| \geq 64\eta_l\sqrt{\tau}\sigma + 64\eta_l\tau\zeta$, *the updates of (C.1) with step size* $\eta_l \leq \frac{1}{32\tau L}$ *where* $L := L_0 + L_1\max_{i,k}\{||\nabla F_i(\mathbf{y}_{i,k}^{r-1})||, ||\nabla f(\mathbf{x}^{r-1})||\}$ *satisfies:*

$$\mathbb{E}_{r-1}[-\langle \eta_l\tau\nabla f(\mathbf{x}^{r-1}), \mathrm{clip}_c(\eta_l\sum_{k=1}^{\tau}\nabla F_i(\mathbf{y}_{i,k-1}^{r-1}))\rangle] \leq -\frac{1}{10}c||\eta_l\tau\nabla f(\mathbf{x}^{r-1})|| . \tag{C.29}$$

*Proof.* Let $\alpha_i^{r-1} = \min(1, \frac{c}{||\eta_l\sum_{k=1}^{\tau}\nabla F_i(\mathbf{y}_{i,k-1}^{r-1})||})$. We can write the above inner product as:

$$-\langle\eta_l\tau\nabla f(\mathbf{x}^{r-1}), \mathrm{clip}_c(\eta_l\sum_{k=1}^{\tau}\nabla F_i(\mathbf{y}_{i,k-1}^{r-1}))\rangle$$

$$= -\alpha_i^{r-1}||\eta_l\tau\nabla f(\mathbf{x}^{r-1})||^2 - \alpha_i^{r-1}\langle\eta_l\tau\nabla f(\mathbf{x}^{r-1}), \eta_l\sum_{k=1}^{\tau}\nabla F_i(\mathbf{y}_{i,k-1}^{r-1}) - \eta_l\tau\nabla f(\mathbf{x}^{r-1})\rangle \tag{C.30}$$

$$\leq -\alpha_i^{r-1}||\eta_l\tau\nabla f(\mathbf{x}^{r-1})||^2 + \alpha_i^{r-1}||\eta_l\tau\nabla f(\mathbf{x}^{r-1})|| \, ||\eta_l\sum_{k=1}^{\tau}\nabla F_i(\mathbf{y}_{i,k-1}^{r-1}) - \eta_l\tau\nabla f(\mathbf{x}^{r-1})|| .$$

We further upper bound $||\eta_l\sum_{k=1}^{\tau}\nabla F_i(\mathbf{y}_{i,k-1}^{r-1}) - \eta_l\tau\nabla f(\mathbf{x}^{r-1})||$ by:

$$||\eta_l\sum_{k=1}^{\tau}\nabla F_i(\mathbf{y}_{i,k-1}^{r-1}) - \eta_l\tau\nabla f(\mathbf{x}^{r-1})||$$

$$\leq ||\eta_l\sum_{k=1}^{\tau}(\nabla F_i(\mathbf{y}_{i,k-1}^{r-1}) - \nabla f(\mathbf{y}_{i,k-1}^{r-1}))|| + ||\eta_l\sum_{k=1}^{\tau}(\nabla f(\mathbf{y}_{i,k-1}^{r-1}) - \nabla f(\mathbf{x}^{r-1}))|| \tag{C.31}$$

$$\overset{(3)}{\leq} ||\eta_l\sum_{k=1}^{\tau}(\nabla F_i(\mathbf{y}_{i,k-1}^{r-1}) - \nabla f(\mathbf{y}_{i,k-1}^{r-1}))|| + \eta_l L\sum_{k=1}^{\tau}||\mathbf{y}_{i,k-1}^{r-1} - \mathbf{x}^{r-1}|| .$$

Next, we show the probability that (C.31) is larger than $8\eta_l\sqrt{\tau}\sigma + 8\eta_l\tau\zeta + \frac{1}{2}\eta_l\tau||\nabla f(\mathbf{x}^{r-1})||$ is small.

$$\Pr(||\eta_l\sum_{k=1}^{\tau}(\nabla F_i(\mathbf{y}_{i,k-1}^{r-1}) - \nabla f(\mathbf{y}_{i,k-1}^{r-1}))|| + \eta_l L\sum_{k=1}^{\tau}||\mathbf{y}_{i,k-1}^{r-1} - \mathbf{x}^{r-1}|| \geq 8\eta_l\sqrt{\tau}\sigma \tag{C.32}$$

$$+ 8\eta_l\tau\zeta + \frac{1}{2}\eta_l\tau||\nabla f(\mathbf{x}^{r-1})||)$$

$$\leq \Pr(2||\eta_l\sum_{k=1}^{\tau}(\nabla F_i(\mathbf{y}_{i,k-1}^{r-1}) - \nabla f(\mathbf{y}_{i,k-1}^{r-1}))||^2 + 2\eta_l^2 L^2\tau\sum_{k=1}^{\tau}||\mathbf{y}_{i,k-1}^{r-1} - \mathbf{x}^{r-1}||^2 \geq$$

$$64\eta_l^2\tau\sigma^2 + 64\eta_l^2\tau^2\zeta^2 + \frac{1}{4}\eta_l^2\tau^2||\nabla f(\mathbf{x}^{r-1})||^2)$$

$$\overset{(2)+(1)}{\leq} \frac{4\eta_l^2\tau(\sigma^2 + \zeta^2) + 2\eta_l^2 L^2\tau\sum_{k=1}^{\tau}\mathbb{E}_{r-1}[||\mathbf{y}_{i,k-1}^{r-1} - \mathbf{x}^{r-1}||^2]}{64\eta_l^2\tau\sigma^2 + 64\eta_l^2\tau^2\zeta^2 + \frac{1}{4}\eta_l^2\tau^2||\nabla f(\mathbf{x}^{r-1})||^2}$$

$$\leq \frac{4\eta_l^2\tau(\sigma^2 + \zeta^2) + 2\eta_l^2 L^2\tau(8\tau^3\eta_l^2||\nabla f(\mathbf{x}^{r-1})||^2 + 8\tau^3\eta_l^2\zeta^2 + 4\tau^2\eta_l^2\sigma^2)}{64\eta_l^2\tau\sigma^2 + 64\eta_l^2\tau^2\zeta^2 + \frac{1}{4}\eta_l^2\tau^2||\nabla f(\mathbf{x}^{r-1})||^2}$$

$$\leq \frac{4\eta_l^2\tau(\sigma^2+\zeta^2)+16\eta_l^4\tau^4 L^2\zeta^2+8\eta_l^4\tau^3 L^2\sigma^2}{64\eta_l^2\tau\sigma^2+64\eta_l^2\tau^2\zeta^2}+\frac{16\eta_l^4\tau^4 L^2||\nabla f(\mathbf{x}^{r-1})||^2}{\frac{1}{4}\eta_l^2\tau^2||\nabla f(\mathbf{x}^{r-1})||^2}$$

$$\leq \frac{1}{16}+\frac{1}{2048}+\frac{1}{64}\leq \frac{2}{25}\ .$$

In the first inequality, we square both sides and use triangular inequality. The second inequality is due to Markov's inequality and Assumption 2 and 1. The third inequality is due to Lemma 6 and the fifth is due to the assumption that $\eta_l\tau L\leq\frac{1}{32}$. Therefore, with high probability, $||\eta_l\sum_{k=1}^\tau\nabla F_i(\mathbf{y}_{i,k-1}^{r-1})-\eta_l\tau\nabla f(\mathbf{x}^{r-1})||$ is upper bounded by $8\eta_l\sqrt{\tau}\sigma+8\eta_l\tau\zeta+\frac{1}{2}\eta_l\tau||\nabla f(\mathbf{x}^{r-1})||$. In this case, we can plug the upper bound into (C.30):

$$-\langle\eta_l\tau\nabla f(\mathbf{x}^{r-1}),\mathrm{clip}_c(\eta_l\sum_{k=1}^\tau\nabla F_i(\mathbf{y}_{i,k-1}^{r-1}))\rangle \tag{C.33}$$

$$\leq -\alpha_i^{r-1}||\eta_l\tau\nabla f(\mathbf{x}^{r-1})||^2+\alpha_i^{r-1}||\eta_l\tau\nabla f(\mathbf{x}^{r-1})||(8\eta_l\sqrt{\tau}\sigma+8\eta_l\tau\zeta+\frac{1}{2}\eta_l\tau||\nabla f(\mathbf{x}^{r-1})||) \tag{C.34}$$

$$\leq -\alpha_i^{r-1}||\eta_l\tau\nabla f(\mathbf{x}^{r-1})||^2+\alpha_i^{r-1}||\eta_l\tau\nabla f(\mathbf{x}^{r-1})||(\frac{1}{8}+\frac{1}{2})\eta_l\tau||\nabla f(\mathbf{x}^{r-1})|| \tag{C.35}$$

$$\leq -\frac{3}{8}\alpha_i^{r-1}||\eta_l\tau\nabla f(\mathbf{x}^{r-1})||^2\ , \tag{C.36}$$

where the second inequality is due to the assumption that $||\eta_l\tau\nabla f(\mathbf{x}^{r-1})||\geq 64\eta_l\sqrt{\tau}\sigma+64\eta_l\tau\zeta$. We can further lower bound $\alpha_i^{r-1}$ by:

$$\alpha_i^{r-1}=\min(1,\frac{c}{||\eta_l\sum_{k=1}^\tau\nabla F_i(\mathbf{y}_{i,k-1}^{r-1})||})$$

$$\geq \min(1,\frac{c}{||\eta_l\sum_{k=1}^\tau\nabla F_i(\mathbf{y}_{i,k-1}^{r-1})-\eta_l\tau\nabla f(\mathbf{x}^{r-1})||+||\eta_l\tau\nabla f(\mathbf{x}^{r-1})||}) \tag{C.37}$$

$$\geq \frac{8c}{13||\eta_l\tau\nabla f(\mathbf{x}^{r-1})||}\ .$$

The last inequality is because of the assumption that $||\eta_l\tau\nabla f(\mathbf{x}^{r-1})||\geq c$. Combine (C.36) and (C.37), we have:

$$-\langle\eta_l\tau\nabla f(\mathbf{x}^{r-1}),\mathrm{clip}_c(\eta_l\sum_{k=1}^\tau\nabla F_i(\mathbf{y}_{i,k-1}^{r-1}))\rangle\leq -\frac{3c}{13}||\eta_l\tau\nabla f(\mathbf{x}^{r-1})||\ . \tag{C.38}$$

In the other case where $||\eta_l\sum_{k=1}^\tau\nabla F_i(\mathbf{y}_{i,k-1}^{r-1})-\eta_l\tau\nabla f(\mathbf{x}^{r-1})||>8\eta_l\sqrt{\tau}\sigma+8\eta_l\tau\zeta+\frac{1}{2}\eta_l\tau||\nabla f(\mathbf{x}^{r-1})||$, we have:

$$-\langle\eta_l\tau\nabla f(\mathbf{x}^{r-1}),\mathrm{clip}_c(\eta_l\sum_{k=1}^\tau\nabla F_i(\mathbf{y}_{i,k-1}^{r-1}))\rangle\leq c||\eta_l\tau\nabla f(\mathbf{x}^{r-1})||\ . \tag{C.39}$$

We conclude by using the law of total expectation:

$$\mathbb{E}_{r-1}[-\langle\eta_l\tau\nabla f(\mathbf{x}^{r-1}),\mathrm{clip}_c(\eta_l\sum_{k=1}^\tau\nabla F_i(\mathbf{y}_{i,k-1}^{r-1}))\rangle]\leq c||\eta_l\tau\nabla f(\mathbf{x}^{r-1})||\frac{2}{25}-\frac{23}{25}\frac{3}{13}c||\eta_l\tau\nabla f(\mathbf{x}^{r-1})||$$

$$\leq -\frac{1}{10}c||\eta_l\tau\nabla f(\mathbf{x}^{r-1})||\ . \tag{C.40}$$

$\square$

**Lemma 8.** *Under Assumption 1, 2, and 3 and suppose that $||\eta_l\tau\nabla f(\mathbf{x}^{r-1})||\leq c$ and that $\eta_l\leq\frac{1}{2\tau L}$ where $L=L_0+L_1\max_{i,k}\{||\nabla F_i(\mathbf{y}_{i,k}^{r-1})||,||\nabla f(\mathbf{x}^{r-1})||\}$, the updates of (C.1) satisfies:*

$$\mathbb{E}_{r-1}[||\mathbf{u}-\eta_l\tau\nabla f(\mathbf{x}^{r-1})||^2]\leq 16\tau^4 L^2\eta_l^4||\nabla f(\mathbf{x}^{r-1})||^2+(16\tau^4 L^2\eta_l^4+2\tau^2\eta_l^2)\zeta^2+(8\tau^3 L^2\eta_l^4+\tau\eta_l^2)\sigma^2\ . \tag{C.41}$$

*where $\mathbf{u}=\mathrm{clip}_c(\eta_l\sum_{k=1}^\tau\nabla F_i(\mathbf{y}_{i,k-1}^{r-1}))$.*

*Proof.* Since $||\eta_l\tau\nabla f(\mathbf{x}^{r-1})||\leq c$, we have that $\eta_l\tau\nabla f(\mathbf{x}^{r-1})=\mathrm{clip}_c(\eta_l\tau\nabla f(\mathbf{x}^{r-1}))$. By the fact that clipping is Lipschitz operator with constant 1, it follows that:

$$\mathbb{E}_{r-1}[||\mathbf{u}-\eta_l\tau\nabla f(\mathbf{x}^{r-1})||^2] \tag{C.42}$$

$$\leq \mathbb{E}_{r-1}[||\eta_l\sum_{k=1}^\tau\nabla F_i(\mathbf{y}_{i,k-1}^{r-1})-\eta_l\tau\nabla f(\mathbf{x}^{r-1})||^2] \tag{C.43}$$

$$\overset{(1)}{\leq} \eta_l^2 \left( \mathbb{E}_{r-1}[|| \sum_{k=1}^{\tau} \nabla f_i(\mathbf{y}_{i,k-1}^{r-1}) - \tau \nabla f(\mathbf{x}^{r-1})||^2] + \tau\sigma^2 \right) \tag{C.44}$$

$$\leq \eta_l^2 \left( \mathbb{E}_{r-1}[|| \sum_{k=1}^{\tau} \nabla f_i(\mathbf{y}_{i,k-1}^{r-1}) - \tau \nabla f_i(\mathbf{x}^{r-1}) + \tau \nabla f_i(\mathbf{x}^{r-1}) - \tau \nabla f(\mathbf{x}^{r-1})||^2] + \tau\sigma^2 \right) \tag{C.45}$$

$$\leq \eta_l^2 \left( 2\mathbb{E}_{r-1}[|| \sum_{k=1}^{\tau} (\nabla f_i(\mathbf{y}_{i,k-1}^{r-1}) - \nabla f_i(\mathbf{x}^{r-1}))||^2 + 2\mathbb{E}_{r-1}[||\tau(\nabla f_i(\mathbf{x}^{r-1}) - \nabla f(\mathbf{x}^{r-1}))||^2] + \tau\sigma^2 \right) \tag{C.46}$$

$$\overset{(3)+(2)}{\leq} \eta_l^2 \left( 2\tau L^2 \sum_{k=1}^{\tau} \mathbb{E}_{r-1}[||\mathbf{y}_{i,k-1}^{r-1} - \mathbf{x}^{r-1}||^2]) + 2\tau^2\zeta^2 + \tau\sigma^2 \right) \tag{C.47}$$

$$\leq \eta_l^2 \left( 2\tau L^2(8\tau^3\eta_l^2||\nabla f(\mathbf{x}^{r-1})||^2 + 8\tau^3\eta_l^2\zeta^2 + 4\tau^2\eta_l^2\sigma^2) + 2\tau^2\zeta^2 + \tau\sigma^2 \right) \tag{C.48}$$

$$\leq 16\tau^4 L^2\eta_l^4(||\nabla f(\mathbf{x}^{r-1})||^2 + \zeta^2) + 8\tau^3 L^2\eta_l^4\sigma^2 + 2\tau^2\eta_l^2\zeta^2 + \tau\eta_l^2\sigma^2 , \tag{C.49}$$

where in the second inequality, we separate the mean and the variance, in the last third inequality, we use assumption 2 and 3 and in the last second inequality, we apply Lemma 6. □

**Lemma 9.** *Under Assumption 1, 2, and 3 and suppose that $||\eta_l\tau\nabla f(\mathbf{x}^{r-1})|| \leq \frac{1}{2}c$, $\eta_l \leq \frac{1}{32\tau L}$ where $L = L_0 + L_1 \max_{i,k}\{||\nabla F_i(\mathbf{y}_{i,k}^{r-1})||, ||\nabla f(\mathbf{x}^{r-1})||\}$ and that $c \geq 64\eta_l\sqrt{\tau}\sigma + 64\eta_l\tau\zeta$, the updates of (C.1) satisfies:*

$$|| \mathbb{E}_{r-1}[\mathbf{u}] - \eta_l\tau\nabla f(\mathbf{x}^{r-1})||^2 \leq \frac{1}{3}\eta_l^2\tau^2||\nabla f(\mathbf{x}^{r-1})||^2 + 96\eta_l^4\tau^4 L^2(\frac{\sigma^2}{\tau} + \zeta^2) + \mathcal{H} . \tag{C.50}$$

*where $\mathbf{u} = \text{clip}_c(\eta_l \sum_{k=1}^{\tau} \nabla F_i(\mathbf{y}_{i,k-1}^{r-1}))$ and higher order term $\mathcal{H} = \frac{340\eta_l^4\tau^4}{c^2}(\frac{\sigma^4}{\tau^2} + \zeta^4)$.*

*Proof.*

$$|| \mathbb{E}_{r-1}[\mathbf{u}] - \eta_l\tau\nabla f(\mathbf{x}^{r-1})||^2$$

$$= || \mathbb{E}_{r-1}[\mathbf{u}] - \eta_l \mathbb{E}_{r-1}[\sum_{k=1}^{\tau} \nabla f_i(\mathbf{y}_{i,k-1}^{r-1})] + \eta_l \mathbb{E}_{r-1}[\sum_{k=1}^{\tau} \nabla f_i(\mathbf{y}_{i,k-1}^{r-1})] - \eta_l\tau\nabla f(\mathbf{x}^{r-1})||^2$$

$$\leq 2|| \mathbb{E}_{r-1}[\mathbf{u}] - \eta_l \mathbb{E}_{r-1}[\sum_{k=1}^{\tau} \nabla f_i(\mathbf{y}_{i,k-1}^{r-1})]||^2 + 2||\eta_l \mathbb{E}_{r-1}[\sum_{k=1}^{\tau} \nabla f_i(\mathbf{y}_{i,k-1}^{r-1})] - \eta_l\tau\nabla f(\mathbf{x}^{r-1})||^2$$

$$= 2|| \mathbb{E}_{r-1}[\mathbf{u}] - \eta_l \mathbb{E}_{r-1}[\sum_{k=1}^{\tau} \nabla f_i(\mathbf{y}_{i,k-1}^{r-1})]||^2 + 2||\eta_l \sum_{k=1}^{\tau} \mathbb{E}_{r-1}[\nabla f_i(\mathbf{y}_{i,k-1}^{r-1}) - \nabla f_i(\mathbf{x}^{r-1})]||^2$$

$$\overset{(3)}{\leq} 2|| \mathbb{E}_{r-1}[\mathbf{u}] - \eta_l \mathbb{E}_{r-1}[\sum_{k=1}^{\tau} \nabla f_i(\mathbf{y}_{i,k-1}^{r-1})]||^2 + 2\tau L^2\eta_l^2 \sum_{k=1}^{\tau} \mathbb{E}_{r-1}[||\mathbf{y}_{i,k-1}^{r-1} - \mathbf{x}^{r-1}||^2]$$

$$\leq 2|| \mathbb{E}_{r-1}[\mathbf{u}] - \eta_l \mathbb{E}_{r-1}[\sum_{k=1}^{\tau} \nabla f_i(\mathbf{y}_{i,k-1}^{r-1})]||^2 + 2\tau L^2\eta_l^2(8\tau^3\eta_l^2||\nabla f(\mathbf{x}^{r-1})||^2 + 8\tau^3\eta_l^2\zeta^2 + 4\tau^2\eta_l^2\sigma^2) , \tag{C.51}$$

where in the second equality, we use the fact that $\mathbb{E}_{r-1}[\nabla f_i(\mathbf{x}^{r-1})] = \nabla f(\mathbf{x}^{r-1})$, in the last second inequality, we use Assumption 3 and apply Jensen's inequality on the squared norm function, and in the last inequality, we apply Lemma 6. We next bound the first term in (C.51) by showing the probability that $\eta_l \sum_{k=1}^{\tau} \nabla F_i(\mathbf{y}_{i,k-1}^{r-1})$ gets clipped is low because $||\eta_l\tau\nabla f(\mathbf{x}^{r-1})||$ is sufficiently smaller than $c$. Note that:

$$||\eta_l \sum_{k=1}^{\tau} \nabla F_i(\mathbf{y}_{i,k-1}^{r-1})||$$

$$\leq \eta_l|| \sum_{k=1}^{\tau} (\nabla F_i(\mathbf{y}_{i,k-1}^{r-1}) - \nabla f(\mathbf{y}_{i,k-1}^{r-1}))|| + \eta_l|| \sum_{k=1}^{\tau} (\nabla f(\mathbf{y}_{i,k-1}^{r-1}) - \nabla f(\mathbf{x}^{r-1}))|| + \eta_l\tau||\nabla f(\mathbf{x}^{r-1})||$$

$$\overset{(3)}{\leq} \eta_l \left( || \sum_{k=1}^{\tau} (\nabla F_i(\mathbf{y}_{i,k-1}^{r-1}) - \nabla f(\mathbf{y}_{i,k-1}^{r-1}))|| + L \sum_{k=1}^{\tau} ||\mathbf{y}_{i,k-1}^{r-1} - \mathbf{x}^{r-1}|| + \tau||\nabla f(\mathbf{x}^{r-1})|| \right) . \tag{C.52}$$

By the assumption that $||\eta_l \tau \nabla f(\mathbf{x}^{r-1})|| \leq \frac{1}{2}c$, it follows that:

$$||\eta_l \sum_{k=1}^{\tau} \nabla F_i(\mathbf{y}_{i,k-1}^{r-1})|| \leq \frac{1}{2}c + \eta_l \left(|| \sum_{k=1}^{\tau} (\nabla F_i(\mathbf{y}_{i,k-1}^{r-1}) - \nabla f(\mathbf{y}_{i,k-1}^{r-1}))|| + L \sum_{k=1}^{\tau} ||\mathbf{y}_{i,k-1}^{r-1} - \mathbf{x}^{r-1}||\right) . \quad \text{(C.53)}$$

Define $\delta^{r-1} = \mathbb{1}\{||\eta_l \sum_{k=1}^{\tau} \nabla F_i(\mathbf{y}_{i,k-1}^{r-1})|| > c\}$. We can compute an upper bound of $\mathbb{E}_{r-1}[\delta^{r-1}]$ by:

$$\mathbb{E}_{r-1}[\delta^{r-1}]$$

$$= \Pr[||\eta_l \sum_{k=1}^{\tau} \nabla F_i(\mathbf{y}_{i,k-1}^{r-1})|| > c]$$

$$\overset{\text{(C.53)}}{\leq} \Pr\Big[|| \sum_{k=1}^{\tau} (\nabla F_i(\mathbf{y}_{i,k-1}^{r-1}) - \nabla f(\mathbf{y}_{i,k-1}^{r-1}))|| + L \sum_{k=1}^{\tau} ||\mathbf{y}_{i,k-1}^{r-1} - \mathbf{x}^{r-1}|| > \frac{c}{2\eta_l}\Big]$$

$$\leq \Pr\Big[2|| \sum_{k=1}^{\tau} \nabla F_i(\mathbf{y}_{i,k-1}^{r-1}) - \nabla f(\mathbf{y}_{i,k-1}^{r-1})||^2 + 2L^2\tau \sum_{k=1}^{\tau} ||\mathbf{y}_{i,k-1}^{r-1} - \mathbf{x}^{r-1}||^2 > \frac{c^2}{4\eta_l^2}\Big] \quad \text{(C.54)}$$

$$\leq \frac{4\eta_l^2}{c^2}\Big(2\,\mathbb{E}_{r-1}[|| \sum_{k=1}^{\tau} \nabla F_i(\mathbf{y}_{i,k-1}^{r-1}) - \nabla f(\mathbf{y}_{i,k-1}^{r-1})||^2] + 2L^2\tau \sum_{k=1}^{\tau} \mathbb{E}_{r-1}[||\mathbf{y}_{i,k-1}^{r-1} - \mathbf{x}^{r-1}||^2]\Big)$$

$$\overset{(1)+(2)}{\leq} \frac{4\eta_l^2}{c^2}\Big(4\tau(\sigma^2 + \zeta^2) + 2L^2\tau(8\tau^3\eta_l^2||\nabla f(\mathbf{x}^{r-1})||^2 + 8\tau^3\eta_l^2\zeta^2 + 4\tau^2\eta_l^2\sigma^2)\Big)$$

$$= \frac{16\eta_l^2\tau + 32L^2\tau^3\eta_l^4}{c^2}\sigma^2 + \frac{16\eta_l^2\tau + 64L^2\tau^4\eta_l^4}{c^2}\zeta^2 + 32\eta_l^2\tau^2L^2\frac{\eta_l^2\tau^2||\nabla f(\mathbf{x}^{r-1})||^2}{c^2} .$$

The first inequality is due to (C.53). The last second inequality is due to Markov's inequality and we apply Lemma 6 to the last inequality. Recall that we assume $c \geq 64\eta_l\sqrt{\tau}\sigma + 64\eta_l\tau\zeta$ and therefore we have $c^2 \geq 64^2\eta_l^2\tau\sigma^2 + 64^2\eta_l^2\tau^2\zeta^2$. Using $\eta_l\tau||\nabla f(\mathbf{x}^{r-1})|| \leq \frac{1}{2}c$ and $\eta_l \leq \frac{1}{32\tau L}$, it follows that:

$$\mathbb{E}_{r-1}[\delta^{r-1}] \leq \frac{32}{64^2} + \frac{1}{4*32} \leq \frac{1}{64}, \text{ and } \mathbb{E}_{r-1}[\delta^{r-1}] \leq \frac{17\eta_l^2\tau^2(\frac{\sigma^2}{\tau} + \zeta^2)}{c^2} + 8\eta_l^2\tau^2L^2 . \quad \text{(C.55)}$$

We are now ready to upper bound $|| \mathbb{E}_{r-1}[\mathbf{u}] - \eta_l \mathbb{E}_{r-1}[\sum_{k=1}^{\tau} \nabla f_i(\mathbf{y}_{i,k-1}^{r-1})]||^2$. Plug in the definition of $\mathbf{u}$, we obtain:

$$|| \mathbb{E}_{r-1}[\mathbf{u}] - \eta_l \mathbb{E}_{r-1}[\sum_{k=1}^{\tau} \nabla f_i(\mathbf{y}_{i,k-1}^{r-1})]||^2 \quad \text{(C.56)}$$

$$= ||\eta_l \mathbb{E}_{r-1}[\sum_{k=1}^{\tau} \nabla f_i(\mathbf{y}_{i,k-1}^{r-1})] - \Big(\mathbb{E}_{r-1}[\text{clip}_c(\eta_l \sum_{k=1}^{\tau} \nabla F_i(\mathbf{y}_{i,k-1}^{r-1}))(\delta^{r-1} + 1 - \delta^{r-1})]\Big)||^2 \quad \text{(C.57)}$$

$$= ||\eta_l \mathbb{E}_{r-1}[\sum_{k=1}^{\tau} \nabla f_i(\mathbf{y}_{i,k-1}^{r-1})] - \Big(\mathbb{E}_{r-1}[\frac{\eta_l \sum_{k=1}^{\tau} \nabla F_i(\mathbf{y}_{i,k-1}^{r-1})}{||\eta_l \sum_{k=1}^{\tau} \nabla F_i(\mathbf{y}_{i,k-1}^{r-1})||}c\delta^{r-1} + (1 - \delta^{r-1})\eta_l \sum_{k=1}^{\tau} \nabla F_i(\mathbf{y}_{i,k-1}^{r-1})]\Big)||^2$$

$$\text{(C.58)}$$

$$= || \mathbb{E}_{r-1}[(1 - \frac{c}{||\eta_l \sum_{k=1}^{\tau} \nabla F_i(\mathbf{y}_{i,k-1}^{r-1})||})\eta_l \sum_{k=1}^{\tau} \nabla F_i(\mathbf{y}_{i,k-1}^{r-1})\delta^{r-1}]||^2 \quad \text{(C.59)}$$

$$= || \mathbb{E}_{r-1}[(1 - \frac{c}{||\eta_l \sum_{k=1}^{\tau} \nabla F_i(\mathbf{y}_{i,k-1}^{r-1})||})\eta_l \sum_{k=1}^{\tau} \nabla F_i(\mathbf{y}_{i,k-1}^{r-1})|\delta^{r-1} = 1]\Pr(\delta^{r-1} = 1)||^2 \quad \text{(C.60)}$$

$$\leq \mathbb{E}_{r-1}[||(1 - \frac{c}{||\eta_l \sum_{k=1}^{\tau} \nabla F_i(\mathbf{y}_{i,k-1}^{r-1})||})\eta_l \sum_{k=1}^{\tau} \nabla F_i(\mathbf{y}_{i,k-1}^{r-1})||^2|\delta^{r-1} = 1]\Pr(\delta^{r-1} = 1)^2 \quad \text{(C.61)}$$

$$\leq \mathbb{E}_{r-1}[||\eta_l \sum_{k=1}^{\tau} \nabla F_i(\mathbf{y}_{i,k-1}^{r-1})||^2|\delta^{r-1} = 1]\Pr(\delta^{r-1} = 1)^2 \quad \text{(C.62)}$$

$$= \mathbb{E}_{r-1}[||\eta_l \sum_{k=1}^{\tau} \nabla F_i(\mathbf{y}_{i,k-1}^{r-1})||^2\delta^{r-1}]\Pr(\delta^{r-1} = 1) \quad \text{(C.63)}$$

$$\leq \mathbb{E}_{r-1}[||\eta_l \sum_{k=1}^{\tau} \nabla F_i(\mathbf{y}_{i,k-1}^{r-1})||^2]\Pr(\delta^{r-1} = 1) . \quad \text{(C.64)}$$

In the fourth equality, we apply the law of total expectation. In the last third inequality, we apply Jensen's inequality to the squared norm function and in the last second inequality, we use the fact that

$c \leq ||\eta_l \sum_{k=1}^{\tau} \nabla F_i(\mathbf{y}_{i,k-1}^{r-1})||$ given that $\delta^{r-1} = 1$. The last inequality is due to the fact that $\delta^{r-1} \leq 1$. We next unroll $\mathbb{E}_{r-1}[||\eta_l \sum_{k=1}^{\tau} \nabla F_i(\mathbf{y}_{i,k-1}^{r-1})||^2]$ and it holds that:

$$
\begin{aligned}
&\mathbb{E}_{r-1}[||\eta_l \sum_{k=1}^{\tau} \nabla F_i(\mathbf{y}_{i,k-1}^{r-1})||^2] \\
&\leq 4 \mathbb{E}_{r-1}[||\eta_l \sum_{k=1}^{\tau} \nabla F_i(\mathbf{y}_{i,k-1}^{r-1}) - \nabla f_i(\mathbf{y}_{i,k-1}^{r-1})||^2 + 4 \mathbb{E}_{r-1} ||\eta_l \sum_{k=1}^{\tau} \nabla f_i(\mathbf{y}_{i,k-1}^{r-1}) - \nabla f(\mathbf{y}_{i,k-1}^{r-1})||^2 \\
&\qquad + 4\eta_l^2 \tau \sum_{k=1}^{\tau} \mathbb{E}_{r-1}[||\nabla f(\mathbf{y}_{i,k-1}^{r-1}) - \nabla f(\mathbf{x}^{r-1})||^2] + 4\eta_l^2\tau^2 ||\nabla f(\mathbf{x}^{r-1})||^2 \\
&\leq 4\eta_l^2\tau(\sigma^2 + \zeta^2) + 4\eta_l^2 \tau L^2 \sum_{k=1}^{\tau} \mathbb{E}_{r-1}[||\mathbf{y}_{i,k-1}^{r-1} - \mathbf{x}^{r-1}||^2] + 4\eta_l^2\tau^2 ||\nabla f(\mathbf{x}^{r-1})||^2 \\
&\leq (4\eta_l^2\tau^2 + 32\eta_l^4\tau^4 L^2)||\nabla f(\mathbf{x}^{r-1})||^2 + (4\eta_l^2\tau + 16\eta_l^4\tau^3 L^2)\sigma^2 + (4\eta_l^2\tau + 32\eta_l^4\tau^4 L^2)\zeta^2 .
\end{aligned}
\tag{C.65}
$$

Plug (C.55) and (C.65) into (C.64) and use $\eta_l \leq \frac{1}{32\tau L}$ gives:

$$
|| \mathbb{E}_{r-1}[\mathbf{u}] - \eta_l \mathbb{E}_{r-1}[\sum_{k=1}^{\tau} \nabla f_i(\mathbf{y}_{i,k-1}^{r-1})]||^2 \leq \frac{1}{8}\eta_l^2\tau^2 ||\nabla f(\mathbf{x}^{r-1})||^2 + \frac{170\eta_l^4\tau^4(\frac{\sigma^4}{\tau^2} + \zeta^4)}{c^2} + 40\eta_l^4\tau^4(\frac{\sigma^2}{\tau} + \zeta^2) .
\tag{C.66}
$$

We conclude by plugging (C.66) into (C.51) and using $\eta_l \leq \frac{1}{32\tau L}$. $\qquad\square$

**Lemma 10.** *Under Assumption 1, 2, and 3 and suppose that $||\eta_l\tau\nabla f(\mathbf{x}^{r-1})|| \leq \frac{1}{2}c$, $\eta_l \leq \frac{1}{32\tau L}$ where $L = L_0 + L_1 \max_{i,k}\{||\nabla F_i(\mathbf{y}_{i,k}^{r-1})||, ||\nabla f(\mathbf{x}^{r-1})||\}$ and that $c \geq 64\eta_l\sqrt{\tau}\sigma + 64\eta_l\tau\zeta$, the updates of* (C.1) *satisfies:*

$$
\begin{aligned}
\mathbb{E}_{r-1}[||\frac{1}{S}\sum_{i\in[S]}\mathbf{u}_i^{r-1} - \mathbb{E}_{r-1}[\mathbf{u}_i^{r-1}]||^2] \leq &\frac{1}{S}[5\eta_l^2\tau^2||\nabla f(\mathbf{x}^{r-1})||^2 + 8\eta_l^2\tau^2\frac{\sigma^2}{\tau} \\
&+ 8\ell\eta_l^2\tau^2\zeta^2 + \frac{680\eta_l^4\tau^4}{c^2}(\frac{\sigma^4}{\tau^2} + \zeta^4) ,
\end{aligned}
\tag{C.67}
$$

*where $\mathbf{u}_i^{r-1} := \text{clip}_c(\eta_l \sum_{k=1}^{\tau} \nabla F_i(\mathbf{y}_{i,k-1}^{r-1}))$ and $\ell = \begin{cases} 1 - \frac{S}{n} & \text{if } c \geq 64\eta_l\sqrt{\tau}\sigma + 64\eta_l\tau\zeta_{\max} \\ 1 & \text{otherwise} \end{cases}$.*

*Proof.* **Case1:** $c \geq 64\eta_l\sqrt{\tau}\sigma + 64\eta_l\tau\zeta_{\max}$. We use $E_{\sigma_i}$ to denote taking the expectation conditioned on the randomness $i$ and all the randomness generated before the round $r$. Consider the following split:

$$
\begin{aligned}
&\mathbb{E}_{r-1}[||\frac{1}{S}\sum_{i\in[S]}\mathbf{u}_i^{r-1} - \mathbb{E}_{r-1}[\mathbf{u}_i^{r-1}]||^2] \\
&= \mathbb{E}_{r-1}[||\frac{1}{S}\sum_{i\in[S]}\mathbf{u}_i^{r-1} - \frac{1}{S}\sum_{i\in[S]}\mathbb{E}_{\sigma_i}[\mathbf{u}_i^{r-1}] + \frac{1}{S}\sum_{i\in[S]}\mathbb{E}_{\sigma_i}[\mathbf{u}_i^{r-1}] - \mathbb{E}_{r-1}[\mathbf{u}_i^{r-1}]||^2] \\
&\leq 2\mathbb{E}_{r-1}[||\frac{1}{S}\sum_{i\in[S]}\mathbf{u}_i^{r-1} - \frac{1}{S}\sum_{i\in[S]}\mathbb{E}_{\sigma_i}[\mathbf{u}_i^{r-1}]||^2] + 2\mathbb{E}_{r-1}[||\frac{1}{S}\sum_{i\in[S]}\mathbb{E}_{\sigma_i}[\mathbf{u}_i^{r-1}] - \mathbb{E}_{r-1}[\mathbf{u}_i^{r-1}]||^2] \\
&\leq \frac{2}{S}\mathbb{E}_{r-1}[||\mathbf{u}_i^{r-1} - \mathbb{E}_{\sigma_i}[\mathbf{u}_i^{r-1}]||^2] + 2(1 - \frac{S}{n})\frac{1}{S}\mathbb{E}_{r-1}[||\mathbb{E}_{\sigma_i}[\mathbf{u}_i^{r-1}] - \mathbb{E}_{r-1}[\mathbf{u}_i^{r-1}]||^2] ,
\end{aligned}
\tag{C.68}
$$

where in the last inequality, we used the fact that:

$$
\mathbb{E}_{r-1}[\mathbb{E}_{\sigma_i,\sigma_j}[\langle \mathbf{u}_i^{r-1} - \mathbb{E}_{\sigma_i}[\mathbf{u}^{r-1}], \mathbf{u}_j^{r-1} - \mathbb{E}_{\sigma_j}[\mathbf{u}^{r-1}]\rangle]]] = 0 ,
\tag{C.69}
$$

and the property of sampling without replacement. We next bound the terms from (C.68) separately.

$$
\begin{aligned}
&\mathbb{E}_{r-1}[||\mathbf{u}_i^{r-1} - \mathbb{E}_{\sigma_i}[\mathbf{u}_i^{r-1}]||^2] \\
&= \mathbb{E}_{r-1}[||\mathbf{u}_i^{r-1} - \eta_l \sum_{k=1}^{\tau} \nabla f_i(\mathbf{y}_{i,k-1}^{r-1}) + \eta_l \sum_{k=1}^{\tau} \nabla f_i(\mathbf{y}_{i,k-1}^{r-1}) - \mathbb{E}_{\sigma_i}[\mathbf{u}_i^{r-1}]||^2] \\
&\leq 2\mathbb{E}_{r-1}[||\mathbf{u}_i^{r-1} - \eta_l \sum_{k=1}^{\tau} \nabla f_i(\mathbf{y}_{i,k-1}^{r-1})||^2] + 2\mathbb{E}_{r-1}[||\eta_l \sum_{k=1}^{\tau} \nabla f_i(\mathbf{y}_{i,k-1}^{r-1}) - \mathbb{E}_{\sigma_i}[\mathbf{u}_i^{r-1}]||^2] .
\end{aligned}
\tag{C.70}
$$

By our assumption that $c \geq 64\eta_l \tau \zeta_{\max}$, it holds that $||\nabla f_i(\mathbf{x})|| \leq \frac{c}{\eta_l \tau}$ for all $\mathbf{x} \in \mathbb{R}^d$ since $||\nabla f_i(\mathbf{x})|| \leq ||\nabla f_i(\mathbf{x}) - \nabla f(\mathbf{x})|| + ||\nabla f(\mathbf{x})|| \leq \zeta_{\max} + \frac{c}{2\eta_l \tau} \leq \frac{3c}{4\eta_l \tau}$. Therefore, we have: $||\eta_l \sum_{k=1}^{\tau} \nabla f_i(\mathbf{y}_{i,k-1}^{r-1})|| \leq \frac{3c}{4}$. We next follow the same proof technique used in Lemma 8 and 9. The first term can be bounded by:

$$\mathbb{E}_{r-1}[||\mathbf{u}_i^{r-1} - \eta_l \sum_{k=1}^{\tau} \nabla f_i(\mathbf{y}_{i,k-1}^{r-1})||^2]$$

$$\leq \mathbb{E}_{r-1}[||\eta_l \sum_{k=1}^{\tau} \nabla F_i(\mathbf{y}_{i,k-1}^{r-1}) - \eta_l \sum_{k=1}^{\tau} \nabla f_i(\mathbf{y}_{i,k-1}^{r-1})||^2] \tag{C.71}$$

$$\leq \eta_l^2 \tau \sigma^2 \ .$$

To bound the second term in (C.70), we first show that $||\eta_l \sum_{k=1}^{\tau} \nabla F_i(\mathbf{y}_{i,k-1}^{r-1})|| \leq c$ with high probability. Note that $||\eta_l \sum_{k=1}^{\tau} \nabla F_i(\mathbf{y}_{i,k-1}^{r-1})|| \leq ||\eta_l \sum_{k=1}^{\tau} \nabla F_i(\mathbf{y}_{i,k-1}^{r-1}) - \eta_l \sum_{k=1}^{\tau} \nabla f_i(\mathbf{y}_{i,k-1}^{r-1})|| + \frac{3c}{4}$. Define $\delta^{r-1} = \mathbb{1}\{||\eta_l \sum_{k=1}^{\tau} \nabla F_i(\mathbf{y}_{i,k-1}^{r-1})|| > c\}$. We can compute an upper bound of $\mathbb{E}_{\sigma_i}[\delta^{r-1}]$:

$$\mathbb{E}_{\sigma_i}[\delta^{r-1}]$$

$$= \Pr[||\eta_l \sum_{k=1}^{\tau} \nabla F_i(\mathbf{y}_{i,k-1}^{r-1})|| > c]$$

$$\leq \Pr\Big[||\eta_l \sum_{k=1}^{\tau} \nabla F_i(\mathbf{y}_{i,k-1}^{r-1}) - \eta_l \sum_{k=1}^{\tau} \nabla f_i(\mathbf{y}_{i,k-1}^{r-1})|| > \frac{c}{4}\Big] \tag{C.72}$$

$$= \Pr\Big[||\eta_l \sum_{k=1}^{\tau} \nabla F_i(\mathbf{y}_{i,k-1}^{r-1}) - \eta_l \sum_{k=1}^{\tau} \nabla f_i(\mathbf{y}_{i,k-1}^{r-1})||^2 > \frac{c^2}{16}\Big] \leq \frac{16\eta_l^2 \tau \sigma^2}{c^2} \ .$$

It follows that:

$$||\eta_l \sum_{k=1}^{\tau} \nabla f_i(\mathbf{y}_{i,k-1}^{r-1}) - \mathbb{E}_{\sigma_i}[\mathbf{u}_i^{r-1}]||^2$$

$$= ||\eta_l \sum_{k=1}^{\tau} \nabla f_i(\mathbf{y}_{i,k-1}^{r-1}) - \mathbb{E}_{\sigma_i}[\text{clip}_c(\eta_l \sum_{k=1}^{\tau} \nabla F_i(\mathbf{y}_{i,k-1}^{r-1})(\delta^{r-1} + 1 - \delta^{r-1}))]||^2$$

$$= ||\eta_l \sum_{k=1}^{\tau} \nabla f_i(\mathbf{y}_{i,k-1}^{r-1}) - \mathbb{E}_{\sigma_i}[\frac{\eta_l \sum_{k=1}^{\tau} \nabla F_i(\mathbf{y}_{i,k-1}^{r-1})}{||\eta_l \sum_{k=1}^{\tau} \nabla F_i(\mathbf{y}_{i,k-1}^{r-1})||} c\delta^{r-1} + \eta_l \sum_{k=1}^{\tau} \nabla F_i(\mathbf{y}_{i,k-1}^{r-1})(1 - \delta^{r-1})]||^2$$

$$= ||\mathbb{E}_{\sigma_i}[(1 - \frac{c}{||\eta_l \sum_{k=1}^{\tau} \nabla F_i(\mathbf{y}_{i,k-1}^{r-1})||})\eta_l \sum_{k=1}^{\tau} \nabla F_i(\mathbf{y}_{i,k-1}^{r-1})\delta^{r-1}]||^2$$

$$= ||\mathbb{E}_{\sigma_i}[(1 - \frac{c}{||\eta_l \sum_{k=1}^{\tau} \nabla F_i(\mathbf{y}_{i,k-1}^{r-1})||})\eta_l \sum_{k=1}^{\tau} \nabla F_i(\mathbf{y}_{i,k-1}^{r-1})|\delta^{r-1} = 1]\mathbb{E}_{\sigma_i}[\delta^{r-1}]||^2$$

$$\leq \mathbb{E}_{\sigma_i}[||\eta_l \sum_{k=1}^{\tau} \nabla F_i(\mathbf{y}_{i,k-1}^{r-1})||^2]\mathbb{E}_{\delta_i}[\delta^{r-1}] \ .$$

$$\tag{C.73}$$

We can now bound the second term in (C.70) by plugging in (C.73) and (C.72):

$$\mathbb{E}_{r-1}[||\eta_l \sum_{k=1}^{\tau} \nabla f_i(\mathbf{y}_{i,k-1}^{r-1}) - \mathbb{E}_{\sigma_i}[\mathbf{u}_i^{r-1}]||^2] \leq \mathbb{E}_{r-1}[\mathbb{E}_{\sigma_i}[||\eta_l \sum_{k=1}^{\tau} \nabla F_i(\mathbf{y}_{i,k-1}^{r-1})||^2]\mathbb{E}_{\delta_i}[\delta^{r-1}]]$$

$$\leq \mathbb{E}_{r-1}[\mathbb{E}_{\sigma_i}[||\eta_l \sum_{k=1}^{\tau} \nabla F_i(\mathbf{y}_{i,k-1}^{r-1})||^2]\frac{16\eta_l^2 \tau \sigma^2}{c^2}]$$

$$= \frac{16\eta_l^2 \tau \sigma^2}{c^2} \mathbb{E}_{r-1}[||\eta_l \sum_{k=1}^{\tau} \nabla F_i(\mathbf{y}_{i,k-1}^{r-1})||^2]$$

$$\leq \frac{16\eta_l^2 \tau \sigma^2}{c^2}[5\eta_l^2 \tau^2 ||\nabla f(\mathbf{x}^{r-1})||^2 + 5\eta_l^2 \tau^2(\frac{\sigma^2}{\tau} + \zeta^2)] \ ,$$

$$\tag{C.74}$$

where the last inequality is due to (C.65). It remains to bound the second term in (C.68).

$$\mathbb{E}_{r-1}[||\mathbb{E}_{\sigma_i}[\mathbf{u}_i^{r-1}] - \mathbb{E}_{r-1}[\mathbf{u}_i^{r-1}]||^2]$$

$$\leq 2\mathbb{E}_{r-1}[||\mathbb{E}_{\sigma_i}[\mathbf{u}_i^{r-1}] - \eta_l \tau \nabla f(\mathbf{x}^{r-1})||^2] + 2\mathbb{E}_{r-1}[||\eta_l \tau \nabla f(\mathbf{x}^{r-1}) - \mathbb{E}_{r-1}[\mathbf{u}_i^{r-1}]||^2] \tag{C.75}$$

$$\leq 2\mathbb{E}_{r-1}[||\mathbf{u}_i^{r-1} - \eta_l \tau \nabla f(\mathbf{x}^{r-1})||^2] + 2||\eta_l \tau \nabla f(\mathbf{x}^{r-1}) - \mathbb{E}_{r-1}[\mathbf{u}_i^{r-1}]||^2 \ ,$$

where in the last line we used Jensen's inequality since the squared norm is a convex function. Applying Lemma 8 and 9 and wrapping up yields:

$$
\mathbb{E}_{r-1}[||\frac{1}{S}\sum_{i\in[S]}\mathbf{u}_i^{r-1} - \mathbb{E}_{r-1}[\mathbf{u}_i^{r-1}]||^2]
$$

$$
\leq \frac{1}{S}[3\eta_l^2\tau^2||\nabla f(\mathbf{x}^{r-1})||^2 + 4\eta_l^2\tau^2\frac{\sigma^2}{\tau} + \frac{320\eta_l^4\tau^4}{c^2}(\frac{\sigma^4}{\tau^2} + \zeta^4)]
$$

$$
+ (1 - \frac{S}{n})\frac{1}{S}[2\eta_l^2\tau^2||\nabla f(\mathbf{x}^{r-1})||^2 + 4\eta_l^2\tau^2(\frac{\sigma^2}{\tau} + \zeta^2) + \frac{340\eta_l^4\tau^4}{c^2}(\frac{\sigma^4}{\tau^2} + \zeta^4)] \tag{C.76}
$$

$$
\leq \frac{1}{S}[5\eta_l^2\tau^2||\nabla f(\mathbf{x}^{r-1})||^2 + 8\eta_l^2\tau^2\frac{\sigma^2}{\tau} + (1 - \frac{S}{n})8\eta_l^2\tau^2\zeta^2 + \frac{680\eta_l^4\tau^4}{c^2}(\frac{\sigma^4}{\tau^2} + \zeta^4)] .
$$

**Case2:** $64\eta_l\sqrt{\tau}\sigma + 64\eta_l\tau\zeta \leq c \leq 64\eta_l\sqrt{\tau}\sigma + 64\eta_l\tau\zeta_{\max}$.

$$
\mathbb{E}_{r-1}[||\frac{1}{S}\sum_{i\in[S]}\mathbf{u}_i^{r-1} - \mathbb{E}_{r-1}[\mathbf{u}_i^{r-1}]||^2]
$$

$$
\leq \frac{1}{S}\mathbb{E}_{r-1}[||\mathbf{u}_i^{r-1} - \mathbb{E}_{r-1}[\mathbf{u}_i^{r-1}]||^2] \tag{C.77}
$$

$$
\leq \frac{2}{S}\mathbb{E}_{r-1}[||\mathbf{u}_i^{r-1} - \eta_l\tau\nabla f(\mathbf{x}^{r-1})||^2] + \frac{2}{S}\mathbb{E}_{r-1}[||\eta_l\tau\nabla f(\mathbf{x}^{r-1}) - \mathbb{E}_{r-1}[\mathbf{u}_i^{r-1}]||^2] .
$$

Applying Lemma 8 and 9 and merging two cases concludes the proof. $\qquad\square$

## C.5 ACCURACY PERSPECTIVE

In this section, we discuss how to choose the stepsizes to reach $\varepsilon$-accuracy for Algorithm 2 $(\min_{r\in[1,R]}\mathbb{E}[||\nabla f(\mathbf{x}_r)||] \leq \varepsilon)$. According to Theorem II, we obtain the following convergence result when $\eta_l \leq \mathcal{O}(\frac{c}{\sqrt{\tau}\sigma+\tau\zeta})$ (by taking the square on both sides):

$$
\min_{r\in[1,R]}\mathbb{E}[||\nabla f(\mathbf{x}_r)||^2] \leq \mathcal{O}\left(\frac{F_0}{\eta_l\eta_g\tau R} + \tau^2 L^2\eta_l^2\zeta^2 + \frac{\eta_l\eta_g\tau L}{n}\left(\frac{\sigma^2}{\tau} + \zeta^2\right) + \frac{F_0^2}{c^2\eta_g^2 R^2} + \frac{\eta_l^2\tau^2}{c^2}\left(\frac{\sigma^4}{\tau^2} + \zeta^4\right)\right) .
$$

Since the second, the third and the last terms do not depend on $R$, we have to pick $\eta_l$ and $\eta_g$ such that each individual term is less than $\varepsilon^2$. From the second and the last terms, we deduce:

$$
\eta_l \leq \min\left\{\mathcal{O}(\frac{\varepsilon}{\tau L\zeta}), \mathcal{O}\left(\frac{c\varepsilon}{\tau(\frac{\sigma^2}{\tau} + \zeta^2)}\right), \mathcal{O}(\frac{c}{\sqrt{\tau}\sigma+\tau\zeta})\right\} := A . \tag{C.78}
$$

It remains to deal with the third term. Recall that $\eta_g \geq \Theta(\sqrt{n})$ and $\eta_l\eta_g\tau L \leq \mathcal{O}(1)$.

**case1:** $\Theta(\frac{A\sqrt{n}\tau L}{n}(\frac{\sigma^2}{\tau} + \zeta^2)) > \Theta(\varepsilon)$: we need to set

$$
\eta_l \leq \mathcal{O}\left(\frac{\sqrt{n}\varepsilon}{\tau L(\frac{\sigma^2}{\tau} + \zeta^2)}\right), \qquad \eta_g = \Theta(\sqrt{n}) . \tag{C.79}
$$

**case2:** $\Theta(\frac{A\sqrt{n}\tau L}{n}(\frac{\sigma^2}{\tau} + \zeta^2)) \leq \Theta(\varepsilon)$: we can choose $\eta_g$ to be as large as possible:

$$
\eta_l = \Theta(A), \quad \eta_g = \min\left\{\Theta\left(\frac{n\varepsilon}{A\tau L(\frac{\sigma^2}{\tau} + \zeta^2)}\right), \Theta(\frac{1}{A\tau L})\right\} . \tag{C.80}
$$

Finally, we can compute the required communication rounds $R$ by plugging in the respective choices of $\eta_l$ and $\eta_g$ and letting the first and the fourth terms be less than $\varepsilon^2$.

## C.6 PRIVACY-UTILITY DISCUSSION

### C.6.1 LOCAL DP GUARANTEE

Assume that the server might be malicious, DP-FedAvg with update rule C.1 (per-update clipping with DP noise) aims at protecting each user's data by providing the formal DP guarantee for the clipped model updates between every two rounds. Note that, for each client, the neighboring dataset in DP-FedAvg is defined by any alterations made to the data points within that specific dataset. In other words, we can treat the whole dataset of each client as one data point. To achieve the formal DP guarantee, the variance $\sigma_{\mathrm{DP}}^2$ should satisfy the following condition. (Note the DP noise in update rule C.1 is set to $\mathbf{z}_i^r \sim \mathcal{N}(0, \frac{\sigma_{\mathrm{DP}}^2}{d}\mathbf{I})$ for all $i \in [n]$ and $r \in \mathbb{N}$.)

**Theorem 11** (Abadi et al. (2016)). *For any $\varepsilon_{DP} \leq \mathcal{O}(\frac{S^2}{n^2}R)$, $0 < \delta < 1$ and $R > 0$, DP-FedAvg with update rule C.1 achieves $(\varepsilon_{DP}, \delta)$-local differential privacy for any client $i \in [n]$ if we choose*

$$\bar{\sigma}_{DP} = \Omega\left(\frac{cS\sqrt{\log(1/\delta)R}}{n\varepsilon_{DP}}\right). \tag{C.81}$$

*where $\bar{\sigma}_{DP} := \frac{\sigma_{DP}}{\sqrt{d}}$.*

*Proof.* Let $q := \frac{S}{N}$. According to the analysis from Abadi et al. (2016), suppose $\Delta_i$ at round $r$ satisfies $(\varepsilon', \delta')$-DP for any $r = 0, ..., R - 1$ and any $i \in [n]$, then the total privacy guarantee after $R$ rounds is $\left(\mathcal{O}(q\sqrt{R}\varepsilon'), \mathcal{O}(\delta')\right)$. It follows from the Gaussian mechanism (and let $\varepsilon_{DP} = q\sqrt{R}\varepsilon'$, $\delta = \delta'$) that, each $\bar{\sigma}_{DP}$ should satisfy

$$\bar{\sigma}_{DP} = \Omega\left(\frac{\Delta q\sqrt{\log(1/\delta)R}}{\varepsilon_{DP}}\right). \tag{C.82}$$

where $\Delta$ is $\ell_2$-sensitivity of the algorithm output for which the server receives at each communication round, and can be bounded by $2c$ since $||\text{clip}_c(\mathbf{x}) - \text{clip}_c(\mathbf{y})|| \leq 2c$ for any $\mathbf{x}, \mathbf{y} \in \mathbb{R}^d$. $\square$

### C.6.2 OPTIMAL PRIVACY-UTILITY TRADE-OFF AND CHOICES FOR HYPER-PARAMETERS

Based on the privacy budget $\varepsilon_{DP}$ and $\delta$, the ideal approach involves an initial selection of a clipping threshold with the aim of achieving a good privacy and utility trade-off. Subsequently, the noise scale $\sigma_{DP}$ can be set based on Theorem 11. The final step entails choosing appropriate step sizes to attain the optimal iteration complexity. We next provide the theoretical good choices of hyper-parameters including $c$, $\eta_l$ and $\eta_g$ to obtain the optimal privacy-utility trade-off and the communication rounds $R$.

Throughout this section, we assume $\Theta(\frac{\sigma^2}{\tau} + \zeta^2) \geq \Theta(\frac{\sigma}{\sqrt{\tau}} + \zeta)$. A similar argument can also be made if the other way around. Let $A := \frac{\sqrt{F_0 L\nu}}{\sqrt{n}}$, $B := \frac{\sigma^4}{\tau^2} + \zeta^4$ and $N_r := \min_{r \in [1,R]} \mathbb{E}[||\nabla f(\mathbf{x}_r)||^2]$.

Let $c \geq \Theta(\eta_l\sqrt{\tau}\sigma + \eta_l\tau\zeta)$. We recall the constraints on the stepsizes $\eta_l\tau \leq \mathcal{O}(\frac{1}{L})$, $\eta_l\eta_g\tau L \leq \mathcal{O}(1)$ and $\eta_g \geq \Theta(\sqrt{n})$.

Taking the square of (7) on both sides and plugging $\sigma_{DP}^2 = \Theta(\nu c^2 R)$ into the convergence rate, where $\nu = \frac{d\log(1/\delta)}{\varepsilon^2}$ (suppose full client participation, i.e. $S = n$), we obtain

$$N_r \leq \mathcal{O}\left((\frac{F_0}{\eta_l\eta_g\tau R} + \frac{L\eta_g}{n\eta_l\tau}\nu c^2 R) + (\frac{F_0^2}{c^2\eta_g^2 R^2} + \frac{L^2\eta_g^2}{n^2}\nu^2 c^2 R^2) + \eta_l^2 L^2\tau^2\zeta^2 + \frac{\eta_l\eta_g\tau L}{n}(\frac{\sigma^2}{\tau} + \zeta^2) + \frac{\eta_l^2\tau^2}{c^2}B\right). \tag{C.83}$$

We minimize the first four dominating terms w.r.t $\eta_g R$ by picking $R = \Theta\left(\frac{\sqrt{F_0 n}}{c\eta_g\sqrt{L\nu}}\right)$ and (C.83) thus reduces to:

$$N_r \leq \mathcal{O}\left(\frac{c}{\eta_l\tau}A + A^2 + \eta_l^2 L^2\tau^2\zeta^2 + \frac{\eta_l\eta_g\tau L}{n}(\frac{\sigma^2}{\tau} + \zeta^2) + \frac{\eta_l^2\tau^2}{c^2}B\right). \tag{C.84}$$

**Case 1:** $\Theta(A) \geq \Theta(B)$

It is clear that $A^2$ dominates the error terms. Therefore, to reduce the number of rounds $R$, we should pick the largest $c\eta_g$ in the denominator of $R$. Note $\eta_g \leq \mathcal{O}(\frac{1}{\eta_l\tau L})$, by picking $\eta_g = \Theta(\frac{1}{\eta_l\tau L})$, it remains to find the largest number for $\frac{c}{\eta_l\tau L}$. It is clear that we can pick $\frac{c}{\eta_l\tau L}$ as large as $\frac{A}{L}$ from the first term in (C.83). For that, we can pick $\eta_l = \Theta(\frac{1}{\sqrt{n}\tau L})$ and $c = \Theta(\frac{A}{\sqrt{n}})$ so that $\eta_g \geq \Theta(\sqrt{n})$ to finally achieve the guarantee. By these choices of hyper-parameters, the optimal privacy-utility trade-off is then:

$$N_r \leq \mathcal{O}(A^2) = \mathcal{O}\left(\frac{F_0 L\nu}{n}\right) \quad \text{with} \quad R = \Theta\left(\frac{\sqrt{F_0 n}}{A\sqrt{L\nu}}\right) = \Theta\left(\frac{n}{L\nu}\right). \tag{C.85}$$

**Case 2:** $\Theta(A) < \Theta(B)$

To balance $\frac{c}{\eta_l\tau}A$ and $\frac{\eta_l^2\tau^2}{c^2}B$, we let $\frac{c}{\eta_l\tau} = \frac{B^{1/3}}{A^{1/3}}$. It follows that $c\eta_g = \frac{B^{1/3}}{A^{1/3}}\eta_l\tau\eta_g$. Therefore, to reduce the number of communication rounds $R$, we need to maximize $\eta_l\tau\eta_g$ under the constraints of:

$$\eta_l^2 L^2 \tau^2 \zeta^2 \leq B^{1/3} A^{2/3} \quad \text{and} \quad \frac{\eta_l \eta_g \tau L}{n} B^{1/2} \leq B^{1/3} A^{2/3} . \tag{C.86}$$

With additional constraints on the stepsizes: $\eta_g \leq \mathcal{O}(\frac{1}{\eta_l \tau L})$ and $\eta_l \tau \leq \mathcal{O}(\frac{1}{L})$. The largest $\eta_l \eta_g \tau$ we can obtain is $D_1 := \frac{1}{L} \min\left(1, \frac{A^{2/3} n}{B^{\frac{1}{6}}}\right)$. To obtain this bound, we can let $\eta_g = \Theta(\sqrt{n})$ and let $\eta_l = \Theta\left(\frac{1}{\tau} \min\left(\frac{D_1}{\sqrt{n}}, D_2\right)\right)$ where $D_2 := \frac{1}{L} \min\left(\frac{B^{1/6} A^{1/3}}{\zeta}, 1\right)$. By these choices of hyper-parameters, the privacy-utility trade-off is then:

$$N_r \leq \mathcal{O}(B^{1/3} A^{2/3}) \quad \text{with} \quad R = \Theta\left(\frac{\sqrt{F_0 n L}}{B^{1/6} \min\left(\frac{B^{1/6}}{A^{1/3}}, A^{1/3} n\right) \sqrt{\nu}}\right) . \tag{C.87}$$

### C.6.3 IMPLICATION

Let us begin by examining Case 1 characterized by a relatively low privacy budget, with $\sigma$ and $\zeta$ being relatively small. Plugging $\nu = \frac{d \log(1/\delta)}{\varepsilon_{\text{DP}}^2}$ (suppose full client participation, i.e. $S = n$) into C.85, we obtain the optimal privacy-utility trade-off as well as the best communication rounds as follows:

$$\min_{r \in [1,R]} \mathbb{E}[||\nabla f(\mathbf{x}_r)||^2] \leq \mathcal{O}\left(\frac{F_0 L d \log(1/\delta)}{n \varepsilon_{\text{DP}}^2}\right), \qquad R = \Theta\left(\frac{n \varepsilon_{\text{DP}}^2}{L d \log(1/\delta)}\right) , \tag{C.88}$$

by choosing

$$c = \Theta\left(\frac{\sqrt{F_0 L d \log(1/\delta)}}{\sqrt{n} \varepsilon_{\text{DP}}}\right), \quad \eta_l = \Theta\left(\frac{1}{\sqrt{n} \tau L}\right), \quad \eta_g = \Theta(\sqrt{n}) . \tag{C.89}$$

Due to the relaxed assumption on bounded variance, the optimal utility is of order $\mathcal{O}\left(\frac{d \log(1/\delta)}{n \varepsilon_{\text{DP}}^2}\right)$ which is the square of the standard utility $\mathcal{O}\left(\frac{\sqrt{d \log(1/\delta)}}{\sqrt{n} \varepsilon_{\text{DP}}}\right)$ for local DP under the bounded stochastic gradient assumption. Since the ratio between the number of client $n$ and $(\varepsilon_{\text{DP}}, \delta)$ is unchanged, increasing $n$ can still linearly decrease the utility error. Further, the effective stepsize $\eta_l \eta_g = \Theta(\frac{1}{\tau L})$ decrease as $\tau$ increases and $R$ does not depend on $\tau$. This implies that performing $\tau$ local steps is equivalent to using one local step with effective stepsize $\Theta(\frac{1}{L})$. Hence, under the current analysis, we cannot prove effectiveness for doing multiple local steps when the privacy budget $\varepsilon$ and $\delta$ are relatively small.

Let us now consider Case 2 where we have large stochastic noise and heterogeneity. Assume $\sigma$ and $\zeta$ is large enough such that $B^{1/6} \geq A^{2/3} n$. Plugging $\nu = \frac{d \log(1/\delta)}{\varepsilon^2} n$ (suppose full client participation, i.e. $S = n$) into C.87, we obtain the optimal privacy-utility trade-off as well as the best communication rounds as follows:

$$\min_{r \in [1,R]} \mathbb{E}[||\nabla f(\mathbf{x}_r)||^2] \leq \mathcal{O}\left(\frac{(F_0 L d \log(1/\delta))^{\frac{1}{3}}}{n^{\frac{1}{3}} \varepsilon_{\text{DP}}^{\frac{2}{3}}} \left(\frac{\sigma^{\frac{4}{3}}}{\tau^{\frac{2}{3}}} + \zeta^{\frac{4}{3}}\right)\right), \qquad R = \Theta\left(\frac{\varepsilon_{\text{DP}}^{\frac{4}{3}} (F_0 L)^{\frac{1}{3}}}{n^{\frac{1}{3}} (d \log(1/\delta))^{\frac{2}{3}} \left(\frac{\sigma^{\frac{2}{3}}}{\tau^{\frac{1}{3}}} + \zeta^{\frac{2}{3}}\right)}\right),$$
$$\tag{C.90}$$

The utility error increases as $\sigma$ and $\zeta$ become larger. While local steps can reduce the stochasticity coming from the noise $\sigma^2$, the error caused by heterogeneity can not be reduced. However, this phenomenon is not only related to DP training but is also known for the common local SGD methods for solving heterogeneous systems.

### C.6.4 CONNECTION TO CENTRALIZED DP-SGD

While DP-FedAvg with update rule C.1 is a federated learning algorithm that aims at protecting user-level privacy, it is still comparable with centralized DP-SGD from a pure algorithmic point of view when $\tau = 1$.

Let us now forget about the federated learning setting and think about DP-FedAvg in a centralized manner. Let $\zeta = 0$ and $\tau = 1$. It is clear that conceptually, DP-FedAvg is equivalent to DP-SGD with both inner and outer stepsizes, and $n$ becomes the size of the minibatch:

$$\mathbf{x}_{r+1} = \mathbf{x}_r - \eta_g \frac{1}{n} \sum_{i=1}^{n} \left[\text{clip}_c\left(\eta_l \nabla F_i(\mathbf{x}_r)\right) + \mathbf{z}_i\right] . \tag{C.91}$$

Suppose the size of the whole dataset is $N_{\text{total}}$. Then the target optimization problem becomes:

$$f(\mathbf{x}) := \frac{1}{N_{\text{total}}} \sum_{j=1}^{N_{\text{total}}} F_j(\mathbf{x}) . \tag{C.92}$$

with

$$\mathbb{E}_j[||\nabla F_j(\mathbf{x}) - \nabla f(\mathbf{x})||^2] \le \sigma^2 . \tag{C.93}$$

Essentially, the analysis in section C.6.2 recovers this special case. However, the notion of DP is no longer the same as before. In the centralized setting, DP-SGD considers the global DP guarantee. In other words, what we should protect is the averaged clipped gradients $\frac{1}{n} \sum_{i=1}^{n} \text{clip}_c\big(\eta_l \nabla F_i(\mathbf{x}_r)\big)$ rather than each individual update. Therefore, if running $R$ steps, in order to achieve $(\varepsilon, \delta)$-DP guarantee, the noise $\sigma_{\text{DP}}$ should satisfy (by treating $\frac{1}{n} \sum_{i=1}^{n} \mathbf{z}_i$ as a single noise):

$$\frac{\sigma_{\text{DP}}^2}{n} = \text{Var}(\frac{1}{n} \sum_{i=1}^{n} \mathbf{z}_i) = \frac{dc^2 \log(1/\delta) R}{N_{\text{total}}^2 \varepsilon_{\text{DP}}^2} . \tag{C.94}$$

From the formula of $\sigma_{\text{DP}}^2$, we obtain $\nu = \frac{dn \log(1/\delta)}{N_{\text{total}}^2 \varepsilon_{\text{DP}}^2}$. Following the same discussion as in Section C.6.3, suppose $\sigma$ is relatively small, then we obtain the following optimal privacy-utility trade-off and required number of iterations:

$$\min_{r \in [1,R]} \mathbb{E}[||\nabla f(\mathbf{x}_r)||^2] \le \mathcal{O}\bigg(\frac{F_0 L d \log(1/\delta)}{N_{\text{total}}^2 \varepsilon_{\text{DP}}^2}\bigg), \qquad R = \Theta\bigg(\frac{N_{\text{total}}^2 \varepsilon_{\text{DP}}^2}{L d \log(1/\delta)}\bigg) . \tag{C.95}$$

The obtained trade-off is the same as DP-SGD with a single stepsize. Introducing an inner stepsize to the clipping operator could potentially drive convergence towards any desired accuracy level. However, it is crucial to note that this may come at the cost of an increased total number of iterations, consequently amplifying the associated noise levels.

# D    PER-SAMPLE CLIPPING

We first present our more refined convergence result formally, then prove the results for the convergence.

## D.1    CONVERGENCE RESULT

**Theorem IV** (Per-sample clipping). *Suppose function $f_i$ satisfies Assumption 1 to 3, then if we run Algorithm 1 for $T := R \cdot \tau$ steps with $R$ communication rounds, $\tau$ local steps, clipping threshold $c$, and step size $\eta \leq \frac{1}{14L\tau}$ with $L := L_0 + \min(c, M)L_1$ and $M := \max_t ||\nabla f(\bar{\mathbf{x}}_t)||$, then it holds that:*

$$
\begin{aligned}
\min_{t \in [1,T]} \mathbb{E}||\nabla f(\bar{\mathbf{x}}_t)|| \leq \mathcal{O}\Bigg( & \frac{f(\mathbf{x}_0) - f^*}{\eta cT} + \sqrt{\frac{f(\mathbf{x}_0) - f^*}{\eta T}} \\
& + \sqrt{\frac{\eta L\sigma^2}{n}} + \sqrt{\frac{\eta L\zeta^2}{n}}\mathbb{1}_{c < \mathcal{O}(\sigma + \zeta_{\max})} \\
& + \min\left(L\eta\sqrt{\hat{\tau}\sigma^2 + \tau^2\zeta^2}, c\right) + \min\left(\sigma + \zeta, \frac{\sigma^2 + \zeta^2}{c}\right)\Bigg).
\end{aligned}
\tag{D.1}
$$

*where $\hat{\tau} := \tau$ if $c > \mathcal{O}(\sigma + \zeta_{\max})$ else $\tau^2$*

Theorem IV is a more refined convergence rate compared to Theorem I in the manuscript. In Theorem I, we observe that even if when $c \to \infty$, we cannot recover the unclipped convergence rate as demonstrated in Koloskova et al. (2020) due to the terms $\mathcal{O}\left(\sqrt{\eta L\zeta^2/n}\right)$ and $\mathcal{O}\left(L\eta\sqrt{\tau^2\sigma^2}\right)$. However, we can cancel the influence of $\mathcal{O}(\sqrt{\zeta^2/n})$ and show the influence of the noise as $\mathcal{O}(\sqrt{\tau\sigma^2})$ such that we can recover the convergence rate provided in Koloskova et al. (2020) when $c \to \infty$ based on a more refined analysis by discussing the rate when $c > \mathcal{O}(\sigma + \zeta_{\max})$.

Note in practice, we can set the stepsize to be $\eta \leq \mathcal{O}\left(\frac{1}{14(L_0 + cL_1)\tau}\right)$ to avoid knowing the maximum of the norm of the gradient $M$. Since this stepsize can be smaller than the theoretical $\eta$ defined in Theorem IV, the convergence rate is also guaranteed.

We next give an updated Corollary which takes into account the differential private noise. After that, we give the proof for Theorem IV in the following sections.

**Corollary IV.** *Suppose function $f_i$ satisfies Assumption 1 to 3, then if we run Algorithm 1 with updating rule $\mathbf{y}_i \leftarrow \mathbf{y}_i - \eta(\mathbf{g}_i + \mathbf{z}_i)$ for $T := R \cdot \tau$ steps with $R$ communication rounds, $\tau$ local steps, clipping threshold $c$, stepsize $\eta \leq \frac{1}{14L\tau}$ with $L := L_0 + ML_1$ and $M := \max_t ||\nabla f(\bar{\mathbf{x}}_t)||$, and $\sigma_{DP}^2$ as the variance of the added Gaussian noise such that $\mathbf{z}_{i,t} \sim \mathcal{N}\left(0, \frac{\sigma_{DP}^2}{d}\mathbf{I}_d\right)$, then it holds:*

$$
\begin{aligned}
\min_{t \in [1,T]} \mathbb{E}||\nabla f(\bar{\mathbf{x}}_t)|| \leq \mathcal{O}\Bigg( & \frac{f(\mathbf{x}_0) - f^*}{\eta cT} + \sqrt{\frac{f(\mathbf{x}_0) - f^*}{\eta T}} \\
& + \sqrt{\frac{\eta L\sigma^2}{n}} + \sqrt{\frac{\eta L\zeta^2}{n}}\mathbb{1}_{c < \mathcal{O}(\sigma + \zeta_{\max})} \\
& + \min\left(L\eta\sqrt{\hat{\tau}\sigma^2 + \tau^2\zeta^2}, c\right) + \min\left(\sigma + \zeta, \frac{\sigma^2 + \zeta^2}{c}\right) \\
& + \frac{\eta L\sigma_{DP}^2}{nc} + \sqrt{\frac{\eta L}{n}}\sigma_{DP}\Bigg).
\end{aligned}
\tag{D.2}
$$

*where $\hat{\tau} := \tau$ if $c > \mathcal{O}(\sigma + \zeta_{\max})$ else $\tau^2$*

Note the stepsize defined in the above Corollary has to depend on $M$ instead of $\min(c, M)$. It cannot avoided due to the conditions in Assumption 3 (see discussion in Appendix A.1.1), as the algorithm does not clip the update after adding the noise.

**Local SGD** To facilitate the proof, we describe the Local SGD algorithm following Stich (2019). There are $n$ nodes in total. At iteration $t$ in parallel on all the nodes $i \in [n]$, we have:

$$
\mathbf{g}_{i,t} = \text{clip}_c(\nabla F_i(\mathbf{x}_{i,t})) := \min\left(1, \frac{c}{||\nabla F_i(\mathbf{x}_{i,t})||}\right)\nabla F_i(\mathbf{x}_{i,t}).
\tag{D.3a}
$$

If $t + 1$ is a multiple of $\tau$

$$
\mathbf{x}_{i,t+1} = \frac{1}{n}\sum_{i=1}^{n}\mathbf{x}_{i,t} - \eta\mathbf{g}_{i,t} \quad \text{global averaging}.
\tag{D.3b}
$$

Otherwise:

$$\mathbf{x}_{i,t+1} = \mathbf{x}_{i,t} - \eta\mathbf{g}_{i,t} \quad \text{local step}. \tag{D.3c}$$

**Additional definitions** Following Stich (2019), we first define virtual sequence $\{\bar{\mathbf{x}}_t\}_{t\geq 0}$ as:

$$\bar{\mathbf{x}}_0 = \mathbf{x}_0, \quad \bar{\mathbf{x}}_t = \frac{1}{n}\sum_{i=1}^{n}\mathbf{x}_{i,t}. \tag{D.4}$$

We then define the consensus distance between $\bar{\mathbf{x}}_t$ and $\mathbf{x}_{i,t}$ as:

$$R_t := \frac{1}{n}\sum_{i=1}^{n}||\bar{\mathbf{x}}_t - \mathbf{x}_{i,t}||^2. \tag{D.5}$$

As we are discussing the convergence rate based on the norm of the gradient, to distinguish between different cases, given $c$ as the clipping threshold, we define:

$$\mathcal{J}_{c+} := \{t; ||\nabla f(\bar{\mathbf{x}}_t)|| \geq c\}, \mathcal{J}_{\frac{c}{2}+} := \{t; \frac{c}{2} \leq ||\nabla f(\bar{\mathbf{x}}_t)|| < c\}, \mathcal{J}_{\frac{c}{2}-} := \{t; ||\nabla f(\bar{\mathbf{x}}_t)|| < \frac{c}{2}\}. \tag{D.6}$$

Specifically, we also define the following two sets to distinguish between the norm of the gradient within two communication rounds given $t - 1 - k_t$ as the index of the last communication round ($k_t \leq \tau - 1$):

$$\mathcal{J}_{c+,t} := \{j; ||\nabla f(\bar{\mathbf{x}}_j)|| \geq c, j \in [t-1-k_t, t-1]\} \tag{D.7a}$$

$$\mathcal{J}_{\frac{c}{2}+,t} := \{j; c/2 \leq ||\nabla f(\bar{\mathbf{x}}_j)|| < c, j \in [t-1-k_t, t-1]\} \tag{D.7b}$$

$$\mathcal{J}_{\frac{c}{2}-,t} := \{j; ||\nabla f(\bar{\mathbf{x}}_j)|| < c/2, j \in [t-1-k_t, t-1]\}, \quad \mathcal{J}_{c-,t} := \mathcal{J}_{\frac{c}{2}+,t} \cup \mathcal{J}_{\frac{c}{2}-,t} \tag{D.7c}$$

## D.2 PROOF OF CONVERGENCE

**When** $c < \mathcal{O}(\sigma + \zeta_{\max})$

We give the proof for the convergence of per-sample clipping in Local SGD. We first bound the difference between the model update in Lemma 12. To give the convergence rate, we consider when the clipping threshold is large $c \geq \mathcal{O}(\sigma + \zeta)$ and when the clipping threshold is small $c \leq \mathcal{O}(\sigma + \zeta)$. When the clipping threshold is large, we present the convergence by discussing when the norm of the gradient is large in Lemma 13, is intermediate large in Lemma 14, and is small in Lemma 15.

**Lemma 12** (Difference). *For $\eta \leq \eta_{crit} = \frac{1}{14L\tau}$ with $L := L_0 + \min(c, M)L_1$, it holds:*

$$\mathbb{E}R_t \leq \min\left(\frac{c^2}{196L^2},\right.$$
$$\left.\frac{1}{190L^2\tau}\sum_{j\in\mathcal{J}_{c+,t}}c^2 + \frac{1}{95L^2\tau}\sum_{j\in\mathcal{J}_{c-,t}}||\nabla f(\bar{\mathbf{x}}_j)||^2 + \frac{588\eta^2\tau}{95}\sum_{j\in\mathcal{J}_{c-,t}}(\sigma^2+\zeta^2)\right). \tag{D.8}$$

*Proof.* Following Koloskova et al. (2020), we assume that there exists a $k_t \leq \tau - 1$ such that:

$$R_t := \frac{1}{n}\sum_{i=1}^{n}||\bar{\mathbf{x}}_t - \mathbf{x}_{i,t}||^2 = \frac{\eta^2}{n}\sum_{i=1}^{n}||\sum_{j=t-1-k}^{t-1}(\mathbf{g}_{i,j} - \bar{\mathbf{g}}_j)||^2 \leq \frac{\eta^2}{n}\sum_{i=1}^{n}||\sum_{j=t-1-k_t}^{t-1}\mathbf{g}_{i,j}||^2. \tag{D.9}$$

Due to the clipping operation, we know $||\mathbf{g}_{i,t}||^2 \leq c^2$. Therefore, given $\eta \leq \frac{1}{14L\tau}$, we can bound $R_t$ by:

$$\mathbb{E}R_t \leq \frac{\eta^2}{n}\sum_{i=1}^{n}||\sum_{j-1-k_t}^{t-1}\mathbf{g}_{i,t}||^2 \leq \eta^2\tau^2 c^2 \leq \frac{c^2}{196L^2}. \tag{D.10}$$

We then give a more general bound for $R_t$ given the definition of the set of indices $\mathcal{J}_{c+,t}$ and $\mathcal{J}_{c-,t}$ from Eq. D.7:

$$R_t \leq \frac{\tau\eta^2}{n}\sum_{i=1}^{n}\sum_{j\in\mathcal{J}_{c+,t}}||\mathbf{g}_{i,j}||^2 + \frac{\tau\eta^2}{n}\sum_{i=1}^{n}\sum_{j\in\mathcal{J}_{c-,t}}||\mathbf{g}_{i,j} - \nabla f(\bar{\mathbf{x}}_j) + \nabla f(\bar{\mathbf{x}}_j)||^2$$

$$\leq \tau\eta^2\sum_{j\in\mathcal{J}_{c+,t}}c^2 + \frac{2\tau\eta^2}{n}\sum_{i=1}^{n}\sum_{j\in\mathcal{J}_{c-,t}}\mathbb{E}||\mathbf{g}_{ij} - \nabla f(\bar{\mathbf{x}}_j)||^2 + 2\tau\eta^2\sum_{j\in\mathcal{J}_{c-,t}}||\nabla f(\bar{\mathbf{x}}_j)||^2 \tag{D.11}$$

$$\leq \tau\eta^2\sum_{j\in\mathcal{J}_{c+,t}}c^2 + \frac{2\tau\eta^2}{n}\sum_{i=1}^{n}\sum_{j\in\mathcal{J}_{c-,t}}\underbrace{\mathbb{E}||\nabla F_i(\mathbf{x}_{ij}) - \nabla f(\bar{\mathbf{x}}_j)||^2}_{\mathcal{A}_1} + 2\tau\eta^2\sum_{j\in\mathcal{J}_{c-,t}}||\nabla f(\bar{\mathbf{x}}_j)||^2,$$

The first inequality uses the assumption that $k_t \leq \tau$ and Cauchy-Schwartz inequality. The second inequality uses the condition that if $||\nabla f(\bar{\mathbf{x}}_j)|| > c$ for $j \in \mathcal{J}_{c+,t}$, then a tighter upper bound we can give for $||\mathbf{g}_{ij}||^2$ is to simply to use the fact that $||\mathbf{g}_{ij}|| \leq c$. However, when $||\nabla f(\bar{\mathbf{x}}_j)|| < c$, we can give a more precise bound using triangle inequality. The third inequality uses Lemma 3.

We next give the bound for $\mathcal{A}_1 := \mathbb{E}||\nabla F_i(\mathbf{x}_{ij}) - \nabla f(\bar{\mathbf{x}}_j)||^2$ using the $(L_0, L_1)-$smoothness assumption:

$$
\begin{aligned}
\mathcal{A}_1 &\leq 3\mathbb{E}||\nabla F_i(\mathbf{x}_{ij}) - \nabla f_i(\mathbf{x}_{ij})||^2 + 3\mathbb{E}||\nabla f_i(\mathbf{x}_{ij}) - \nabla f_i(\bar{\mathbf{x}}_j)||^2 + 3||\nabla f_i(\bar{\mathbf{x}}_j) - \nabla f(\bar{\mathbf{x}}_j)||^2 \\
&\leq 3\sigma^2 + 3(L_0 + L_1||\nabla f(\bar{\mathbf{x}}_j)||)^2||\mathbf{x}_{ij} - \bar{\mathbf{x}}_j||^2 + 3||\nabla f_i(\bar{\mathbf{x}}_j) - \nabla f(\bar{\mathbf{x}}_j)||^2
\end{aligned} \tag{D.12}
$$

Plug Eq. D.12 back to Eq. D.11, we have:

$$
\begin{aligned}
\mathbb{E}R_t &\leq \tau\eta^2 \sum_{j \in \mathcal{J}_{c+,t}} c^2 + \frac{2\tau\eta^2}{n} \sum_{i=1}^{n} \sum_{j \in \mathcal{J}_{c-,t}} (3\sigma^2 + 3(L_0 + L_1||\nabla f(\bar{\mathbf{x}}_j)||)^2||\mathbf{x}_{ij} - \bar{\mathbf{x}}_j||^2 + 3||\nabla f_i(\bar{\mathbf{x}}_j) - \nabla f(\bar{\mathbf{x}}_j)||^2) \\
&\quad + 2\tau\eta^2 \sum_{j \in \mathcal{J}_{c-,t}} ||\nabla f(\bar{\mathbf{x}}_j)||^2 \\
&\leq \tau\eta^2 \sum_{j \in \mathcal{J}_{c+,t}} c^2 + 6\tau\eta^2 \sum_{j \in \mathcal{J}_{c-,t}} (\sigma^2 + \zeta^2) + 6\tau\eta^2 \sum_{j \in \mathcal{J}_{c-,t}} (L_0 + ||\nabla f(\bar{\mathbf{x}}_j)||L_1)^2 R_j + 2\tau\eta^2 \sum_{j \in \mathcal{J}_{c-,t}} ||\nabla f(\bar{\mathbf{x}}_j)||^2 \\
&\leq \tau\eta^2 \sum_{j \in \mathcal{J}_{c+,t}} c^2 + 6\tau\eta^2 \sum_{j \in \mathcal{J}_{c-,t}} (\sigma^2 + \zeta^2) + 6\tau^2\eta^2 L^2 R_t + 2\tau\eta^2 \sum_{j \in \mathcal{J}_{c-,t}} ||\nabla f(\bar{\mathbf{x}}_j)||^2,
\end{aligned} \tag{D.13}
$$

In the last inequality, we use the definition that $L := L_0 + \min(c, M)L_1$ and $\sum_{j \in \mathcal{J}_{c-,t}} L_0 + ||\nabla f(\bar{\mathbf{x}}_j)||L_1 \leq \sum_{j \in \mathcal{J}_{c-,t}} L$. We also uses the fact that $\mathbb{E}R_j \leq \mathbb{E}R_t, \forall j \in [t - 1 - k_t, t - 1]$. With $\eta \leq \frac{1}{14L\tau}$, we have:

$$
\mathbb{E}R_t \leq \frac{1}{190L^2\tau} \sum_{j \in \mathcal{J}_{c+,t}} c^2 + \frac{1}{95L^2\tau} \sum_{j \in \mathcal{J}_{c-,t}} ||\nabla f(\bar{\mathbf{x}}_j)||^2 + \frac{588\tau\eta^2}{95} \sum_{j \in \mathcal{J}_{c-,t}} (\sigma^2 + \zeta^2). \tag{D.14}
$$

$\square$

**When $c > \mathcal{O}(\sigma + \zeta)$**

**Lemma 13** (descent lemma for $||\nabla f(\bar{\mathbf{x}}_t)|| \geq c$). *Under Assumption 1, 2, and 3, and $||\nabla f(\bar{\mathbf{x}}_t)|| \geq c$ with stepsize $\eta \leq \frac{1}{14(L_0 + L_1 M)\tau}$, we have:*

$$
\frac{53c||\nabla f(\bar{\mathbf{x}}_t)||}{128} \leq \frac{f(\bar{\mathbf{x}}_t) - f(\bar{\mathbf{x}}_{t+1})}{\eta}. \tag{D.15}
$$

*Proof.* We start by using the smoothness of function $f$ (Lemma 4) and taking the conditional expectation:

$$
\mathbb{E}[f(\bar{\mathbf{x}}_{t+1})] \leq f(\bar{\mathbf{x}}_t) + \langle \nabla f(\bar{\mathbf{x}}_t), \mathbb{E}[\Delta\mathbf{x}] \rangle + \frac{L_0 + L_1||\nabla f(\bar{\mathbf{x}}_t)||}{2} \mathbb{E}||\Delta\mathbf{x}||^2. \tag{D.16}
$$

We first look at the second term. Using Lemma 1 and letting $\alpha = \frac{c}{||\nabla f(\bar{\mathbf{x}}_t)||}$, we have:

$$
\begin{aligned}
\langle \nabla f(\bar{\mathbf{x}}_t), \mathbb{E}[\Delta\mathbf{x}] \rangle &= -\frac{\eta}{n} \sum_{i=1}^{n} \mathbb{E}\langle \nabla f(\bar{\mathbf{x}}_t), \mathbf{g}_{i,t} \rangle \\
&= -\frac{\eta}{n} \sum_{i=1}^{n} \left( \frac{c}{2}||\nabla f(\bar{\mathbf{x}}_t)|| + \frac{1}{2\alpha}\mathbb{E}||\mathbf{g}_{i,t}||^2 - \frac{1}{2\alpha}\mathbb{E}||\mathbf{g}_{i,t} - \text{clip}_c(\nabla f(\bar{\mathbf{x}}_t))||^2 \right) \\
&\leq -\frac{\eta c}{2}||\nabla f(\bar{\mathbf{x}}_t)|| - \frac{\eta}{2\alpha n} \sum_{i=1}^{n} \mathbb{E}||\mathbf{g}_{i,t}||^2 \\
&\quad + \frac{\eta}{2\alpha} \frac{1}{n} \sum_{i=1}^{n} (3\sigma^2 + 3(L_0 + L_1||\nabla f(\bar{\mathbf{x}}_t)||)^2||\mathbf{x}_{i,t} - \bar{\mathbf{x}}_t||^2 + 3||\nabla f_i(\bar{\mathbf{x}}_t) - \nabla f(\bar{\mathbf{x}}_t)||^2))) \\
&\leq -\frac{\eta c}{2}||\nabla f(\bar{\mathbf{x}}_t)|| - \frac{\eta}{2\alpha n} \sum_{i=1}^{n} \mathbb{E}||\mathbf{g}_{i,t}||^2 + \frac{3\eta(\sigma^2 + \zeta^2)}{2\alpha} + \frac{3\eta(L_0 + L_1||\nabla f(\bar{\mathbf{x}}_t)||)^2}{2\alpha} R_t \\
&= -\frac{\eta c||\nabla f(\bar{\mathbf{x}}_t)||}{2}(1 - \frac{3\sigma^2}{c^2} - \frac{3\zeta^2}{c^2}) + \frac{3\eta(L_0 + L_1||\nabla f(\bar{\mathbf{x}}_t)||)^2}{2\alpha} R_t - \frac{\eta}{2\alpha n} \sum_{i=1}^{n} \mathbb{E}||\mathbf{g}_{i,t}||^2
\end{aligned}
$$

$$\leq -\frac{7\eta c||\nabla f(\bar{\mathbf{x}}_t)||}{16} + \frac{3\eta(L_0 + L_1||\nabla f(\bar{\mathbf{x}}_t)||)^2}{2\alpha}R_t - \frac{\eta}{2\alpha n}\sum_{i=1}^{n}\mathbb{E}||\mathbf{g}_{i,t}||^2$$

$$\leq -\frac{7\eta c||\nabla f(\bar{\mathbf{x}}_t)||}{16} + \frac{3\eta(L_0 + L_1||\nabla f(\bar{\mathbf{x}}_t)||)^2}{2\alpha}\left(\frac{1}{196(L_0 + ML_1)^2}c^2\right) - \frac{\eta}{2\alpha n}\sum_{i=1}^{n}\mathbb{E}||\mathbf{g}_{i,t}||^2$$

$$\leq -\frac{7\eta c||\nabla f(\bar{\mathbf{x}}_t)||}{16} + \frac{3\eta c||\nabla f(\bar{\mathbf{x}}_t)||}{392} - \frac{\eta}{2\alpha n}\sum_{i=1}^{n}\mathbb{E}||\mathbf{g}_{i,t}||^2$$

$$\leq -\frac{337}{784}\eta c||\nabla f(\bar{\mathbf{x}}_t)|| - \frac{\eta}{2\alpha n}\sum_{i=1}^{n}\mathbb{E}||\mathbf{g}_{i,t}||^2\,.$$

The first equality uses the update rule $\mathbb{E}[\Delta\mathbf{x}] := -\eta\mathbb{E}\mathbf{g}_{i,t}$. The second equality uses Lemma 1 with $\alpha = \frac{c}{||\nabla f(\bar{\mathbf{x}}_t)||}$. The first inequality uses the projection Lemma 3 and follows the same procedure as in bounding Eq. D.12 given Assumption 1, and 3. The second inequality uses Assumption 2 and the definition of the consensus distance $R_t := \frac{1}{n}\sum_{i=1}^{n}||\mathbf{x}_{i,t} - \bar{\mathbf{x}}_t||^2$. The third inequality uses the assumption that $\frac{3\sigma^2 + 3\zeta^2}{c^2} \leq \frac{1}{8}$. The fourth inequality uses Lemma 12, $(L_0 + L_1||\nabla f(\bar{\mathbf{x}}_t)||)^2 R_t \leq \min(\frac{c^2}{196}, \frac{1}{190\tau}\sum_{j\in\mathcal{J}_{c+,t}}c^2 + \frac{1}{95\tau}\sum_{j\in\mathcal{J}_{c-,t}}||\nabla f(\bar{\mathbf{x}}_j)||^2 + \frac{3}{95\tau}\sum_{j\in\mathcal{J}_{c-,t}}(\sigma^2 + \zeta^2))$. The smallest value in the second term will be $\frac{1}{95\tau}\sum_{j=t-1-k_t}^{t-1}||\nabla f(\bar{\mathbf{x}}_j)||^2 + \frac{3}{95\cdot 24}c^2$. For this term to be smaller than $\frac{c^2}{196}$, we would require $||\nabla f(\bar{\mathbf{x}}_j)||^2 \leq \frac{9}{25}c^2, \forall j \in [t-1-t_k, t-1]$. However, the gradient norm for the current time step $t$ is larger than $c$, $||\nabla f(\bar{\mathbf{x}}_t)|| > c$, which makes the requirement of $||\nabla f(\bar{\mathbf{x}}_j)||^2 \leq \frac{9}{25}c^2, \forall j \in [t-1-t_k, t-1]$ difficult. Therefore, we here bound $(L_0 + L_1||\nabla f(\bar{\mathbf{x}}_t)||)^2 R_t \leq \frac{c^2}{196}$.

Take the above equation back to Eq. D.16 and with $\eta \leq \frac{1}{14(L_0 + ML_1)\tau}$, we have:

$$f(\bar{\mathbf{x}}_{t+1}) \leq f(\bar{\mathbf{x}}_t) - \frac{53\eta c||\nabla f(\bar{\mathbf{x}}_t)||}{128} - \frac{\eta||\nabla f(\bar{\mathbf{x}}_t)||}{2cn}\sum_{i=1}^{n}\mathbb{E}||\mathbf{g}_{i,t}||^2 + \frac{\eta^2(L_0 + L_1||\nabla f(\bar{\mathbf{x}}_t)||)}{2n}\sum_{i=1}^{n}\mathbb{E}||\mathbf{g}_{i,t}||^2$$

$$\leq f(\bar{\mathbf{x}}_t) - \frac{53\eta c||\nabla f(\bar{\mathbf{x}}_t)||}{128} - \frac{\eta||\nabla f(\bar{\mathbf{x}}_t)||}{2cn}\sum_{i=1}^{n}\mathbb{E}||\mathbf{g}_{i,t}||^2(1 - \eta||\nabla f(\bar{\mathbf{x}}_t)||L_1) + \frac{\eta^2 L_0}{2n}\sum_{i=1}^{n}\mathbb{E}||\mathbf{g}_{i,t}||^2$$

$$\leq f(\bar{\mathbf{x}}_t) - \frac{53\eta c||\nabla f(\bar{\mathbf{x}}_t)||}{128} - \frac{\eta}{2n}\sum_{i=1}^{n}\mathbb{E}||\mathbf{g}_{i,t}||^2(1 - \eta||\nabla f(\bar{\mathbf{x}}_t)||L_1 - \eta L_0) \leq f(\bar{\mathbf{x}}_t) - \frac{53\eta c||\nabla f(\bar{\mathbf{x}}_t)||}{128}\,.$$

$$\text{(D.17)}$$

The third inequality uses the assumption that $||\nabla f(\bar{\mathbf{x}}_t)|| > c$. The last inequality uses the condition that $\eta \leq \frac{1}{14(L_0 + ML_1)\tau}$ given $\tau \geq 1$. $\qquad\square$

**Lemma 14.** (descent lemma for $\frac{c}{2} \leq ||\nabla f(\bar{\mathbf{x}}_t)|| < c$) Under Assumption 1, 2, 3, and condition $\frac{c}{2} \leq ||\nabla f(\bar{\mathbf{x}}_t)|| < c$ with stepsize $\eta \leq \frac{1}{14L\tau}$ given $L := L_0 + \min(c, M)L_1$, we have:

$$\frac{7c||\nabla f(\bar{\mathbf{x}}_t)||}{32} \leq \frac{f(\bar{\mathbf{x}}_t) - f(\bar{\mathbf{x}}_{t+1})}{\eta} + \frac{3L^2}{2}R_t\,. \qquad\qquad \text{(D.18)}$$

*Proof.* The proof is very similar to the previous case, we again first look at the term $\langle\nabla f(\bar{\mathbf{x}}_t), \mathbb{E}[\Delta\mathbf{x}]\rangle$:

$$\langle\nabla f(\bar{\mathbf{x}}_t), \mathbb{E}[\Delta\mathbf{x}]\rangle = -\frac{\eta}{n}\sum_{i=1}^{n}\mathbb{E}\langle\nabla f(\bar{\mathbf{x}}_t), \mathbf{g}_{i,t}\rangle = -\frac{\eta}{2}||\nabla f(\bar{\mathbf{x}}_t)||^2 - \frac{\eta}{2n}\sum_{i=1}^{n}\mathbb{E}||\mathbf{g}_{i,t}||^2 + \frac{\eta}{2n}\sum_{i=1}^{n}\mathbb{E}||\mathbf{g}_{i,t} - \nabla f(\bar{\mathbf{x}}_t)||^2$$

$$\leq -\frac{\eta}{2}||\nabla f(\bar{\mathbf{x}}_t)||^2 - \frac{\eta}{2n}\sum_{i=1}^{n}\mathbb{E}||\mathbf{g}_{i,t}||^2$$

$$+ \frac{\eta}{2n}\sum_{i=1}^{n}(3\sigma^2 + 3(L_0 + ||\nabla f(\bar{\mathbf{x}}_t)||L_1)^2||\mathbf{x}_{i,t} - \bar{\mathbf{x}}_t||^2 + 3||\nabla f_i(\bar{\mathbf{x}}_t) - \nabla f(\bar{\mathbf{x}}_t)||^2)$$

$$\leq -\frac{\eta}{2}||\nabla f(\bar{\mathbf{x}}_t)||^2 - \frac{\eta}{2n}\sum_{i=1}^{n}\mathbb{E}||\mathbf{g}_{i,t}||^2$$

$$+ \frac{3\eta\sigma^2 + 3\eta\zeta^2}{2} + \frac{3\eta(L_0 + ||\nabla f(\bar{\mathbf{x}}_t)||L_1)^2}{2n}\sum_{i=1}^{n}||\mathbf{x}_{i,t} - \bar{\mathbf{x}}_t||^2$$

$$\leq -\frac{\eta}{2}||\nabla f(\bar{\mathbf{x}}_t)||^2 - \frac{\eta}{2n}\sum_{i=1}^{n}\mathbb{E}||\mathbf{g}_{i,t}||^2 + \frac{3\eta\sigma^2 + 3\eta\zeta^2}{2} + \frac{3(L_0 + ||\nabla f(\bar{\mathbf{x}}_t)||L_1)^2\eta}{2}R_t$$

$$\leq -\frac{\eta c}{4}||\nabla f(\bar{\mathbf{x}}_t)|| + \frac{3\eta\sigma^2||\nabla f(\bar{\mathbf{x}}_t)||}{c} + \frac{3\eta\zeta^2||\nabla f(\bar{\mathbf{x}}_t)||}{c} + \frac{3(L_0 + ||\nabla f(\bar{\mathbf{x}}_t)||L_1)^2\eta}{2}R_t - \frac{\eta}{2n}\sum_{i=1}^{n}\mathbb{E}||\mathbf{g}_{i,t}||^2$$

$$= -\frac{\eta c||\nabla f(\bar{\mathbf{x}}_t)||}{4}(1 - \frac{3\sigma^2 + 3\zeta^2}{c^2}) + \frac{3(L_0 + ||\nabla f(\bar{\mathbf{x}}_t)||L_1)^2\eta}{2}R_t - \frac{\eta}{2n}\sum_{i=1}^{n}\mathbb{E}||\mathbf{g}_{i,t}||^2$$

$$\leq -\frac{7\eta c||\nabla f(\bar{\mathbf{x}}_t)||}{32} + \frac{3(L_0 + ||\nabla f(\bar{\mathbf{x}}_t)||L_1)^2\eta}{2}R_t - \frac{\eta}{2n}\sum_{i=1}^{n}\mathbb{E}||\mathbf{g}_{i,t}||^2.$$

The second equality uses Lemma 1 with $\alpha = 1$. In the first inequality, we consider if $||\nabla f(\bar{\mathbf{x}}_t)|| < c$, $\text{clip}_c(\nabla f(\bar{\mathbf{x}}_t)) = \nabla f(\bar{\mathbf{x}}_t)$, so then $||\mathbf{g}_{i,t} - \text{clip}_c(\nabla f(\bar{\mathbf{x}}_t))||$ can be bounded using Lemma 3 and Eq. D.12. The fourth inequality uses the condition that if $\frac{c}{2} \leq ||\nabla f(\bar{\mathbf{x}}_t)|| \leq c$, $-||\nabla f(\bar{\mathbf{x}}_t)|| \leq -\frac{c}{2}$ and $\frac{2||\nabla f(\bar{\mathbf{x}}_t)||}{c} \geq 1$. The fifth inequality uses the assumption that $\frac{3\sigma^2 + 3\zeta^2}{c^2} \leq \frac{1}{8}$. Take the above equation back in Eq. D.16 and given the condition $\eta \leq \frac{1}{14L\tau}$, we can obtain:

$$f(\bar{\mathbf{x}}_{t+1}) \leq f(\bar{\mathbf{x}}_t) - \frac{7\eta c||\nabla f(\bar{\mathbf{x}}_t)||}{32} + \frac{3(L_0 + ||\nabla f(\bar{\mathbf{x}}_t)||L_1)^2}{2}R_t - \frac{\eta}{2n}\sum_{i=1}^{n}\mathbb{E}||\mathbf{g}_{i,t}||^2 + \frac{\eta^2(L_0 + L_1||\nabla f(\bar{\mathbf{x}}_t)||)}{2}\frac{1}{n}\sum_{i=1}^{n}\mathbb{E}||\mathbf{g}_{i,t}||^2$$

$$\leq f(\bar{\mathbf{x}}_t) - \frac{7\eta c||\nabla f(\bar{\mathbf{x}}_t)||}{32} + \frac{3(L_0 + ||\nabla f(\bar{\mathbf{x}}_t)||L_1)^2}{2}R_t - \frac{\eta}{2n}\sum_{i=1}^{n}\mathbb{E}||\mathbf{g}_{i,t}||^2(1 - \eta L_0 - \eta||\nabla f(\bar{\mathbf{x}}_t)||L_1)$$

$$\leq f(\bar{\mathbf{x}}_t) - \frac{7\eta c||\nabla f(\bar{\mathbf{x}}_t)||}{32} + \frac{3(L_0 + ||\nabla f(\bar{\mathbf{x}}_t)||L_1)^2}{2}R_t$$

$$\leq f(\bar{\mathbf{x}}_t) - \frac{7\eta c||\nabla f(\bar{\mathbf{x}}_t)||}{32} + \frac{3L^2}{2}R_t.$$

$$(D.19)$$

$\square$

**Lemma 15** (descent lemma for $||\nabla f(\bar{\mathbf{x}}_t)|| < \frac{c}{2}$). *Under Assumption 1, 2, 3, and the condition $||\nabla f(\bar{\mathbf{x}}_t)|| < \frac{c}{2}$ with stepsize $\eta \leq \frac{1}{14L\tau}$ given $L := L_0 + \min(c, M)L_1$, we have:*

$$\frac{1}{4}||\nabla f(\bar{\mathbf{x}}_t)||^2 \leq \frac{f(\bar{\mathbf{x}}_t) - \mathbb{E}f(\bar{\mathbf{x}}_{t+1})}{\eta} + \frac{7}{2}L^2 R_t + \frac{16\eta L}{n}(\sigma^2 + \zeta^2) + \frac{72(\sigma^2 + \zeta^2)^2}{c^2}.$$

$$(D.20)$$

*Proof.* In this case, we cannot prove it like in the previous cases. We start by defining an indicator function $\delta_{i,t} := \mathbb{1}\{||\nabla F_i(\mathbf{x}_{it})|| > c\}$, which indicates that the stochastic gradient $\nabla F_i(\mathbf{x}_{it})$ at time step $t$ is clipped. We first show the bound for $\mathbb{E}[\delta_{i,t}] = \Pr(\delta_{i,t} = 1)$ as $\delta_{i,t}$ equals to 1 or 0.

$$\mathbb{E}[\delta_{i,t}] = \Pr[\delta_{i,t} = 1] = \Pr[||\nabla F_i(\mathbf{x}_{it})|| > c] \leq \Pr[||\nabla F_i(\mathbf{x}_{it}) - \nabla f(\bar{\mathbf{x}}_t)|| > \frac{c}{2}]. \quad (D.21)$$

We uses the condition that $||\nabla f(\bar{\mathbf{x}}_t)|| < \frac{c}{2}$ and triangle inequality. We then square both sides and obtain:

$$\mathbb{E}[\delta_{i,t}] \leq \Pr[||\nabla F_i(\mathbf{x}_{i,t}) - \nabla f(\bar{\mathbf{x}}_t)||^2 > \frac{c^2}{4}]$$

$$\leq \frac{4\mathbb{E}||\nabla F_i(\mathbf{x}_{i,t}) - \nabla f(\bar{\mathbf{x}}_t)||^2}{c^2}$$

$$\leq \frac{4(3\mathbb{E}||\nabla F_i(\mathbf{x}_{i,t}) - \nabla f_i(\mathbf{x}_{i,t})||^2 + 3||\nabla f_i(\mathbf{x}_{i,t}) - \nabla f_i(\bar{\mathbf{x}}_t)||^2 + 3||\nabla f_i(\bar{\mathbf{x}}_t) - \nabla f(\bar{\mathbf{x}}_t)||^2)}{c^2}$$

$$\leq \frac{12\sigma^2 + 12||\nabla f_i(\bar{\mathbf{x}}_t) - \nabla f(\bar{\mathbf{x}}_t)||^2}{c^2} + \frac{12(L_0 + ||\nabla f(\bar{\mathbf{x}}_t)||L_1)^2||\mathbf{x}_{i,t} - \bar{\mathbf{x}}_t||^2}{c^2}.$$

The second line uses Markov inequality. The third line uses Jensen inequality. The fourth line uses Assumption 1 and 3. As we have $n$ workers, we next bound the term $\frac{1}{n}\sum_{i=1}^{n}\mathbb{E}[\delta_{i,t}]$.

$$\frac{1}{n}\sum_{i=1}^{n}\mathbb{E}[\delta_{i,t}] \leq \frac{12(\sigma^2 + \zeta^2)}{c^2} + \frac{12(L_0 + ||\nabla f(\bar{\mathbf{x}}_t)||L_1)^2 R_t}{c^2}. \quad (D.22)$$

We next bound the differences between $||\nabla f(\bar{\mathbf{x}}_t) - \mathbb{E}\mathbf{g}_{i,t}||^2$. To approach the bound of $||\nabla f(\bar{\mathbf{x}}_t) - \mathbb{E}\mathbf{g}_{i,t}||^2$, we first find the expression for $\mathbb{E}\mathbf{g}_{i,t}$.

$$\mathbb{E}\mathbf{g}_{i,t} = \mathbb{E}\delta_{i,t}\frac{c}{||\nabla F_i(\mathbf{x}_{i,t})||}\nabla F_i(\mathbf{x}_{i,t}) + \mathbb{E}(1 - \delta_{i,t})\nabla F_i(\mathbf{x}_{i,t})$$

$$= \mathbb{E}\delta_{i,t}\left(\frac{c}{||\nabla F_i(\mathbf{x}_{i,t})||} - 1\right)\nabla F_i(\mathbf{x}_{i,t}) + \nabla f_i(\mathbf{x}_{i,t}),$$

Therefore:

$$||\nabla f(\bar{\mathbf{x}}_t) - \frac{1}{n}\sum_{i=1}^{n}\mathbb{E}\mathbf{g}_{i,t}||^2 = ||\nabla f(\bar{\mathbf{x}}_t) + \frac{1}{n}\sum_{i=1}^{n}\mathbb{E}\delta_{i,t}\left(1 - \frac{c}{||\nabla F_i(\mathbf{x}_{i,t})||}\right)\nabla F_i(\mathbf{x}_{i,t}) - \frac{1}{n}\sum_{i=1}^{n}\nabla f_i(\mathbf{x}_{i,t})||^2$$

$$\leq \frac{2}{n}\sum_{i=1}^{n}||\mathbb{E}\delta_{i,t}\left(1 - \frac{c}{||\nabla F_i(\mathbf{x}_{i,t})||}\right)\nabla F_i(\mathbf{x}_{i,t})||^2 + 2||\frac{1}{n}\sum_{i=1}^{n}\nabla f_i(\mathbf{x}_{i,t}) - \nabla f_i(\bar{\mathbf{x}}_t)||^2$$

$$\leq \frac{2}{n}\sum_{i=1}^{n}(\Pr(\delta_{i,t} = 1))^2\mathbb{E}||\left(1 - \frac{c}{||\nabla F_i(\mathbf{x}_{i,t})||}\right)\nabla F_i(\mathbf{x}_{i,t})|\delta_{i,t} = 1||^2$$

$$+ \frac{2}{n}\sum_{i=1}^{n}||\nabla f_i(\mathbf{x}_{i,t}) - \nabla f_i(\bar{\mathbf{x}}_t)||^2$$

$$\leq \frac{2}{n}\sum_{i=1}^{n}\Pr(\delta_{i,t} = 1)\delta_{i,t}\mathbb{E}||\nabla F_i(\mathbf{x}_{i,t})||^2 + 2(L_0 + ||\nabla f(\bar{\mathbf{x}}_t)||L_1)^2\frac{1}{n}\sum_{i=1}^{n}||\bar{\mathbf{x}}_t - \mathbf{x}_{i,t}||^2$$

$$\leq \frac{2}{n}\Pr(\delta_{i,t} = 1)\mathbb{E}||\nabla F_i(\mathbf{x}_{i,t})||^2 + 2(L_0 + ||\nabla f(\bar{\mathbf{x}}_t)||L_1)^2 R_t$$

$$\leq \frac{2}{n}\sum_{i=1}^{n}\Pr(\delta_{i,t} = 1)(6\sigma^2 + 6(L_0 + ||\nabla f(\bar{\mathbf{x}}_t)||L_1)^2||\bar{\mathbf{x}}_t - \mathbf{x}_{i,t}||^2$$

$$+ 6||\nabla f_i(\bar{\mathbf{x}}_t) - \nabla f(\bar{\mathbf{x}}_t)||^2 + 2||\nabla f(\bar{\mathbf{x}}_t)||^2) + 2(L_0 + ||\nabla f(\bar{\mathbf{x}}_t)||L_1)^2 R_t$$

$$\leq \frac{12}{n}\sum_{i=1}^{n}\Pr(\delta_{i,t} = 1)(\sigma^2 + \zeta^2 + (L_0 + ||\nabla f(\bar{\mathbf{x}}_t)||L_1)^2 R_t)$$

$$+ 4\Pr(\delta_{i,t} = 1)||\nabla f(\bar{\mathbf{x}}_t)||^2 + 2(L_0 + ||\nabla f(\bar{\mathbf{x}}_t)||L_1)^2 R_t,$$

The first inequality uses triangle inequality. The second inequality uses the rule that $\mathbb{E}||\delta X|| = \Pr(\delta = 1)\mathbb{E}[X|\delta = 1], ||\mathbb{E}[\delta X]||^2 = (\Pr(\delta = 1))^2||\mathbb{E}[X|\delta = 1]||^2$. The third inequality uses the condition that $||\nabla F_i(\mathbf{x}_{i,t})|| > c$ when $\delta_{i,t} = 1$, and Jensen inequality for conditional expectation. The fourth inequality uses the fact that $\delta_{i,t}$ is either 1 or 0. We next approach each term individually.

$$\frac{12}{n}\sum_{i=1}^{n}\Pr(\delta_{i,t} = 1)(\sigma^2 + \zeta^2) \leq \frac{144(\sigma^2 + \zeta^2)^2}{c^2} + \frac{144L^2 R_t(\sigma^2 + \zeta^2)}{c^2}$$

$$\leq \frac{144(\sigma^2 + \zeta^2)^2}{c^2} + \frac{3L^2 R_t}{4},$$

$$\frac{12}{n}\sum_{i=1}^{n}\Pr(\delta_{i,t} = 1)(L_0 + cL_1)^2 R_t \leq (\frac{3}{4} + \frac{144L^2 R_t}{c^2})L^2 R_t$$

$$\leq (\frac{3}{4} + \frac{144}{196})L^2 R_t,$$

For both terms, we use the definition $L := L_0 + \min(c, M)L_1$ and the condition $L_0 + ||\nabla f(\bar{\mathbf{x}}_t)||L_1 \leq L$. We also assume that $\frac{12(\sigma^2 + \zeta^2)}{c^2} \leq \frac{1}{16}$. For the second equation, we know that $R_t$ is bounded by the minimum between Eq. D.10 and Eq. D.14. We here choose to use the slightly larger bound from Eq. D.10 for $R_t$ to continue the proof. Now combining the results, we have:

$$||\nabla f(\bar{\mathbf{x}}_t) - \frac{1}{n}\sum_{i=1}^{n}\mathbb{E}\mathbf{g}_{i,t}||^2 \leq \frac{144(\sigma^2 + \zeta^2)^2}{c^2} + (\frac{7}{2} + \frac{144}{196})L^2 R_t + (\frac{1}{4} + \frac{12}{49})||\nabla f(\bar{\mathbf{x}}_t)||^2. \quad \text{(D.23)}$$

$$-\frac{1}{n}\sum_{i=1}^{n}\langle\nabla f(\bar{\mathbf{x}}_t), \mathbb{E}[\mathbf{g}_{i,t}]\rangle = -\frac{1}{2}||\nabla f(\bar{\mathbf{x}}_t)||^2 - \frac{1}{2}||\frac{1}{n}\sum_{i=1}^{n}\mathbb{E}\mathbf{g}_{i,t}||^2 + \frac{1}{2}||\frac{1}{n}\sum_{i=1}^{n}\nabla f(\bar{\mathbf{x}}_t) - \mathbb{E}\mathbf{g}_{i,t}||^2$$

$$\leq -\frac{1}{4}||\nabla f(\bar{\mathbf{x}}_t)||^2 + (\frac{7}{4} + \frac{72}{196})L^2 R_t + \frac{72(\sigma^2 + \zeta^2)^2}{c^2} - \frac{1}{2}||\frac{1}{n}\sum_{i=1}^{n}\mathbb{E}\mathbf{g}_{i,t}||^2.$$

We next approach the term $\mathbb{E}||\frac{1}{n}\sum_{i=1}^{n}\mathbf{g}_{i,t}||^2$

$$
\begin{aligned}
\frac{1}{2}\mathbb{E}||\frac{1}{n}\sum_{i=1}^{n}\mathbf{g}_{i,t}||^2 &= \frac{1}{2}\mathbb{E}||\frac{1}{n}\sum_{i=1}^{n}\mathbf{g}_{i,t} - \mathbb{E}\mathbf{g}_{i,t}||^2 + \frac{1}{2}||\frac{1}{n}\sum_{i=1}^{n}\mathbb{E}\mathbf{g}_{i,t}||^2 = \frac{1}{2n^2}\sum_{i=1}^{n}\mathbb{E}||\mathbf{g}_{i,t} - \mathbb{E}\mathbf{g}_{it}||^2 + \frac{1}{2}||\frac{1}{n}\sum_{i=1}^{n}\mathbb{E}\mathbf{g}_{i,t}||^2 \\
&\le \frac{1}{n^2}\sum_{i=1}^{n}\mathbb{E}||\mathbf{g}_{i,t} - \nabla f(\bar{\mathbf{x}}_t)||^2 + \frac{1}{n^2}\sum_{i=1}^{n}||\nabla f(\bar{\mathbf{x}}_t) - \mathbb{E}\mathbf{g}_{i,t}||^2 + \frac{1}{2}||\frac{1}{n}\sum_{i=1}^{n}\mathbb{E}\mathbf{g}_{i,t}||^2 \\
&\le \frac{1}{n^2}\sum_{i=1}^{n}\mathbb{E}||\mathbf{g}_{i,t} - \nabla f(\bar{\mathbf{x}}_t)||^2 + \frac{1}{2}||\frac{1}{n}\sum_{i=1}^{n}\mathbb{E}\mathbf{g}_{i,t}||^2 \\
&\quad + \frac{1}{n}\left(\frac{144(\sigma^2+\zeta^2)^2}{c^2} + (\frac{7}{2}+\frac{144}{196})L^2 R_t + \frac{48(\sigma^2+\zeta^2)+48L^2 R_t}{c^2}||\nabla f(\bar{\mathbf{x}}_t)||^2\right) \\
&\le \frac{1}{n}(3\sigma^2 + 3\zeta^2 + 3L^2 R_t) + \frac{1}{2}||\frac{1}{n}\sum_{i=1}^{n}\mathbb{E}\mathbf{g}_{i,t}||^2 \\
&\quad + \frac{1}{n}\left(\frac{3(\sigma^2+\zeta^2)}{4} + (\frac{7}{2}+\frac{144}{196})L^2 R_t + 12(\sigma^2+\zeta^2) + 12L^2 R_t\right) \\
&\le \frac{16(\sigma^2+\zeta^2)}{n} + (\frac{37}{2}+\frac{144}{196})L^2 R_t + \frac{1}{2}||\frac{1}{n}\sum_{i=1}^{n}\mathbb{E}\mathbf{g}_{i,t}||^2 .
\end{aligned}
$$
$$(D.24)$$

The first equality uses the fact that $\mathbb{E}\frac{1}{n}\sum_{i=1}^{n}\mathbf{g}_{i,t} - \mathbb{E}\mathbf{g}_{i,t} = 0$ and the variance of the sum of independent variables equal to the sum of the variance of the variables. The first inequality uses Jensen inequality. The second inequality uses the result from Eq. D.22 and Eq. D.23. The third inequality uses Jensen inequality and the assumption $||\nabla f(\bar{\mathbf{x}}_t)|| \le \frac{c}{2}$.

Therefore, using the smoothness assumption and taking the conditional expectation, we have:

$$
\begin{aligned}
\mathbb{E}f(\bar{\mathbf{x}}_{t+1}) &\le f(\bar{\mathbf{x}}_t) - \frac{\eta}{4}||\nabla f(\bar{\mathbf{x}}_t)||^2 + \frac{415\eta}{196}L^2 R_t + \frac{72\eta(\sigma^2+\zeta^2)^2}{c^2} - \frac{\eta}{2}||\frac{1}{n}\sum_{i=1}^{n}\mathbb{E}\mathbf{g}_{i,t}||^2 \\
&\quad + \frac{16\eta^2 L(\sigma^2+\zeta^2)}{n} + \frac{3770\eta^2 L^3}{196}R_t + \frac{\eta^2 L}{2}||\frac{1}{n}\sum_{i=1}^{n}\mathbb{E}\mathbf{g}_{i,t}||^2 \\
&\le f(\bar{\mathbf{x}}_t) - \frac{\eta}{4}||\nabla f(\bar{\mathbf{x}}_t)||^2 + (\frac{415}{196}+\frac{3770}{196\cdot 14})\eta L^2 R_t \\
&\quad + \frac{72\eta(\sigma^2+\zeta^2)^2}{c^2} + \frac{16\eta^2 L(\sigma^2+\zeta^2)}{n} .
\end{aligned}
$$
$$(D.25)$$

Rearrange the above equation and divide by $\eta$, we have:

$$
\frac{1}{4}||\nabla f(\bar{\mathbf{x}}_t)||^2 \le \frac{f(\bar{\mathbf{x}}_t) - f(\bar{\mathbf{x}}_{t+1})}{\eta} + \frac{7}{2}L^2 R_t + \frac{16\eta L}{n}(\sigma^2+\zeta^2) + \frac{72(\sigma^2+\zeta^2)^2}{c^2} .
$$
$$(D.26)$$

$\square$

**Wrapping up**

· When $L^2 R_t \le \frac{1}{190\tau}\sum_{j\in\mathcal{J}_{c+,t}}c^2 + \frac{1}{95\tau}\sum_{j\in\mathcal{J}_{c-,t}}||\nabla f(\bar{\mathbf{x}}_j)||^2 + \frac{588\tau\eta^2 L^2}{95}(\sigma^2+\zeta^2)$

Now, we wrap up the above three cases together. Given $\mathcal{A}_3 := \frac{1}{128T}\left(\sum_{t\in\mathcal{J}_{c+}}c||\nabla f(\bar{\mathbf{x}}_t)|| + \sum_{t\in\mathcal{J}_{\frac{c}{2}+}}c||\nabla f(\bar{\mathbf{x}}_t)|| + \sum_{t\in\mathcal{J}_{\frac{c}{2}-}}||\nabla f(\bar{\mathbf{x}}_t)||^2\right)$:

$$
\begin{aligned}
\mathcal{A}_3 &\le \frac{f(\mathbf{x}_0) - f^*}{28\eta T} + \sum_{t\in\mathcal{J}_{\frac{c}{2}+}}\frac{3L^2}{56}R_t + \sum_{t\in\mathcal{J}_{\frac{c}{2}-}}\frac{7L^2}{64}R_t \\
&\quad + \frac{\eta L}{2}(\sigma^2+\zeta^2) + \frac{9(\sigma^2+\zeta^2)^2}{4c^2} \\
&\le \frac{f(\mathbf{x}_0) - f^*}{28\eta T} + \frac{\eta L}{2}(\sigma^2+\zeta^2) + \frac{9(\sigma^2+\zeta^2)^2}{4c^2}
\end{aligned}
$$

$$+ \frac{1}{T}\sum_{t\in\mathcal{J}_{\frac{c}{2}+}} \frac{3}{56}\left(\frac{1}{190\tau}\sum_{j\in\mathcal{J}_{c+,t}} c||\nabla f(\bar{\mathbf{x}}_j)|| + \frac{1}{95\tau}\sum_{j\in\mathcal{J}_{\frac{c}{2}+,t}} c||\nabla f(\bar{\mathbf{x}}_j)|| + \frac{1}{95\tau}\sum_{j\in\mathcal{J}_{\frac{c}{2}-,t}} ||\nabla f(\bar{\mathbf{x}}_j)||^2\right)$$

$$+ \frac{1}{T}\sum_{t\in\mathcal{J}_{\frac{c}{2}-}} \frac{7}{64}\left(\frac{1}{190\tau}\sum_{j\in\mathcal{J}_{c+,t}} c||\nabla f(\bar{\mathbf{x}}_j)|| + \frac{1}{95\tau}\sum_{j\in\mathcal{J}_{\frac{c}{2}+,t}} c||\nabla f(\bar{\mathbf{x}}_j)|| + \frac{1}{95\tau}\sum_{j\in\mathcal{J}_{\frac{c}{2}-,t}} ||\nabla f(\bar{\mathbf{x}}_j)||^2\right)$$

$$+ \frac{1}{T}\sum_{t\in\mathcal{J}_{\frac{c}{2}+}} \frac{3}{56}\frac{588\tau\eta^2 L^2}{95}\sum_{j\in\mathcal{J}_{c-,t}}(\sigma^2+\zeta^2) + \sum_{t\in\mathcal{J}_{\frac{c}{2}-}}\frac{7}{64}\frac{588\tau\eta^2 L^2}{95}\sum_{j\in\mathcal{J}_{c-,t}}(\sigma^2+\zeta^2)$$

$$\leq \frac{f(\mathbf{x}_0)-f^*}{28\eta T} + \frac{\eta L}{2}(\sigma^2+\zeta^2) + \frac{9(\sigma^2+\zeta^2)^2}{4c^2}$$

$$+ \frac{1}{T}\sum_{t\in\mathcal{J}_{c+}}\frac{7}{64\cdot 190}c||\nabla f(\bar{\mathbf{x}}_t)|| + \frac{1}{T}\sum_{t\in\mathcal{J}_{\frac{c}{2}+}}\frac{7}{64\cdot 95}c||\nabla f(\bar{\mathbf{x}}_t)|| + \frac{1}{T}\sum_{t\in\mathcal{J}_{\frac{c}{2}-}}\frac{7}{64\cdot 95}||\nabla f(\bar{\mathbf{x}}_t)||^2$$

$$+ \frac{13}{19}\tau^2\eta^2 L^2(\sigma^2+\zeta^2).$$

In the last equality, each term at time step $t$ in the parentheses can maximum appear $k_t$ times and $k_t \leq \tau - 1$. As for the coefficient for each term, we can simply bound it with the largest coefficient from when $t \in \mathcal{J}_{\frac{c}{2}+}$ and $t \in \mathcal{J}_{\frac{c}{2}-}$. We can also verify this in an example later. Note the same terms appear both in the left-hand side and the right-hand side of the equation, and the convergence behaviour between when $t \in \mathcal{J}_{c+}$ and $t \in \mathcal{J}_{\frac{c}{2}+}$ is similar, so we can rearrange it:

$$\frac{1}{152T}\left(\sum_{t\in\mathcal{J}_{c+}\cup\mathcal{J}_{\frac{c}{2}+}} c||\nabla f(\bar{\mathbf{x}}_t)|| + \sum_{t\in\mathcal{J}_{\frac{c}{2}-}}||\nabla f(\bar{\mathbf{x}}_t)||^2\right) \leq \frac{f(\mathbf{x}_0)-f^*}{28\eta T} + \frac{\eta L}{2n}(\sigma^2+\zeta^2)$$

$$+ \frac{9(\sigma^2+\zeta^2)^2}{4c^2} + \frac{13}{19}\tau^2\eta^2 L^2(\sigma^2+\zeta^2). \tag{D.27}$$

We know that $x^2 \geq 2\varepsilon x - \varepsilon^2$ for any $\varepsilon, x > 0$. Following Koloskova et al. (2023), we can get $\frac{1}{T}\sum_{t\in\mathcal{J}_{\frac{c}{2}-}}(2\varepsilon\mathbb{E}||\nabla f(\bar{\mathbf{x}}_t)|| - \varepsilon^2) \leq A$. letting $A := \frac{f(\mathbf{x}_0)-f^*}{\eta} + \frac{(\sigma^2+\zeta^2)^2}{c^2} + \frac{\eta L(\sigma^2+\zeta^2)}{n} + L^2\eta^2\tau^2(\sigma^2+\zeta^2)$ without considering the coefficient, and $\varepsilon = \sqrt{A}$, we have:

$$\frac{1}{T}\sum_{t\in\mathcal{J}_{\frac{c}{2}-}}||\nabla f(\bar{\mathbf{x}}_t)|| \leq \sqrt{A} \leq \sqrt{\frac{f(\mathbf{x}_0)-f(\mathbf{x}^*)}{\eta T} + \frac{\sigma^2+\zeta^2}{c}}$$

$$+ \sqrt{\frac{\eta L(\sigma^2+\zeta^2)}{n}} + \sqrt{L^2\eta^2\tau^2(\sigma^2+\zeta^2)}.$$

Summing up the two cases together, we have:

$$\frac{1}{T}\sum_{t=0}^{T}||\nabla f(\bar{\mathbf{x}}_t)|| \leq \frac{f(\mathbf{x}_0)-f(\mathbf{x}^*)}{c\eta T} + \sqrt{\frac{f(\mathbf{x}_0)-f(\mathbf{x}^*)}{\eta T} + \frac{\sigma^2+\zeta^2}{c}}$$

$$+ \sqrt{\frac{\eta L(\sigma^2+\zeta^2)}{n}} + L\eta\tau\sqrt{(\sigma^2+\zeta^2)}. \tag{D.28}$$

· **When $L^2 R_t \leq \frac{c^2}{196}$**

If instead $L^2 R_t$ is bounded by $\frac{c^2}{196}, \forall t \in [T]$, then we would have a similar convergence rate as Eq. D.28 but instead of the term $L\eta\tau\sqrt{(\sigma^2+\zeta^2)}$, we would have a term that is $c$.

$$\frac{1}{T}\sum_{t=0}^{T}||\nabla f(\bar{\mathbf{x}}_t)|| \leq \frac{f(\mathbf{x}_0)-f(\mathbf{x}^*)}{c\eta T} + \sqrt{\frac{f(\mathbf{x}_0)-f(\mathbf{x}^*)}{\eta T} + \frac{\sigma^2+\zeta^2}{c}} + \sqrt{\frac{\eta L(\sigma^2+\zeta^2)}{n}} + c. \tag{D.29}$$

Therefore, combining the above two cases from Eq. D.28 and Eq. D.29 together, we have:

$$\frac{1}{T}\sum_{t=0}^{T}||\nabla f(\bar{\mathbf{x}}_t)|| \leq \frac{f(\mathbf{x}_0) - f(\mathbf{x}^*)}{c\eta T} + \sqrt{\frac{f(\mathbf{x}_0) - f(\mathbf{x}^*)}{\eta T}} + \frac{\sigma^2 + \zeta^2}{c}$$
$$+ \sqrt{\frac{\eta L(\sigma^2 + \zeta^2)}{n}} + \min\left(L\eta\tau\sqrt{\sigma^2 + \zeta^2}, c\right). \tag{D.30}$$

**When $c < \mathcal{O}(\sigma + \zeta)$**

In this part, we assume the clipping threshold is small $c < 12\sigma + 12\zeta$ and $20\sigma + 20\zeta \leq ||\nabla f(\bar{\mathbf{x}}_t)||$. Note, the numerical value is chosen high here to make the proof simple and clean.

**Lemma 16.** *(descent lemma for $c^2 \leq \mathcal{O}\left(\sigma^2 + \zeta^2\right)$) Under Assumption 1, 2, 3, and the condition that $c \leq 12\sigma + 12\zeta$, and $20\sigma + 20\zeta \leq ||\nabla f(\bar{\mathbf{x}}_t)||$, we have:*

$$\frac{c}{11}||\nabla f(\bar{\mathbf{x}}_t)|| \leq \frac{f(\mathbf{x}_0) - f^*}{\eta}. \tag{D.31}$$

*Proof.* We again start by using $(L_0, L_1)-$smoothness:

$$\mathbb{E}f(\bar{\mathbf{x}}_{t+1}) \leq f(\bar{\mathbf{x}}_t) - \frac{\eta}{n}\sum_{i=1}^{n}\mathbb{E}\langle\nabla f(\bar{\mathbf{x}}_t), \mathbf{g}_{i,t}\rangle + \frac{\eta^2(L_0 + L_1||\nabla f(\bar{\mathbf{x}}_t)||)}{2}\mathbb{E}||\mathbf{g}_{i,t}||^2$$

$$\leq f(\bar{\mathbf{x}}_t) - \frac{\eta}{n}\sum_{i=1}^{n}\mathbb{E}\langle\nabla f(\bar{\mathbf{x}}_t), \mathbf{g}_{i,t}\rangle + \frac{\eta^2(L_0 + L_1||\nabla f(\bar{\mathbf{x}}_t)||)c^2}{2} \tag{D.32}$$

$$\leq f(\bar{\mathbf{x}}_t) - \frac{\eta}{n}\sum_{i=1}^{n}\mathbb{E}\langle\nabla f(\bar{\mathbf{x}}_t), \mathbf{g}_{i,t}\rangle + \frac{3\eta^2(L_0 + L_1||\nabla f(\bar{\mathbf{x}}_t)||)c}{10}||\nabla f(\bar{\mathbf{x}}_t)||.$$

The second inequality uses the fact that after clipping, the norm of the stochastic gradients is smaller than $c$. the last inequality uses the assumption that $c < \frac{3}{5}||\nabla f(\bar{\mathbf{x}}_t)||$.

We then define an indicator function $\delta_{c-} := \mathbf{1}\{||\nabla F_i(\mathbf{x}_{i,t}) - \nabla f(\bar{\mathbf{x}}_t)|| > 12\sigma + 12||\nabla f_i(\bar{\mathbf{x}}_t) - \nabla f(\bar{\mathbf{x}}_t)|| + (L_0 + cL_1)||\mathbf{x}_{i,t} - \bar{\mathbf{x}}||\}$. We will derive the convergence rate based on $\delta_{c-}$ next.

**- When $\delta_{c-} = 0$**

If $\delta_{c-} = 0$, then $||\nabla F_i(\mathbf{x}_{i,t}) - \nabla f(\bar{\mathbf{x}}_t)|| \leq 12\sigma + 12||\nabla f_i(\bar{\mathbf{x}}_t) - \nabla f(\bar{\mathbf{x}}_t)|| + (L_0 + ||\nabla f(\bar{\mathbf{x}}_t)||L_1)||\mathbf{x}_{i,t} - \bar{\mathbf{x}}_t||$. This means that strong variance holds for some of the stochastic noises. Assuming $\alpha_i := \min\left(1, \frac{c}{||\nabla F_i(\mathbf{x}_{i,t})||}\right)$, we have:

$$-\frac{1}{n}\sum_{i=1}^{n}\langle\nabla f(\bar{\mathbf{x}}_t), \mathbf{g}_{i,t}\rangle = -\frac{1}{n}\sum_{i=1}^{n}\langle\nabla f(\bar{\mathbf{x}}_t), \alpha_i\nabla F_i(\mathbf{x}_{i,t}) - \alpha_i\nabla f(\bar{\mathbf{x}}_t) + \alpha_i\nabla f(\bar{\mathbf{x}}_t)\rangle$$

$$\leq -\frac{1}{n}\sum_{i=1}^{n}\alpha_i||\nabla f(\bar{\mathbf{x}}_t)||^2 + \frac{1}{n}\sum_{i=1}^{n}\alpha_i\langle\nabla f(\bar{\mathbf{x}}_t), \nabla f(\bar{\mathbf{x}}_t) - \nabla F(\mathbf{x}_{i,t})\rangle$$

$$\leq -\frac{1}{n}\sum_{i=1}^{n}\alpha_i||\nabla f(\bar{\mathbf{x}}_t)||^2 + \frac{1}{n}\sum_{i=1}^{n}\alpha_i||\nabla f(\bar{\mathbf{x}}_t)||||\nabla f(\bar{\mathbf{x}}_t) - \nabla F(\mathbf{x}_{i,t})||$$

$$\leq -\frac{1}{n}\sum_{i=1}^{n}\alpha_i||\nabla f(\bar{\mathbf{x}}_t)||^2$$

$$+ \frac{1}{n}\sum_{i=1}^{n}\alpha_i||\nabla f(\bar{\mathbf{x}}_t)||\left(12\sigma + 12||\nabla f_i(\bar{\mathbf{x}}_t) - \nabla f(\bar{\mathbf{x}}_t)|| + (L_0 + ||\nabla f(\bar{\mathbf{x}}_t)||L_1)||\mathbf{x}_{i,t} - \bar{\mathbf{x}}_t||\right)$$

$$\leq -\frac{1}{n}\sum_{i=1}^{n}\alpha_i||\nabla f(\bar{\mathbf{x}}_t)||^2$$

$$+ \left(\frac{1}{n}\sum_{i=1}^{n}\alpha_i||\nabla f(\bar{\mathbf{x}}_t)||\right)\left(\frac{1}{n}\sum_{i=1}^{n}12\sigma + 12||\nabla f_i(\bar{\mathbf{x}}_t) - \nabla f(\bar{\mathbf{x}}_t)|| + (L_0 + ||\nabla f(\bar{\mathbf{x}}_t)||L_1)||\mathbf{x}_{i,t} - \bar{\mathbf{x}}_t||\right)$$

$$\leq -\frac{1}{n}\sum_{i=1}^{n}\alpha_i||\nabla f(\bar{\mathbf{x}}_t)||^2 + \frac{1}{n}\sum_{i=1}^{n}\alpha_i||\nabla f(\bar{\mathbf{x}}_t)||(12\sigma + 12\zeta + (L_0 + ||\nabla f(\bar{\mathbf{x}}_t)||L_1)\eta\tau c)$$

$$\leq -\frac{1}{n}\sum_{i=1}^{n}\alpha_i||\nabla f(\bar{\mathbf{x}}_t)||^2 + \frac{1}{n}\sum_{i=1}^{n}\alpha_i||\nabla f(\bar{\mathbf{x}}_t)||\left(\frac{3}{5}||\nabla f(\bar{\mathbf{x}}_t)|| + \frac{3}{40}||\nabla f(\bar{\mathbf{x}}_t)||\right)$$

$$= -\frac{13}{40n}\sum_{i=1}^{n}\alpha_i||\nabla f(\bar{\mathbf{x}}_t)||^2 .$$

We use triangle inequality in the third line. and $12(\sigma + \zeta) \leq \frac{3}{5}||\nabla f(\bar{\mathbf{x}}_t)||, \eta \leq \frac{1}{8(L_0+\min(c,M)L_1)\tau}$ in the seventh inequality.

As $\alpha_i := \min\left(1, \frac{c}{||\nabla F_i(\mathbf{x}_{i,t})||}\right)$, we can bound $\frac{1}{n}\sum_{i=1}^{n}\alpha_i$:

$$\frac{1}{n}\sum_{i=1}^{n}\alpha_i \geq \frac{1}{n}\sum_{i=1}^{n}\frac{c}{||\nabla F_i(\mathbf{x}_{i,t})||} \geq \frac{c}{\frac{1}{n}\sum_{i=1}^{n}||\nabla F_i(\mathbf{x}_{i,t})||}$$

$$\geq \frac{c}{12\sigma + 12\zeta + \eta(L_0 + ||\nabla f(\bar{\mathbf{x}}_t)||L_1)\tau c + ||\nabla f(\bar{\mathbf{x}}_t)||}$$

$$\geq \frac{c}{\frac{8}{5}||\nabla f(\bar{\mathbf{x}}_t)|| + \frac{1}{8}\frac{3}{5}||\nabla f(\bar{\mathbf{x}}_t)||} \tag{D.33}$$

$$= \frac{40c}{67||\nabla f(\bar{\mathbf{x}}_t)||} .$$

The second inequality uses $||\nabla f_i(\bar{\mathbf{x}}_t)|| \leq 12\zeta + ||\nabla f(\bar{\mathbf{x}}_t)||$, $||\nabla f_i(\mathbf{x}_{i,t})|| \leq (L_0 + ||\nabla f(\bar{\mathbf{x}}_t)||L_1)\eta\tau c + ||\nabla f_i(\bar{\mathbf{x}}_t)||$, and $||\nabla F_i(\mathbf{x}_{i,t})|| \leq 12\sigma + ||\nabla f_i(\mathbf{x}_{i,t})||$. The third line uses the condition $\eta \leq \frac{1}{8(L_0+\min(c,M)L_1)\tau}, \sigma + \zeta \leq \frac{1}{20}||\nabla f(\bar{\mathbf{x}}_t)||$. Therefore, we can now bound the inner product:

$$-\frac{1}{n}\sum_{i=1}^{n}\langle\nabla f(\bar{\mathbf{x}}_t), \mathbf{g}_{i,t}\rangle \leq -\frac{13}{67}c||\nabla f(\bar{\mathbf{x}}_t)|| . \tag{D.34}$$

**- When $\delta_{c-} = 1$**

If $\delta_{c-} = 1$, then some stochastic noises can be larger than the assumed variance. Therefore, we cannot bound the inner product as before. Instead, we can use Cauchy-Schwarts inequality to bound:

$$-\frac{1}{n}\sum_{i=1}^{n}\langle\nabla f(\bar{\mathbf{x}}_t), \alpha_i\nabla F_i(\mathbf{x}_{i,t})\rangle \leq |\frac{1}{n}\sum_{i=1}^{n}\langle\nabla f(\bar{\mathbf{x}}_t), \alpha_i\nabla F_i(\mathbf{x}_{i,t})\rangle| \leq \frac{1}{n}\sum_{i=1}^{n}||\nabla f(\bar{\mathbf{x}}_t)||||\alpha_i\nabla F_i(\mathbf{x}_{i,t})|| \leq c||\nabla f(\bar{\mathbf{x}}_t)|| .$$
$$\tag{D.35}$$

We next derive the probability $\Pr(||\nabla F_i(\mathbf{x}_{i,t}) - \nabla f(\bar{\mathbf{x}}_t)|| > 12\sigma + 12||\nabla f_i(\bar{\mathbf{x}}_t) - \nabla f(\bar{\mathbf{x}}_t)|| + 12(L_0 + ||\nabla f(\bar{\mathbf{x}}_t)||L_1)||\mathbf{x}_{i,t} - \bar{\mathbf{x}}_t||$

$$\Pr(||\nabla F_i(\mathbf{x}_{i,t}) - \nabla f(\bar{\mathbf{x}}_t)|| > 12\sigma + 12||\nabla f_i(\bar{\mathbf{x}}_t) - \nabla f(\bar{\mathbf{x}}_t)|| + 12(L_0 + L_1||\nabla f(\bar{\mathbf{x}}_t)||)||\mathbf{x}_{i,t} - \bar{\mathbf{x}}_t||)$$

$$\leq \frac{\mathbb{E}||\nabla F_i(\mathbf{x}_{i,t}) - \nabla f(\bar{\mathbf{x}}_t)||}{12\sigma + 12||\nabla f_i(\bar{\mathbf{x}}_t) - \nabla f(\bar{\mathbf{x}}_t)|| + 12(L_0 + ||\nabla f(\bar{\mathbf{x}}_t)||L_1)||\mathbf{x}_{i,t} - \bar{\mathbf{x}}_t||}$$

$$\leq \frac{1}{12} \tag{D.36}$$

Therefore, combining the two cases, we have:

$$-\frac{1}{n}\sum_{i=1}^{n}\alpha_i\langle\nabla f(\bar{\mathbf{x}}_t), \nabla F_i(\mathbf{x}_{i,t})\rangle \leq -\Pr(\delta_{c-} = 0)\frac{1}{n}\sum_{i=1}^{n}\alpha_i\mathbb{E}[\langle\nabla f(\bar{\mathbf{x}}_t), \nabla F_i(\mathbf{x}_{i,t})|\delta_{c-} = 0]$$

$$- \Pr(\delta_{c-} = 1)\frac{1}{n}\sum_{i=1}^{n}\alpha_i\mathbb{E}[\langle\nabla f(\bar{\mathbf{x}}_t), \nabla F_i(\nabla\mathbf{x}_{i,t})\rangle|\delta_{c-} = 1] \tag{D.37}$$

$$\leq (-\frac{11}{12}\frac{13}{67} + \frac{1}{12})c||\nabla f(\bar{\mathbf{x}}_t)|| \leq -\frac{1}{11}c||\nabla f(\bar{\mathbf{x}}_t)|| .$$

$$\mathbb{E}f(\bar{\mathbf{x}}_{t+1}) \leq f(\bar{\mathbf{x}}_t) - \frac{\eta}{n}\sum_{i=1}^{n}\mathbb{E}\langle\nabla f(\bar{\mathbf{x}}_t), \nabla F_i(\bar{\mathbf{x}}_t)\rangle + \frac{3\eta^2(L_0 + L_1||\nabla f(\bar{\mathbf{x}}_t)||)}{10}c||\nabla f(\bar{\mathbf{x}}_t)||$$

$$\leq f(\bar{\mathbf{x}}_t) - \frac{\eta c}{11}||\nabla f(\bar{\mathbf{x}}_t)|| . \tag{D.38}$$

Rearrange the above equation, we have:

$$\frac{c}{11}||\nabla f(\bar{\mathbf{x}}_t)|| \leq \frac{f(\bar{\mathbf{x}}_t) - f(\bar{\mathbf{x}}_{t+1})}{\eta} . \tag{D.39}$$

$\square$

**Wrap up** If there is at least one iteration such that the gradient norm is small $||\nabla f(\bar{\mathbf{x}}_t)|| < \mathcal{O}(\sigma + \zeta)$, then it simply holds that:

$$\min_{t \in [1,T]} \mathbb{E}||\nabla f(\bar{\mathbf{x}}_t)|| \leq \mathcal{O}(\sigma + \zeta) .$$

Otherwise, we can use Eq. D.38 to bound the gradient norm, combining the above two cases and averaging over $T$ iterations, we have:

$$\min_{t \in [1,T]} \mathbb{E}||\nabla f(\bar{\mathbf{x}}_t)|| \leq \mathcal{O}\left(\frac{f(\mathbf{x}_0) - f(\mathbf{x}^*)}{\eta c T} + \sigma + \zeta\right) . \tag{D.40}$$

**When $c > \mathcal{O}(\sigma + \zeta_{\max})$**

When we look at Theorem I and let $c \to \infty$, compared to the convergence rate from Koloskova et al. (2020), the only terms that are different from the unclipped federated optimization convergence are: $\mathcal{O}(\tau^2 \sigma^2)$ and $\mathcal{O}(\sqrt{\frac{\zeta^2}{n}})$. These two terms appear in the case discussion of when $||\nabla f(\bar{\mathbf{x}}_t)|| < \frac{c}{2}$. Therefore, we mainly show the proof in the case $||\nabla f(\bar{\mathbf{x}}_t)|| < \frac{c}{2}$ to illustrate that we can recover FedAvg.

For the remaining of the proof, it is important to note that when $||\nabla f(\bar{\mathbf{x}}_t)|| < \frac{c}{2}$ and $c > 32 \cdot 12(\sigma + \zeta_{\max})$, we have:

$$||\nabla f_i(\bar{\mathbf{x}}_t)|| \leq ||\nabla f_i(\bar{\mathbf{x}}_t) - \nabla f(\bar{\mathbf{x}}_t)|| + ||\nabla f(\bar{\mathbf{x}}_t)|| \leq \zeta_{\max} + \frac{c}{2} < c$$

We first give an updated difference lemma that recovers $\tau \sigma^2$:

**Lemma 17.** *For $\eta \leq \frac{1}{14(L_0 + \min(c,M)L_1)\tau}$ and $c > \mathcal{O}(\sigma + \zeta_{\max})$, it holds:*

$$\mathbb{E}R_t \leq \frac{109\tau\eta^2}{99} \sum_{j \in \mathcal{J}_{c+,t} \cup \mathcal{J}_{\frac{c}{2}+,t}} c^2 + \frac{77\eta^2\tau}{20} \sum_{j \in \mathcal{J}_{\frac{c}{2}-,t}} ||\nabla f(\bar{\mathbf{x}}_j)||^2 + \frac{288\eta(\sigma^2 + \zeta^2)^2}{c^2} + \frac{436\eta^2\tau}{99}\sigma^2 + \frac{218\eta^2\tau^2}{99}\zeta^2$$

*Proof.*

$$\frac{1}{n}\sum_{i=1}^{n}||\sum_{j \in \mathcal{J}_{\frac{c}{2}-,t}}\mathbf{g}_{i,j}||^2 = \frac{1}{n}\sum_{i=1}^{n}||\sum_{j \in \mathcal{J}_{\frac{c}{2}-,t}}\mathbf{g}_{i,j} - \mathbb{E}\mathbf{g}_{ij}||^2 + \frac{1}{n}\sum_{i=1}^{n}||\sum_{j \in \mathcal{J}_{\frac{c}{2}-,t}}\mathbb{E}\mathbf{g}_{ij}||^2$$

$$\leq \sum_{j \in \mathcal{J}_{\frac{c}{2}-,t}} \underbrace{\frac{1}{n}\sum_{i=1}^{n}||\mathbf{g}_{i,j} - \mathbb{E}\mathbf{g}_{i,j}||^2}_{\mathcal{A}_2} + \underbrace{\frac{1}{n}\sum_{i=1}^{n}||\sum_{j \in \mathcal{J}_{\frac{c}{2}-,t}}\mathbb{E}\mathbf{g}_{ij}||^2}_{\mathcal{A}_3} . \tag{D.41}$$

The first equality uses mean variance separation. The second inequality uses the fact that the variance of the sum of the independent variables equal to the sum of the variance. We next bound $\mathcal{A}_2$ and $\mathcal{A}_3$ separately.

$$\mathcal{A}_2 := \frac{1}{n}\sum_{i=1}^{n}||\mathbf{g}_{ij} - \nabla f_i(\bar{\mathbf{x}}_j) + \nabla f_i(\bar{\mathbf{x}}_j) - \mathbb{E}\mathbf{g}_{ij}||^2$$

$$\leq \frac{2}{n}\sum||\mathbf{g}_{ij} - \nabla f_i(\bar{\mathbf{x}}_j)||^2 + \frac{2}{n}\sum||\nabla f_i(\bar{\mathbf{x}}_j) - \mathbb{E}\mathbf{g}_{ij}||^2$$

$$\leq \frac{2}{n}\sum||\nabla F_i(\mathbf{x}_{ij}) - \nabla f_i(\bar{\mathbf{x}}_j)||^2 + \frac{2}{n}\sum||\nabla f_i(\bar{\mathbf{x}}_j) - \mathbb{E}\mathbf{g}_{i,j}||^2$$

$$\leq 4\sigma^2 + 4(L_0 + ||\nabla f(\bar{\mathbf{x}}_j)||L_1)^2 R_j + \frac{4}{n}\sum \Pr(\delta_{ij} = 1)||\nabla F_i(\mathbf{x}_{ij})||^2 + 4(L_0 + ||\nabla f(\bar{\mathbf{x}}_j)||L_1)^2 R_j$$

$$\leq 4\sigma^2 + \frac{288(\sigma^2 + \zeta^2)^2}{c^2} + \left(\frac{19}{2} + \frac{72}{49}\right)L^2 R_j + (\frac{1}{4} + \frac{96}{196})||\nabla f(\bar{\mathbf{x}}_j)||^2 , \tag{D.42}$$

The first inequality uses the triangle inequality. The second inequality uses the projection Lemma 3. The third inequality following the same procedure as in Eq. D.12. The last inequality uses the result from Eq. D.22, the definition of $L := L_0 + \min(c,M)L_1$ and the condition $L_0 + ||\nabla f(\bar{\mathbf{x}}_j)||L_1 \leq L$.

$$\mathcal{A}_3 := \frac{1}{n}\sum||\sum_{j \in \mathcal{J}_{\frac{c}{2}-}}\mathbb{E}\mathbf{g}_{ij}||^2 = \frac{1}{n}\sum||\sum_{j \in \mathcal{J}_{\frac{c}{2}-,t}}\mathbb{E}\mathbf{g}_{ij} - \nabla f_i(\bar{\mathbf{x}}_j) + \nabla f_i(\bar{\mathbf{x}}_j)||^2$$

$$\leq \frac{2}{n}\sum\|\sum_{j\in\mathcal{J}_{\frac{c}{2}-,t}}\mathbb{E}\mathbf{g}_{ij}-\nabla f_i(\mathbf{x}_j)\|^2+\frac{2}{n}\sum\|\sum_{j\in\mathcal{J}_{\frac{c}{2}-,t}}\nabla f_i(\bar{\mathbf{x}}_j)-\nabla f(\bar{\mathbf{x}}_j)+\nabla f(\bar{\mathbf{x}}_j)\|^2$$

$$\leq \frac{2}{n}\sum\|\sum_{j\in\mathcal{J}_{\frac{c}{2}-}}\delta_{ij}\nabla F_i(\mathbf{x}_{ij})+\nabla f_i(\mathbf{x}_{ij})-\nabla f_i(\bar{\mathbf{x}}_j)\|^2+2\tau^2\zeta^2+2\tau\sum_{j\in\mathcal{J}_{\frac{c}{2}-,t}}\|\nabla f(\bar{\mathbf{x}}_j)\|^2$$

$$\leq \frac{4}{n}\sum_n\prod_j\Pr(\delta_{ij}=1)\|\sum_{j\in\mathcal{J}_{\frac{c}{2}-,t}}\nabla F_i(\mathbf{x}_{ij})\|^2+4\tau\sum_{j\in\mathcal{J}_{\frac{c}{2}-,t}}(L_0+\|\nabla f(\bar{\mathbf{x}}_j)\|L_1)^2R_j+2\tau^2\zeta^2+2\tau\sum_{j\in\mathcal{J}_{\frac{c}{2}-,t}}\|\nabla f(\bar{\mathbf{x}}_j)\|^2$$

$$\leq \frac{4\tau}{n}\left(\frac{12(\sigma^2+\zeta^2)}{c^2}+\frac{12(L_0+\|\nabla f(\bar{\mathbf{x}}_j)\|L_1)^2R_j}{c^2}\right)^\tau\sum_i\sum_{j\in\mathcal{J}_{\frac{c}{2}-}}\|\nabla F_i(\mathbf{x}_{ij})\|^2+4\tau(L_0+\|\nabla f(\bar{\mathbf{x}}_j)\|L_1)^2\sum_{j\in\mathcal{J}_{\frac{c}{2}-,t}}R_j$$

$$+2\tau^2\zeta^2+2\tau\sum_{j\in\mathcal{J}_{\frac{c}{2}-,t}}\|\nabla f(\bar{\mathbf{x}}_j)\|^2$$

$$\leq \left(\frac{48(\sigma^2+\zeta^2)}{c^2}+\frac{48(L_0+\|\nabla f(\bar{\mathbf{x}}_j)\|L_1)^2R_j}{c^2}\right)\frac{1}{n}\sum_i\sum_{j\in\mathcal{J}_{\frac{c}{2}-,t}}\|\nabla F_i(\mathbf{x}_{ij})\|^2+4\tau(L_0+\|\nabla f(\bar{\mathbf{x}}_j)\|L_1)^2\sum_{j\in\mathcal{J}_{\frac{c}{2}-,t}}R_j$$

$$+2\tau^2\zeta^2+2\tau\sum_{j\in\mathcal{J}_{\frac{c}{2}-,t}}\|\nabla f(\bar{\mathbf{x}}_j)\|^2$$

$$\leq \frac{288(\sigma^2+\zeta^2)^2\tau}{c^2}+\frac{291L^2}{98}\sum_{j\in\mathcal{J}_{\frac{c}{2}-,t}}R_j+(\frac{145}{196}+2\tau)\sum_{j\in\mathcal{J}_{\frac{c}{2}-,t}}\|\nabla f(\bar{\mathbf{x}}_j)\|^2$$

$$+4\tau L^2\sum_{j\in\mathcal{J}_{\frac{c}{2}-,t}}R_j+2\tau^2\zeta^2$$

$$\leq \frac{288\tau(\sigma^2+\zeta^2)^2}{c^2}+\frac{291}{98}L^2\sum_{j\in\mathcal{J}_{\frac{c}{2}-,t}}R_j+(\frac{145}{196}+2\tau)\sum_{j\in\mathcal{J}_{\frac{c}{2}-,t}}\|\nabla f(\bar{\mathbf{x}}_j)\|^2+4\tau L^2\sum_{j\in\mathcal{J}_{\frac{c}{2}-,t}}R_j+2\tau^2\zeta^2,$$

The first inequality uses triangle inequality. The second inequality uses Cauchy-Schwartz inequality. The third inequality uses triangle inequality. The fourth inequality uses the result from Eq. D.22. The fifth inequality uses the fact that $a^\tau\leq\frac{1}{a^\tau}$ when $a<1$. The sixth inequality uses the definition $L:=L_0+\min(c,M)L_1$ and $L_0+\|\nabla f(\bar{\mathbf{x}}_j)\|L_1\leq L$ since $j\in\mathcal{J}_{\frac{c}{2}-,t}$. Combining the above two equations, we have:

$$\frac{\eta^2}{n}\sum_{i=1}^n\|\sum_{j\in\mathcal{J}_{\frac{c}{2}-,t}}\mathbf{g}_{i,t}\|^2\leq 4\eta^2\tau\sigma^2+\frac{288\eta^2\tau(\sigma^2+\zeta^2)^2}{c^2}+\frac{1075\eta^2L^2}{98}\sum_{j\in\mathcal{J}_{\frac{c}{2}-}}R_j+\frac{145\eta^2}{196}\sum_{j\in\mathcal{J}_{\frac{c}{2}-,t}}\|\nabla f(\bar{\mathbf{x}}_j)\|^2$$

$$+\frac{288\eta^2\tau(\sigma^2+\zeta^2)^2}{c^2}+\frac{291\eta^2L^2}{98}\sum_{j\in\mathcal{J}_{\frac{c}{2}-,t}}R_j+\frac{537\eta^2\tau}{196}\sum_{j\in\mathcal{J}_{\frac{c}{2}-,t}}\|\nabla f(\bar{\mathbf{x}}_j)\|^2$$

$$+4\eta^2\tau L^2\sum_{j\in\mathcal{J}_{\frac{c}{2}-,t}}R_j+2\eta^2\tau^2\zeta^2$$

$$\leq \frac{576\eta^2\tau(\sigma^2+\zeta^2)^2}{c^2}+\frac{1758\eta^2\tau L^2}{98}\sum_{j\in\mathcal{J}_{\frac{c}{2}-,t}}R_j+\frac{682\eta^2\tau}{196}\sum_{j\in\mathcal{J}_{\frac{c}{2}-,t}}\|\nabla f(\bar{\mathbf{x}}_j)\|^2$$

$$+4\eta^2\tau\sigma^2+2\eta^2\tau^2\zeta^2. \tag{D.43}$$

$$\mathbb{E}R_t\leq\tau\eta^2\sum_{j\in\mathcal{J}_{c+,t}\cup\mathcal{J}_{\frac{c}{2}+,t}}c^2+\frac{576\eta^2\tau(\sigma^2+\zeta^2)^2}{c^2}+\frac{1758\eta^2\tau L^2}{98}\sum_{j\in\mathcal{J}_{\frac{c}{2}-,t}}R_j$$

$$+\frac{682\eta^2\tau}{196}\sum_{j\in\mathcal{J}_{\frac{c}{2}-,t}}\|\nabla f(\bar{\mathbf{x}}_j)\|^2+4\eta^2\tau\sigma^2+2\eta^2\tau^2\zeta^2. \tag{D.44}$$

Therefore, given $\eta \leq \frac{1}{14L\tau}$, we have:

$$\mathbb{E}R_t \leq \frac{109\tau\eta^2}{99} \sum_{j \in \mathcal{J}_{c+,t} \cup \mathcal{J}_{\frac{c}{2}+,t}} c^2 + \frac{77\eta^2\tau}{20} \sum_{j \in \mathcal{J}_{\frac{c}{2}-,t}} ||\nabla f(\bar{\mathbf{x}}_j)||^2 + \frac{288\eta(\sigma^2+\zeta^2)^2}{c^2} + \frac{436\eta^2\tau}{99}\sigma^2 + \frac{218\eta^2\tau^2}{99}\zeta^2 .$$

(D.45)

$\square$

We next bound the term $\mathbb{E}||\frac{1}{n}\sum_{i=1}^{n} \mathbf{g}_{i,t}||^2$ given $L := L_0 + \min(c, M)L_1$.

$$\frac{1}{2}\mathbb{E}||\frac{1}{n}\sum_{i=1}^{n} \mathbf{g}_{i,t}||^2 = \frac{1}{2}\mathbb{E}||\frac{1}{n}\sum_{i=1}^{n} \mathbf{g}_{i,t} - \mathbb{E}\mathbf{g}_{i,t}||^2 + \frac{1}{2}||\frac{1}{n}\sum_{i=1}^{n} \mathbb{E}\mathbf{g}_{i,t}||^2 = \frac{1}{2n^2}\sum_{i=1}^{n} \mathbb{E}||\mathbf{g}_{i,t} - \mathbb{E}\mathbf{g}_{it}||^2 + \frac{1}{2}||\frac{1}{n}\sum_{i=1}^{n} \mathbb{E}\mathbf{g}_{i,t}||^2$$

$$\leq \frac{1}{n^2}\sum_{i=1}^{n} \mathbb{E}||\mathbf{g}_{i,t} - \nabla f_i(\bar{\mathbf{x}}_t)||^2 + \frac{1}{n^2}\sum_{i=1}^{n} ||\nabla f_i(\bar{\mathbf{x}}_t) - \mathbb{E}\mathbf{g}_{i,t}||^2 + \frac{1}{2}||\frac{1}{n}\sum_{i=1}^{n} \mathbb{E}\mathbf{g}_{i,t}||^2$$

$$\leq \frac{1}{n^2}\sum_{i=1}^{n} \mathbb{E}||\nabla F_i(\mathbf{x}_{i,t}) - \nabla f_i(\bar{\mathbf{x}}_t)||^2 + \frac{1}{n^2}\sum_{i=1}^{n} \mathbb{E}||\nabla f_i(\bar{\mathbf{x}}_t) - \mathbb{E}\mathbf{g}_{i,t}||^2 + \frac{1}{2}||\frac{1}{n}\sum_{i=1}^{n} \mathbb{E}\mathbf{g}_{i,t}||^2$$

$$\leq \frac{2\sigma^2}{n} + \frac{2L^2}{n}R_t + \frac{1}{n^2}\sum_{i=1}^{n} \Pr(\delta_{i,t}=1)\mathbb{E}||\nabla F_i(\mathbf{x}_{i,t})||^2 + \frac{1}{2}||\frac{1}{n}\sum_{i=1}^{n} \mathbb{E}\mathbf{g}_{i,t}||^2$$

$$\leq \frac{2\sigma^2}{n} + \frac{2L^2}{n}R_t + \frac{1}{2}||\frac{1}{n}\sum_{i=1}^{n} \mathbb{E}\mathbf{g}_{i,t}||^2$$
$$+ \frac{1}{n}\left(\frac{144(\sigma^2+\zeta^2)^2}{c^2} + (\frac{7}{2} + \frac{144}{196})L^2 R_t + \frac{48(\sigma^2+\zeta^2) + 48L^2 R_t}{c^2}||\nabla f(\bar{\mathbf{x}}_t)||^2\right)$$

$$\leq \frac{2\sigma^2}{n} + \frac{611L^2}{98}R_t + \frac{144(\sigma^2+\zeta^2)^2}{c^2} + \frac{31}{125}||\nabla f(\bar{\mathbf{x}}_t)||^2 + \frac{1}{2}||\frac{1}{n}\sum_{i=1}^{n} \mathbb{E}\mathbf{g}_{i,t}||^2 .$$

(D.46)

Therefore, replacing the result of $\mathbb{E}||\mathbf{g}_{i,t}||^2$ from Eq. D.25 with the above updated result and replacing the difference lemma with the updated difference lemma in Eq. D.27, with carefully tuned stepsize, we obtain:

$$\min_{t \in [1,T]} \mathbb{E}||\nabla f(\bar{\mathbf{x}}_t)|| \leq \mathcal{O}\left(\frac{f(\mathbf{x}_0) - f^*}{\eta c T} + \sqrt{\frac{f(\mathbf{x}_0) - f^*}{\eta T}}\right.$$
$$+ \sqrt{\frac{\eta L \sigma^2}{n}}$$
$$\left. + \min\left(L\eta\sqrt{\tau\sigma^2 + \tau^2\zeta^2}, c\right) + \min\left(\sigma + \zeta, \frac{\sigma^2+\zeta^2}{c}\right)\right).$$

When $c \to \infty$, the above result recover the convergence rate provided in Koloskova et al. (2020).

## D.3 EXTENSION TO DIFFERENTIALLY PRIVATE LOCAL SGD

In this section, we add random noise $z_{i,t} \sim \mathcal{N}(0, \frac{\sigma_{DP}^2}{d}\mathbf{I})$ where $d$ is the dimension of the parameter on the clipped gradients. Therefore, the conditional expectation becomes:

$$f(\bar{\mathbf{x}}_{t+1}) \leq f(\bar{\mathbf{x}}_t) - \frac{\eta}{n}\sum_{i=1}^{n} \mathbb{E}\langle\nabla f(\bar{\mathbf{x}}_t), \mathbf{g}_{i,t}\rangle - \frac{\eta}{n}\sum_{i=1}^{n} \mathbb{E}\langle\nabla f(\bar{\mathbf{x}}_t), \mathbf{z}_{i,t}\rangle$$
$$+ \frac{\eta^2(L_0 + L_1||\nabla f(\bar{\mathbf{x}}_t)||)}{2}\mathbb{E}||\frac{1}{n}\sum_{i=1}^{n} \mathbf{g}_{i,t}||^2 + \frac{\eta^2(L_0 + L_1||\nabla f(\bar{\mathbf{x}}_t)||)}{2n}\mathbb{E}||\mathbf{z}_{i,t}||^2$$
$$\leq f(\bar{\mathbf{x}}_t) - \frac{\eta}{n}\sum_{i=1}^{n} \mathbb{E}\langle\nabla f(\bar{\mathbf{x}}_t), \mathbf{g}_{i,t}\rangle + \frac{\eta^2(L_0 + L_1||\nabla f(\bar{\mathbf{x}}_t)||)}{2}\mathbb{E}||\frac{1}{n}\sum_{i=1}^{n} \mathbf{g}_{i,t}||^2 + \frac{\eta^2(L_0 + L_1||\nabla f(\bar{\mathbf{x}}_t)||)}{2n}\mathbb{E}||\mathbf{z}_{i,t}||^2$$
$$\leq f(\bar{\mathbf{x}}_t) - \frac{\eta}{n}\sum_{i=1}^{n} \mathbb{E}\langle\nabla f(\bar{\mathbf{x}}_t), \mathbf{g}_{i,t}\rangle + \frac{\eta^2(L_0 + L_1||\nabla f(\bar{\mathbf{x}}_t)||)}{2}\mathbb{E}||\frac{1}{n}\sum_{i=1}^{n} \mathbf{g}_{i,t}||^2 + \frac{\eta^2(L_0 + L_1||\nabla f(\bar{\mathbf{x}}_t)||)}{2n}\sigma_{DP}^2 .$$

(D.47)

The second inequality uses the fact that $\mathbb{E}[\mathbf{z}_{it}] = 0$. The third inequality uses the fact that $\text{Var}[\mathbf{z}_{i,t}] = \sigma_{DP}^2$. The rest of the proof is the same as before.

### D.4 ACCURACY PERSPECTIVE

In this section, we discuss how to choose $\eta$ and $c$ to reach $\varepsilon$-accuracy for Algorithm 1 ($\min_{t\in[1,T]} \mathbb{E}[||\nabla f(\bar{\mathbf{x}}_t)||] \leq \varepsilon$). According to Theorem I, we obtain the following convergence result:

$$\min_{t\in[1,T]} \mathbb{E}||\nabla f(\bar{\mathbf{x}}_t)|| \leq \mathcal{O}\left(\frac{F_0}{\eta cT} + \sqrt{\frac{F_0}{\eta T}} + \sqrt{\frac{\eta L(\sigma^2+\zeta^2)}{n}}\right.$$
$$\left. + \eta L \min\left(\tau\sqrt{\sigma^2+\zeta^2}, c\right) + \min\left(\sigma+\zeta, \frac{\sigma^2+\zeta^2}{c}\right)\right).$$

Since the last three terms do not depend on $T$, we have to pick $\eta$ and $c$ such that each individual term is less than $\varepsilon$. It is clear from the last term that we can pick $c$ to be:

$$c = \Theta\left(\frac{\sigma^2+\zeta^2}{\varepsilon}\right). \tag{D.48}$$

Let $A := \min(\tau\sqrt{\sigma^2+\zeta^2}, \frac{\sigma^2+\zeta^2}{\varepsilon})$. Together with the constraint $\eta \leq \mathcal{O}(\frac{1}{\tau L})$, we obtain the choice for $\eta$ as follows:

$$\eta \leq \min\left\{\Theta\left(\frac{n\varepsilon^2}{L(\sigma^2+\zeta^2)}\right), \Theta\left(\frac{\varepsilon}{LA}\right), \Theta\left(\frac{1}{\tau L}\right)\right\}. \tag{D.49}$$

Finally, we can compute the required iteration $T$ by plugging in the respective choices of $\eta$ and $c$ and letting the first and the second terms be less than $\varepsilon^2$.

### D.5 PRIVACY-UTILITY DISCUSSION

#### D.5.1 LOCAL DP GUARANTEE

When the mini-batch size is one on each client, per-sample clipping with the DP-noise can limit the contribution of each individual data point on the client model update, and achieve a certain level of differential privacy. Here, the neighborhood dataset is defined such that two datasets only differ by one data point.

To proceed with the analysis, we assume the mini-batch size on each client is one and denote the minimum size among all clients by $N$, i.e., $\forall i \in [N]$, we have:

$$f_i(\mathbf{x}) := \frac{1}{N_i}\sum_{j=1}^{N_i} F_{i,j}(\mathbf{x}), \quad N := \min_i\{N_i\}, \tag{D.50}$$

$$\mathbb{E}_j[||\nabla F_{i,j}(\mathbf{x}) - \nabla f_i(\mathbf{x})||^2] = \frac{1}{N_i}\sum_{j=1}^{N_i}||\nabla F_{i,j}(\mathbf{x}) - \nabla f_i(\mathbf{x})||^2 \leq \sigma^2. \tag{D.51}$$

To achieve the formal local DP guarantee, the variance $\sigma_{\text{DP}}^2$ should satisfy the following condition. (Note the DP noise is set to $\mathbf{z}_i \sim \mathcal{N}(0, \frac{\sigma_{\text{DP}}^2}{d}\mathbf{I})$ for all $i \in [n]$.)

**Theorem VI** (Theorem I Abadi et al. (2016)). *For any $\varepsilon_{DP} \leq \mathcal{O}(\frac{\tau R}{N^2})$, $0 < \delta < 1$, $\tau \in \mathbb{N}^+$, $R \in \mathbb{N}^+$, Algorithm 1 with updating rule $\mathbf{y}_i \leftarrow \mathbf{y}_i - \eta(\mathbf{g}_i + \mathbf{z}_i)$ with $\mathbf{z}_i \sim \mathcal{N}(0, \frac{\sigma_{DP}^2}{d}\mathbf{I})$ for all $i \in [n]$ for $T := R \cdot \tau$ steps with $R$ communication rounds, $\tau$ local steps, achieves $(\varepsilon_{DP}, \delta)$-local differential privacy for any client $i \in [n]$ if we choose*

$$\bar{\sigma}_{DP} = \Omega\left(\frac{c\sqrt{\log(1/\delta)T}}{N\varepsilon_{DP}}\right). \tag{D.52}$$

*where $\bar{\sigma}_{DP} := \frac{\sigma_{DP}}{\sqrt{d}}$.*

*Proof.* Let $q := \frac{1}{N}$. According to the analysis from Abadi et al. (2016), suppose $\mathbf{y}_{i,t}$ satisfies $(\varepsilon', \delta')$-DP for any $t = 0, ..., \tau R - 1$ and any $i \in [n]$, then the total privacy guarantee after $R$ rounds is $\left(\mathcal{O}(q\sqrt{\tau R}\varepsilon'), \mathcal{O}(\delta')\right)$. It follows from the Gaussian mechanism (and let $\varepsilon_{\text{DP}} = q\sqrt{\tau R}\varepsilon'$, $\delta = \delta'$) that, each $\bar{\sigma}_{\text{DP}}$ should satisfy

$$\bar{\sigma}_{\text{DP}} = \Omega\left(\frac{\Delta q\sqrt{\log(1/\delta)\tau R}}{\varepsilon_{\text{DP}}}\right). \tag{D.53}$$

where $\Delta$ is $\ell_2$-sensitivity of the algorithm output at each iteration, which can be bounded by $2c$ since $||\text{clip}_c(\mathbf{x}) - \text{clip}_c(\mathbf{y})|| \leq 2c$ for any $\mathbf{x}, \mathbf{y} \in \mathbb{R}^d$. (Note for each client, the minibatch size is assumed to be 1.) $\qquad\square$

### D.5.2 OPTIMAL PRIVACY-UTILITY TRADE-OFF AND CHOICES FOR HYPER-PARAMETERS

In this section, we provide the choices of hyper-parameters including $\eta$ and $c$ to achieve the optimal privacy utility trade-off and the optimal iteration complexity.

Let $\mathcal{K} := \sigma^2 + \zeta^2$ and take the square for both sides in Eq. 3, we have:

$$\min_t \mathbb{E}||\nabla f(\bar{\mathbf{x}}_t)||^2 \leq \mathcal{O}\left(\frac{F_0^2}{\eta^2 c^2 T^2} + \frac{F_0}{\eta T} + \frac{\eta L \mathcal{K}}{n} + L^2 \eta^2 \min(\tau^2 \mathcal{K}, c^2) + \min\left((\sigma+\zeta)^2, \frac{\mathcal{K}^2}{c^2}\right) + \frac{\eta^2 L^2 \sigma_{\mathrm{DP}}^4}{n^2 c^2} + \frac{\eta L}{n}\sigma_{\mathrm{DP}}^2\right)$$

According to Theorem VI, in order to achieve $(\varepsilon_{\mathrm{DP}}, \delta)$-local DP for each client, the noise scale should satisfy $\sigma_{\mathrm{DP}}^2 \geq \Omega(c^2 T \nu)$ where $\nu = \frac{d \log(1/\delta)}{N^2 \varepsilon_{\mathrm{DP}}^2}$. Plugging the bound for $\sigma_{\mathrm{DP}}$ into the previous display, and minimize the first and the last two terms w.r.t $c$ and $\eta$, we have $T = \Theta\left(\frac{\sqrt{F_0 n}}{\eta c \sqrt{Lv}}\right) = \Theta\left(\frac{B}{\eta c}\right)$ where $B := \frac{\sqrt{F_0 n}}{\sqrt{Lv}}$. Let $A := \frac{F_0}{B}$, we can then rewrite the above rate as:

$$\min_t \mathbb{E}||\nabla f(\bar{\mathbf{x}}_t)||^2 \leq \mathcal{O}\left(Ac + \frac{\eta L \mathcal{K}}{n} + L^2 \eta^2 \min(\tau^2 \mathcal{K}, c^2) + \min\left((\sigma+\zeta)^2, \frac{\mathcal{K}^2}{c^2}\right) + A^2\right) . \quad (D.54)$$

**Case 1:** When $A^2$ is larger than $\mathcal{K}$ such that $\mathcal{O}(Ac + A^2)$ dominates the error, we can set $c$ as large as $A$ and pick the stepsize as large as $\eta = \Theta\left(\frac{1}{L\tau}\right)$ to reduce the number of iterations $T$. We then obtain the privacy-utility trade-off as:

$$\min_t \mathbb{E}||\nabla f(\bar{\mathbf{x}}_t)||^2 \leq \mathcal{O}\left(\frac{F_0 Lv}{n}\right), \qquad T = \Theta\left(\frac{\tau \sqrt{F_0 n L}}{c\sqrt{v}}\right) . \quad (D.55)$$

**Case 2:** Suppose $\mathcal{K}^2$ is much larger than $A$ such that $\min\left((\sigma+\zeta)^2, \frac{\mathcal{K}^2}{c^2}\right)$ dominates the error. As $\mathcal{O}\left((\sigma+\zeta)^2\right)$ is fixed, we can instead reduce the term $\mathcal{O}\left(\frac{\mathcal{K}^2}{c^2}\right)$ by letting $c = \Theta\left(\frac{\mathcal{K}^{2/3}}{A^{1/3}}\right)$ such that the rate from Eq. D.54 can be reduced to:

$$\min_t \mathbb{E}||\nabla f(\bar{\mathbf{x}}_t)||^2 \leq \mathcal{O}\left(\frac{\eta L \mathcal{K}}{n} + L^2 \eta^2 \min(\tau^2 \mathcal{K}, \frac{\mathcal{K}^{4/3}}{A^{2/3}}) + \mathcal{K}^{2/3} A^{2/3}\right) .$$

We can then pick $\eta$ as large as $\eta = \Theta\left(\frac{A^{1/3}}{L\tau \mathcal{K}^{1/3}}\right)$ to reduce the number of iterations $T$ so that: $\mathcal{O}\left(\frac{\eta L \mathcal{K}}{n}\right) \leq \mathcal{O}\left(\frac{A^{1/3}\mathcal{K}^{2/3}}{\tau n}\right) \leq \mathcal{O}\left(\mathcal{K}^{2/3} A^{2/3}\right)$ and $\mathcal{O}(L^2 \eta^2 \tau^2 \mathcal{K}) \leq \mathcal{O}\left(A^{2/3}\mathcal{K}^{1/3}\right) \leq \mathcal{O}(\mathcal{K}^{2/3} A^{2/3})$. We then obtain the privacy-utility trade-off as:

$$\min_t \mathbb{E}||\nabla f(\bar{\mathbf{x}}_t)||^2 \leq \mathcal{O}\left(\mathcal{K}^{2/3}\left(\sqrt{\frac{F_0 Lv}{n}}\right)^{2/3}\right), \quad T = \Theta\left(\frac{\tau(L\mathcal{K}F_0)^{1/3}n^{2/3}}{cv^{2/3}}\right) . \quad (D.56)$$

### D.5.3 IMPLICATION

**Case 1:** Let us begin by examining Case 1 characterized by a relatively low privacy budget, with $\sigma$ and $\zeta$ being relatively small. Plugging $\nu = \frac{d \log(1/\delta)}{N^2 \varepsilon_{\mathrm{DP}}^2}$ into D.55, we obtain the optimal privacy-utility trade-off as well as the best iteration complexity as follows:

$$\min_t \mathbb{E}[||\nabla f(\bar{\mathbf{x}}_t)||^2] \leq \mathcal{O}\left(\frac{d \log(1/\delta)}{n N^2 \varepsilon_{\mathrm{DP}}^2}\right), \qquad T = \Theta\left(\frac{\tau n N^2 \varepsilon_{\mathrm{DP}}^2}{d \log(1/\delta)}\right), \quad (D.57)$$

by choosing

$$c = \Theta\left(\frac{\sqrt{d \log(1/\delta)}}{\sqrt{n} N \varepsilon_{\mathrm{DP}}}\right), \quad \eta = \Theta\left(\frac{1}{\tau L}\right) . \quad (D.58)$$

Due to the weaker notion of neighboring dataset compared with per-update clipping, the optimal utility is $N^2$ times better than $\mathcal{O}(\frac{d \log(1/\delta)}{n \varepsilon_{\mathrm{DP}}^2})$ obtained in Section C.6.3. Further, the best iteration complexity is achieved by letting $\tau = 1$ which is the same as per-update clipping. Again, under the current analysis, we cannot prove effectiveness for doing multiple local steps when the privacy budget $\varepsilon$ and $\delta$ are relatively small.

**Case 2:** Let us now consider Case 2 where we have large stochastic noise and heterogeneity. Plugging in the value $\nu = \frac{d \log(1/\delta)}{N^2 \varepsilon_{\text{DP}}^2}$, we then have:

$$\min_t \mathbb{E}||\nabla f(\bar{\mathbf{x}}_t)||^2 \leq \mathcal{O}\left((\sigma^2 + \zeta^2)^{\frac{2}{3}}\left(\frac{d \log(1/\delta)}{nN^2\varepsilon_{\text{DP}}}\right)^{\frac{1}{3}}\right), \quad T = \Theta\left(\frac{\tau}{(\sigma^2+\zeta^2)^{\frac{1}{3}}}\sqrt{\frac{nN^2\varepsilon_{\text{DP}}^2}{d\log(1/\delta)}}\right)$$

by choosing:

$$c = \Theta\left((\sigma^2+\zeta^2)^{\frac{2}{3}}\left(\frac{nN^2\varepsilon_{\text{DP}}^2}{d\log(1/\delta)}\right)^{1/6}\right), \quad \eta = \Theta\left(\frac{1}{\tau(\sigma^2+\zeta^2)^{1/3}}\left(\frac{d\log(1/\delta)}{nN^2\varepsilon_{\text{DP}}^2}\right)^{\frac{1}{6}}\right)$$

In this case, the higher stochastic noise $\sigma^2$ and data heterogeneity $\zeta^2$ can worse the optimal utility. We need to pick a slightly large clipping threshold and small stepsize seeing that $c = \Theta\left((\sigma^2+\zeta^2)^{2/3}\right)$ and $\eta = \Theta\left(\frac{1}{\tau(\sigma^2+\zeta^2)^{1/3}}\right)$. Again, a larger number of local steps $\tau$ can increase the number of iterations $T$ to reach such utility. The optimal clipping threshold and stepsize in this case is a complicated interplay between $\sigma^2$, $\zeta^2$, problem dimension $d$, number of data point $N$, and the privacy cost $(\varepsilon_{\text{DP}}, \delta)$.

### D.5.4 CONNECTION TO CENTRALIZED DP-SGD

Following the same discussion as presented in C.6.4, per-sample clipping (Algorithm 1 with DP noise) can also be treated as a centralized algorithm when $\tau = 1$ from a pure algorithmic point of view.

Again, let $\zeta = 0$, $\tau = 1$, and mini-batch size on each client to be one. It is clear that, Algorithm 1 with DP noise is **exactly** equivalent to DP-SGD (where $n$ can be interpreted as the mini-batch used in Koloskova et al. (2023)):

$$\mathbf{x}_{t+1} = \mathbf{x}_t - \eta\frac{1}{n}\sum_{i=1}^{n}\left[\text{clip}_c\left(\nabla F_i(\mathbf{x}_t)\right) + \mathbf{z}_i\right]. \tag{D.59}$$

Suppose the size of the whole dataset is $N_{\text{total}}$. Then the target optimization problem becomes:

$$f(\mathbf{x}) := \frac{1}{N_{\text{total}}}\sum_{j=1}^{N_{\text{total}}}F_j(\mathbf{x}).$$

Essentially, the analysis in section D.5.2 recovers this special case. To guarantee the global DP guarantee, we can use the noise bound derived in (C.94):

$$\sigma_{\text{DP}}^2 = \frac{dnc^2\log(1/\delta)T}{N_{\text{total}}^2\varepsilon_{\text{DP}}^2}. \tag{D.60}$$

From the formula of $\sigma_{\text{DP}}^2$, we obtain $\nu = \frac{dn\log(1/\delta)}{N_{\text{total}}^2\varepsilon_{\text{DP}}^2}$. Following the same discussion as in Section D.5.3, suppose $\sigma$ is relatively small, then we obtain the following optimal privacy-utility trade-off and required number of iterations:

$$\min_{r\in[1,R]}\mathbb{E}[||\nabla f(\mathbf{x}_r)||^2 \leq \mathcal{O}\left(\frac{d\log(1/\delta)}{N_{\text{total}}^2\varepsilon_{\text{DP}}^2}\right), \quad T = \Theta\left(\frac{N_{\text{total}}^2\varepsilon_{\text{DP}}^2}{d\log(1/\delta)}\right). \tag{D.61}$$

by choosing

$$c = \Theta\left(\frac{\sqrt{d\log(1/\delta)}}{N_{\text{total}}\varepsilon_{\text{DP}}}\right), \quad \eta = \Theta(\frac{1}{L}). \tag{D.62}$$

Many researches Wang et al. (2018); Kifer et al. (2012) have shown that the optimal utility bound for DP-SGD under **bounded stochastic gradient** assumption is $\mathcal{O}\left(\frac{\sqrt{d\log(1/\delta)}}{N_{\text{total}}\varepsilon_{\text{DP}}}\right)$. We here obtain a slightly worse (higher order) utility bound due to the use of the more practical **bounded variance** assumption, but still keeps the same ratio between the privacy cost $(\varepsilon_{\text{DP}}, \delta)$, the problem dimension and the dataset size. The underlying reason is due to the higher order term $\frac{\eta^2 L^2 \sigma_{\text{DP}}^4}{n^2 c^2}$ that appears when the gradient norm $||\nabla f(\mathbf{x}_t)||$ is large.

# E  CONNECTION BETWEEN ALGORITHM 1 AND ALGORITHM 2

**Similarity in the special case:** When $\tau = 1$, the two algorithms are exactly the same given $\eta = \eta_l \eta_g$ and $c_{\text{per\_sample}} = \frac{c_{\text{per\_update}}}{\eta_l}$. This is because, by definition, $\eta_g \text{clip}_{c_{\text{per-update}}}(\eta_l \nabla F_i(\mathbf{y}_i)) = \eta_g \min(1, \frac{c_{\text{per-update}}}{\eta_l||\nabla F_i(\mathbf{y}_i)||})\eta_l \nabla F_i(\mathbf{y}_i) = \eta_l \eta_g \min(1, \frac{c_{\text{per-sample}}}{||\nabla F_i(\mathbf{y}_i)||})\nabla F_i(\mathbf{y}_i) = \eta \text{clip}_{c_{\text{per-sample}}}(\nabla F_i(\mathbf{y}_i))$. Theorem I and II can be simplified and give the same convergence guarantee in this case, that is:

Theorem I: $\mathcal{O}\left(\sqrt{\frac{F_0}{\eta T}} + \frac{F_0}{\eta_g CT} + \sqrt{\frac{\eta L(\sigma^2 + \zeta^2)}{n}} + \eta L \min(\sqrt{\sigma^2 + \zeta^2}, c) + \min(\sigma + \zeta, \frac{\eta_l(\sigma^2 + \zeta^2)}{c_{\text{per\_update}}})\right)$

Theorem II: $\mathcal{O}\left(\sqrt{\frac{F_0}{\eta R}} + \frac{F_0}{\eta_g CR} + \sqrt{\frac{\eta L}{n}}(\sigma + \zeta) + \eta_l L \zeta + \min(\sigma + \zeta, \frac{\eta_l(\sigma^2 + \zeta^2)}{c_{\text{per\_update}}})\right)$

Except for the fourth term that is caused by different distance lemma, we obtain the same convergence guarantee. Therefore, in this special case, we can adjust $c_{\text{per\_sample}}$ to allow per-sample clipping converges to any accuracy, which is essentially equivalent to the strategy of adjusting the inner stepsize in per-update clipping to reach any accuracy.

**Dissimilarity under arbitrary clipping threshold** We here provide an example where per-sample only converges to the neighborhood of the stationary point given *arbitrary* clipping threshold assuming $\tau = 1$.

Let $x = \begin{cases} 1, & p = 1/3 \\ 1, & p = 1/3 \\ -2, & p = 1/3 \end{cases}$, we here assume $x$ is a random vector as the gradient is stochastic:

- For per-update clipping, for any $c_{\text{per\_update}} > 0$, we can always **pick** $\eta_l \leq \frac{c_{\text{per\_update}}}{2}$ such that $\mathbb{E}[\eta_g \text{clip}_{c_{\text{per\_update}}}(\eta_l x)] = 0$.

- For per-sample clipping, suppose $c_{\text{per\_sample}} = 1$. Then for any $\eta > 0$, it holds that $\mathbb{E}[\eta \text{clip}_{c_{\text{per\_sample}}}(x)] = \frac{\eta}{3} > 0$.

**Per-sample clipping recovers the result from Koloskova et al. (2023) when $\tau = 1$ and $\zeta = 0$**

Let us consider the special case where $\tau = 1$, and $f_i = f$ for any $i \in [n]$. Each client performs clipped stochastic gradient descent with a minibatch size of 1. Then the resulting algorithm is equivalent to the centralized mini-batch SGD with a mini-batch size of $n$. Note in this case, Theorem I can be simplified as:

$$\mathcal{O}\left(\sqrt{\frac{F_0}{\eta T}} + \frac{F_0}{\eta c T} + \sqrt{\frac{\eta L \sigma^2}{n}} + L\eta \min(\sigma, c) + \min(\sigma, \frac{\sigma^2}{c})\right) \text{ where } L := L_0 + cL_1.$$

Since $\eta \leq \frac{1}{14\tau L}$, the last two terms become $\min(\sigma, c) + \min(\sigma, \frac{\sigma^2}{c})$. By simply discussing the relation between $\sigma$ and $c$, it holds that $\Theta(\min(\sigma, c) + \min(\sigma, \frac{\sigma^2}{c})) = \Theta(\min(\sigma, \frac{\sigma^2}{c}))$. Therefore, the resulting rate covers the clipped mini-batch SGD rate as presented in Koloskova et al. (2023).

# F  COMPARISON AGAINST EXISTING WORKS

## F.1  PER-SAMPLE CLIPPING

Our result covers the convergence rate of the centralized clipped mini-batch SGD (single worker, $n = 1$) from Koloskova et al. (2023) when we assume $\zeta = 0$ and communicates at every iteration ($\tau = 1$), except for the fourth term in Theorem I, which only mildly influences the convergence (Koloskova et al., 2020). Compared to CELGC (Liu et al., 2022), we present the convergence rate given *any* arbitrary clipping threshold $c$ with heterogeneous workers ($\zeta^2 > 0$). Compared to Yang et al. (2022), we provide a more explicit influence of stochastic noise and data heterogeneity on the convergence rate.

## F.2  PER-UPDATE CLIPPING

Zhang et al. (2022) studied the same algorithm under uniformly bounded gradient dissimilarity (see Table 1). Note that Corollary 3.2.1 from Zhang et al. (2022) exactly recovers the standard FedAvg result under the assumption that $c > \eta_l \tau G$. However, assuming the norm of the update $||\eta_l \sum_{k=1}^{\tau} \nabla F_i(\mathbf{x})|| \leq \eta_l \tau G$ is always smaller than $c$ is strong and can hardly hold in practice. Yang et al. (2022) relaxed the assumptions by introducing the bounded $\beta$ moment (Zhang et al., 2020c). While $||\nabla F_i(\mathbf{x})||$ is allowed to follow heavy-tailed distribution, this assumption implies $||\nabla f(\mathbf{x})|| \leq G$ for any $\mathbf{x} \in \mathbb{R}^d$ which excludes some interesting functions. Moreover,

the effect caused by the stochastic noise and heterogeneity are hidden in the $G$ parameter. Comparably, we use tighter Assumptions 1, 2 and our result is tight and interpretable.

## G  EXPERIMENTAL SETUP AND DETAILS

We illustrate the performance using multinomial logistic regression (Greene, 2003) on the MNIST dataset (LeCun & Cortes, 2010). We use ten workers with full participation. We randomly subsample 1024 images into each worker to use full-batch gradients ($\sigma = 0$). We vary the number of classes in each worker to simulate different levels of data heterogeneity following Hsu et al. (2019). The data heterogeneity is highest when each worker only has images from a single class. We tune the stepsize for all the experiments to reach the desired target accuracy $\varepsilon := ||\nabla f(\mathbf{x}_t)||$ with the fewest rounds. See Appendix I for a more complicated NN experiment on CIFAR10 dataset.

We implement all the models with PyTorch 1.7.1 and Python 3.7.9. We prepare the MNIST training dataset by simply randomly subsampling 1024 images per digit. For per-update clipping, we experiment with using stepsize from $\{0.00625, 0.0125, 0.025, 0.05, 0.1, 0.2\}$. For per-sample clipping, we experiment with using stepsize from $\{0.1, 0.2, 0.4, 0.8\}$. We choose the stepsize such that we can reach a specified norm of the gradient $\varepsilon := ||\nabla f(\mathbf{x}_t)||$ with fewest communication rounds. For implementing the clipped FedAvg, the main components are two functions where we apply per-sample and per-update clipping.

```python
def per_sample_clipping(local_model, clipping_threshold):
"""Args:
local_model: the local model $\yy_{i,k}$
clipping_threshold: constant, c
"""
    if clipping_threshold > 0:
        grad_group = torch.cat([p.grad.data.detach().clone().view(-1)
            for _, p in local_model.named_parameters()
            if p.requires_grad and p.grad is not None],
                        dim=0)
        grad_norm = torch.norm(grad_group)
        coef = min(1, clipping_threshold / grad_norm.item())
        if coef < 1:
            for name, p in local_model.named_parameters():
                if p.requires_grad:
                    p.grad.data *= coef
    else:
        coef = 1.0
    return local_model, coef
```

```python
def per_update_clipping(prev_s, current_l, clipping_threshold):
"""Args:
prev_s: the server model from the previous round
current_l: the current updated local model
clipping_threshold: constant, c
"""
    diff = {}
    for k in current_l.keys():
        diff[k] = current_l[k].data - prev_s[k].data
    diff_reshape = torch.cat([diff[k].detach().clone().view(-1) for k in
    diff.keys()], dim=0)
    diff_norm = torch.norm(diff_reshape)

    coef = min(1, clipping_threshold / diff_norm.item())
    for k in current_l.keys():
        diff[k] *= coef
    return diff, coef
```

## H  MNIST EXPERIMENT

We here show the server accuracy evaluated on the MNIST test dataset (10,000 images) for per-sample and per-update clipping. Note, the test accuracy can be improved further with a bigger and/or deeper neural network.

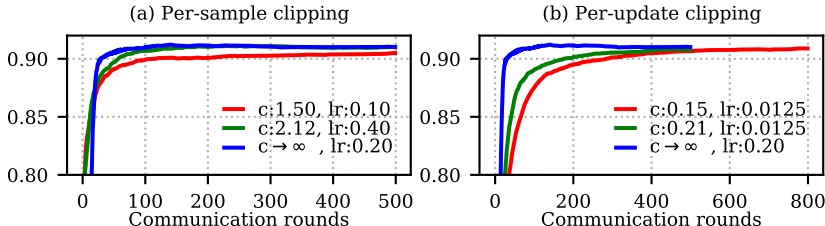

Figure H.1: Server test accuracy (local step $\tau = 7$, the number of class per client is 1) using (a) per-sample clipping and (b) per-update clipping.

# I CIFAR10 EXPERIMENT USING PER-UPDATE CLIPPING

We conduct an experiment considering stochastic noise, local steps, data heterogeneity, non-convex neural networks, different clipping threshold, and more complicated datasets. We use a simple deep neural network with two convolution layers (32 and 64 channels) and two fully connected layers (hidden dimension 512) on CIFAR10 for classification. We split the CIFAR10 training data into 10 clients following Dirichlet distribution Kairouz et al. (2019) with concentration parameter 0.1 (0.1 usually means that the client heterogeneity is high). We use $\tau = 10$ local steps, batch size of 1024, tune the stepsize from {0.05, 0.1, 0.2}. We here mainly performed per-update clipping experiment as we would like to highlight the non-clipping-bias behaviour. We clearly show that when the clipping threshold is small, we can still converge to a similar level as without clipping but at the cost of more communication rounds.

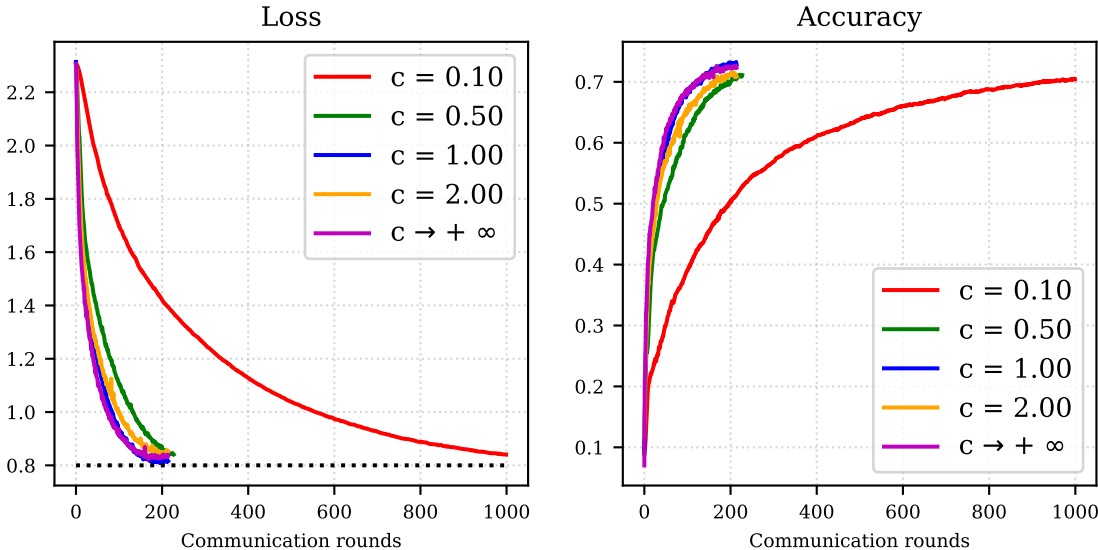

Figure I.1: Test classification loss and accuracy on the CIFAR10 dataset with per-update clipping. When the clipping threshold is small, e.g., c=0.1, we can still obtain similar performance as vanilla FedAvg ($c \to \infty$) at the cost of more communication rounds. We can improve the classification accuracy by e.g., using more advanced deep neural networks.

