| **Algorithm 1** Per-sample clipping | **Algorithm 2** Per-update clipping |
|---|---|
| 1: **procedure** PER-SAMPLE CLIPPING | 1: **procedure** PER-UPDATE CLIPPING |
| 2:   Initialize stepsize $\eta$ | 2:   Initialize local and global stepsize $\eta_l, \eta_g$ |
| 3:   **for** $r = 1, \ldots, R$ **do** | 3:   **for** $r = 1, \ldots, R$ **do** |
| 4:     Send server model $\mathbf{x}$ to all clients | 4:     Send server model $\mathbf{x}$ to all clients |
| 5:     **for** client $i = 1, \ldots, n$ **in parallel do** | 5:     **for** client $i = 1, \ldots, n$ **in parallel do** |
| 6:       initialize local model $\mathbf{y}_i \leftarrow \mathbf{x}$ | 6:       initialize local model $\mathbf{y}_i \leftarrow \mathbf{x}$ |
| 7:       **for** $k = 1, \ldots, \tau$ **do** | 7:       **for** $k = 1, \ldots, \tau$ **do** |
| 8:         $\mathbf{g}_i \leftarrow \min\left(1, \frac{c}{\lVert \nabla F_i(\mathbf{y}_i) \rVert}\right) \nabla F_i(\mathbf{y}_i)$ | 8:         $\mathbf{y}_i \leftarrow \mathbf{y}_i - \eta_l \nabla F_i(\mathbf{y}_i)$ |
| 9:         $\mathbf{y}_i \leftarrow \mathbf{y}_i - \eta \mathbf{g}_i$ | 9:       **end for** |
| 10:      **end for** | 10:      $\Delta_i \leftarrow \mathbf{y}_i - \mathbf{x}$ |
| 11:      Communicate $\mathbf{y}_i$ to the server | 11:      $\Delta_i \leftarrow \min\left(1, \frac{c}{\lVert \Delta_i \rVert}\right)\Delta_i$ |
| 12:    **end for** | 12:      Communicate $\Delta_i$ to the server |

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

**Relation with Koloskova et al. (2023)** While we follow a similar case distinction discussion as Koloskova et al. (2023) in our proof, there are some fundamental challenges arising from the complication of the decentralised algorithms ($\tau > 1$). We both assume $(L_0, L_1)$-smoothness, but our revised $(L_0, L_1)$-smooth assumption is more tailored to the distributed setup. Adapting the theory from centralised SGD to distributed SGD with multiple local steps is non-trivial due to the entangled effect of stochastic noise, data heterogeneity Stich (2019); Koloskova et al. (2020). We have carefully addressed them in Lemma 8, 9, and 11. Additionally, we observe promising convergence results, i.e., can reach any target accuracy, by incorporating two stepsizes compared to the *unavoidable clipping bias* behaviour as shown in Koloskova et al. (2023). Additionally, we have provided a more thorough privacy analysis under the more practical bounded variance assumption.