# OpenReview forum: "An improved analysis of per-sample and per-update clipping in federated learning"
_ICLR.cc/2024/Conference — ICLR 2024 poster_

### Official Review · Reviewer_EwA6 · 2023-10-26

**Soundness:** 4 excellent
**Presentation:** 3 good
**Contribution:** 2 fair
**Rating:** 6
**Confidence:** 2

**Summary:**

The paper provides a tight convergence analysis of federated averaging with clipping under two scenarios: per-sample clipping, where sample gradients are clipped during local optimization, and per-update clipping, where sample gradients are not clipped but the entire user update for each round is clipped. It demonstrates that per-sample clipping converges to a neighborhood of a stationary point, while per-update can converge to any accuracy if the inner step size is small enough. An extended analysis that includes added noise for differential privacy is provided.

**Strengths:**

The writing is very clear and the analysis is insightful. Clipping in FL is an important problem to study, as some means of bounding sensitivity is needed to achieve differential privacy.

**Weaknesses:**

The experiments would be stronger if continued for more communication rounds.

Minor things:
The "assumptions" column of Table 1 needs formatting (sometimes assumptions are referred to as An, sometimes just as n)

**Questions:**

What is the meaning of the dotted line in Figs 1/2c?

It is unusual as far as I know to obtain a DP guarantee with per-sample clipping. Does that come from bounding the sensitivity by taking the worst case ||y_i|| given the fixed number of local steps \tau? That seems like it would be really weak. So I'm surprised that "per-update is no better than per-sample in terms of the optimal privacy/utility trade-off". Can you provide any intuition there?

Could there be any utility in doing *both* per-sample and per-update clipping? How hard would it be to extend the analysis to that case, and if you can, what does it say?

---

> ### Author Response · Authors · 2023-11-20
>
> We thank the reviewer for the interesting and helpful comments. You can find our replies below:
>
> > W1. The experiments would be stronger if continued for more communication rounds.
>
> Thanks for the suggestion. We agree with the reviewer and have updated Fig.2 (c) in the revised manuscript. With the updated Fig.2 (c), we clearly see that even when $c=0.15$, we can still reach the target accuracy but at the cost of more communication rounds.
>
> >W2. The "assumptions" column of Table 1 needs formatting
>
> Thank you for providing suggestions for better formatting. We have improved Table 1 in the revised manuscript.
>
> >Q1. The meaning of the dotted line in Figs 1/2c
>
> The dotted lines represent the target accuracy $||\nabla f(\mathbf{x}_t)||=0.18$. We have clarified this in the caption for Fig.1 and 2.
>
> > Q2. It is unusual as far as I know to obtain a DP guarantee with per-sample clipping. Does that come from bounding the sensitivity by taking the worst case ||y_i|| given the fixed number of local steps \tau? That seems like it would be really weak.
>
> We refer to Appendix C.6 and D.5 for further detailed discussions (especially regarding the notion of privacy, sensitivity and how to analyze the trade-off for both algorithms). Please let us know if there is anything unclear.
>
> >Q2. I'm surprised that "per-update is no better than per-sample in terms of the optimal privacy/utility trade-off". Can you provide any intuition there?
>
> Thank you for your question.  We refer to the **Privacy discussion** section of per-update clipping in the main text. More detailed
>  information can be found in Appendix C.6.3 and Appendix C.6.4.
>
> > Q3. Could there be any utility in doing both per-sample and per-update clipping? How hard would it be to extend the analysis to that case, and if you can, what does it say?
>
> Thank you for the interesting question. We, unfortunately, are not sure what you mean by doing both per-sample and per-update clipping. If you refer to using two stepsizes in Algorithm 1, here is the answer:
>
> Let us first focus on the convergence behaviour without considering the DP noise. Then, adding an inner stepsize to per-sample clipping makes it better, as the resulting algorithm can converge to any accuracy given any clipping threshold. Also, a larger stepsize can be used when $L_1\neq 0$ compared with per-update clipping.
>
> However, when we add DP noise, adding an inner stepsize into Algorithm 1 is unlikely to improve the best privacy-utility trade-off. We refer to Appendix C.6.4 for details.
>
> We hope we have addressed your concern. If not, we would like to engage in further discussion if the reviewer could
> elaborate more on this question
>
> > General
>
> We thank the reviewer for the positive feedback. If you believe this paper should be accepted, please consider increasing your score - this would boost our chances! Thank you!

---

> > ### Comment · Reviewer_EwA6 · 2023-11-22
> >
> > Thank you for you comment. I will argue for acceptance but I will keep my rating. (Maybe I would raise to 7 if it were an option.)

---

> > > ### Author Response · Authors · 2023-11-22
> > >
> > > Thanks a lot for your support!

---

### Official Review · Reviewer_VZxT · 2023-10-30

**Soundness:** 3 good
**Presentation:** 3 good
**Contribution:** 2 fair
**Rating:** 6
**Confidence:** 3

**Summary:**

In summary, this work focuses on analyzing the effect of per-sample clipping and per-update clipping in private federated learning theoretically, and improves the theoretical results from previous works. Specifically, the major improvement lies in two aspects: 1. Fewer assumptions than previous works. Previous works like Zhang et al. (2022) and Yang et al. (2022) rely on extra assumptions like the uniformly bounded stochastic gradient or bounded $\beta$-moment of the stochastic gradient; 2. Convergence rate under the arbitrary clipping threshold, while previous works only provide rate under specific choices of the clipping threshold.

**Strengths:**

As far as I can see, when compared to existing works, the theoretical bounds in this work are certainly more appealing from three aspects.

1. Relying on minimal assumptions for federated learning.

2. Convergence guarantee under arbitrarily clipping threshold. In my view, this is the most important improvement compared to existing work, because, in practice, the clipping threshold is usually a hyperparameter. A continuous bound on the clipping threshold is certainly more helpful for understanding the effect of this hyperparameter.

3. A more interpretable bounds that uncover the relationship among convergence, data heterogeneity, and clipping threshold.

**Weaknesses:**

So far I didn't see a major weakness in this work, and the theoretical results appear to be correct, although I didn't carefully check the math details.

While I acknowledge that this is certainly a solid work, I would not consider the contribution significant. Because, firstly, the contribution is mainly on the theoretical exploration, and does not lead to practical guidance for hyperparameter tuning. On the other, there has been some theoretical exploration on this topic, and it seems that many proof techniques of this work come from Koloskova et al. (2023). Therefore, I give a "fair" score for the contribution and only recommend it for borderline acceptance.

**Questions:**

So far I have no other questions

---

> ### Author Response · Authors · 2023-11-20
>
> We thank the reviewer for the constructive evaluation of our paper. We provide an answer to your concerns below:
>
> > W1: The contribution is mainly on the theoretical exploration, and does not lead to practical guidance for hyperparameter tuning
>
> Indeed, we agree that the contribution is mainly on the theoretical improvements.
>
> - Suppose there is no DP noise. We provide explicit formulas for the optimal choices of stepsize to reach $\epsilon$-accuracy in **sections C.5 and D.4** in the appendix.
>
> - When considering DP noise, we provide the optimal choices of stepsize and clipping thresholds for reaching optimal privacy-utility trade-off in **sections C.6 and D.5** in the appendix.
>
> Although setting theoretically optimal hyperparameters can be complex and unlikely in practice, we could still gain some vague insights or ideas from the theories.
>
> > W2: There has been some theoretical exploration on this topic, and it seems that many proof techniques of this work come from Koloskova et al. (2023).
>
> We agree with the reviewer that there has been some theoretical exploration on this topic already. However, there is no tight analysis for these two algorithms under standard assumptions so far.  Presenting tight convergence results and providing detailed privacy-utility discussion for the two popular clipping algorithms with relaxed assumptions can open further research directions for analyzing differential private federated learning.
>
> Similar to many local SGD literature [Stich 2019, Koloskova et al. 2020] that are built upon the proof techniques used in mini-batch SGD, we are also greatly inspired by the case distinction discussion from Koloskova et al. (2023). However, analyzing the algorithms in decentralized settings is non-trivial. Please see **section 3.3 (relation with Koloskova et al. 2023)** for a more detailed comparison in the revised manuscript.
>
> > General
>
> We thank the reviewer for the very positive comments. We hope that we have addressed your concerns. If not, we would like to engage in further discussion.

---

### Official Review · Reviewer_CsYx · 2023-10-31

**Soundness:** 3 good
**Presentation:** 3 good
**Contribution:** 3 good
**Rating:** 8
**Confidence:** 3

**Summary:**

The paper theoretically analyzes the convergence of the FedAvg algorithm with per-sample or per-update gradient clipping on heterogeneous data. They prove the upper bound for the expectation of gradient norm during training for a general class of learning objectives that satisfies bounded gradient variance, bounded gradient dissimilarity, and distributed $(L_0, L_1)$-smoothness. The main theoretical insights are as follows.
- Under per-sample clipping, the expected gradient norm converges to a neighborhood with a size that depends on the gradient dissimilarity, the stochastic variance in gradient, and the clipping threshold (even when we set the step-size to be infinitesimal).
- By contrast, under per-update clipping, the expected gradient norm can converge to an arbitrarily small level when the clipping threshold is reasonably large and the step-size is small, at the cost of more communication rounds.

The authors also perform logistic regression on MNIST to numerically support their insights on the effect of data heterogeneity on convergence under clipping.

**Strengths:**

- The analysis in this paper holds for arbitrary choice of clipping threshold, while prior works generally either assume a large enough clipping threshold or assume homogeneous data.
- The authors drew interesting comparisons between two clipping methods, per-sample clipping, and per-update clipping, highlighting that the quality of converged solution under per-sample clipping is highly limited by data heterogeneity. In contrast, per-update clipping enjoys convergence to arbitrary accuracy under data heterogeneity.

**Weaknesses:**

1. The authors proved drastically different convergence results under per-sample clipping and per-update clipping, even though the two clipping methods are equivalent when $\tau = 1$ (despite a local step-size $\eta_l$). This seems counterintuitive and needs more clarification.
2. Although the analysis holds for arbitrary clipping threshold, satisfying convergence to an accurate solution still relies on setting a large enough clipping threshold. This insight makes sense theoretically (as a larger clipping threshold enables the training process to be closer to unclipped training), yet it is quite different from practice. (For DP learning, generally, a small clipping threshold such as 0.1 enables good performance [a, b].)
3. Another less critical weakness is that the main theorems (Theorem 1 and Theorem 2) seem to be direct extensions of the prior work [Koloskova 2023]. Consequently, it needs to be clarified how non-trivial are the additional efforts made in this work.

References:
- [a] Tramer, Florian, and Dan Boneh. "Differentially Private Learning Needs Better Features (or Much More Data)." In International Conference on Learning Representations. 2020.
- [b] De, Soham, Leonard Berrada, Jamie Hayes, Samuel L. Smith, and Borja Balle. "Unlocking high-accuracy differentially private image classification through scale." arXiv preprint arXiv:2204.13650 (2022).

**Questions:**

- Could the author explain why this local update step-size $\eta_l$ would contribute to significantly different convergence behaviors between per-sample and per-update clipping? See weakness 1 for details.
- Could the authors comment on this discrepancy between recommended large clipping threshold in this paper and the small clipping threshold used for practical DP learning?


Other minor comments:
- Is there a reason why, in the experiments of Figures 1 and 2, the clipping threshold for per-sample clipping is chosen to be much larger than the clipping threshold for per-update clipping?
- In Table 1, $L$ is not defined. In Theorem 1, $M$ is not defined. In Corollary 1, $g_{i, t}$ cannot be found in Algorithm 1.

---

> ### Author Response · Authors · 2023-11-20
>
> We thank the reviewer for the constructive and helpful comments. Your every comment is important to us. You can find our replies below:
>
> > W1. The authors proved drastically different convergence results under per-sample clipping and per-update clipping, even though the two clipping methods are equivalent when $\tau=1$. This needs more clarification.
>
> We hope the discussion of the two algorithms in **Appendix E** has addressed your concern.
>
> Intuitively speaking, in per-update clipping, the local stepsize can control the norm of the model update $\Delta_i := -\eta_l\sum\nabla F_i(\mathbf{y}_i)$ such that $\Delta_i$ might not be clipped. However, if we remove $\eta_l$, we can no longer control the norm of $\Delta_i$ given a fixed $c$ and may risk converging to the neighbourhood. Please let us know if anything is unclear.
>
> > W2. Although the analysis holds for arbitrary clipping threshold, satisfying convergence to an accurate solution still relies on setting a large enough clipping threshold. This insight makes sense theoretically, yet it is quite different from practice.
>
> Thanks for this interesting question and the references!
>
> - Indeed, when we do not add any noise, the large clipping threshold allows Algorithm 1 to converge to a desired solution.
> - When we consider DP training, the noise scale should be proportional to the clipping threshold, i.e. $\sigma_{\text{DP}}^2 \ge \Omega(c^2)$. In other words, if we use a large clipping threshold, a large noise should be added, which might break the utility. In the DP setting, the optimal choice of $c$ is no longer $+\infty$. Instead, it depends on the privacy budget and many other factors. The optimal choice of $c$ can be found in **Appendix C.6.3 and D.5.3**. Therefore, as you pointed out, a small clipping threshold might often be a good choice in practice.
>
> > W3. Another less critical weakness is that the main theorems (Theorem 1 and Theorem 2) seem to be direct extensions of the prior work [Koloskova 2023]. Consequently, it needs to be clarified how non-trivial are the additional efforts made in this work.
>
> Please see **section 3.3 (relation to Koloskova et al. 2023)** in the updated manuscript for a more detailed comparison against Koloskova et al. 2023.
>
> > Q1. Could the author explain why this local update step-size would contribute to significantly different convergence behaviors between per-sample and per-update clipping? See weakness 1 for details.
>
> We hope our answers to your W1 has addressed this question.
>
> > Q2. Could the authors comment on this discrepancy between recommended large clipping threshold in this paper and the small clipping threshold used for practical DP learning?
>
> We hope the clarification in W2 has addressed your concern. When we consider DP training, the optimal choices of clipping threshold depend on the privacy budget and other factors. Please see **Appendix C.6.3 and D.5.3** for detailed optimal choices of $c$.
>
> > Q3: Is there a reason why, in the experiments of Figures 1 and 2, the clipping threshold for per-sample clipping is chosen to be much larger than the clipping threshold for per-update clipping?
>
> Thank you for your question.
>
> In the MNIST experiment, we observed that the norm of the mini-batch gradient i.e., $\nabla F_i(\mathbf{y}_i)$ (Line 8, Algorithm 1) is much larger than the norm of the model update, i.e., $\Delta_i:=-\eta_l\sum\nabla F_i(\mathbf{y}_i)$ (Line 11, Algorithm 2). Using a smaller clipping threshold than 1.5 will possibly give a similar convergence behaviour as when $c=1.5$ in Fig.1 (c), i.e., the norm of the gradient can not reach the target accuracy $||\nabla f(\mathbf{x}_t)||=0.18$.
>
> > Q4. L, M are not defined
>
> Thank you for pointing these out. We have clarified the meaning of $L$, $M$, and $g_{i,t}$ in our revised manuscript.
>
> > General:
>
> If you agree that we addressed all issues, please consider raising your score--this would boost our chances. If you believe this is not the case, please let us know, and we will try our best to answer further question.

---

> > ### Comment · Reviewer_CsYx · 2023-11-21
> >
> > Thanks for the clarifications. I am satisfied with the authors' response, especially with the added discussion about the best choice of clipping threshold for privacy-utility trade-off. I have increased my score accordingly.

---

> > > ### Author Response · Authors · 2023-11-22
> > >
> > > Thank you very much for your support!

---

### Official Review · Reviewer_6Kwh · 2023-10-31

**Soundness:** 2 fair
**Presentation:** 3 good
**Contribution:** 2 fair
**Rating:** 5
**Confidence:** 4

**Summary:**

This paper studies the problem of clipping in federated learning. More specifically, the authors consider the per-sample and per-update clippings in FedAvg, and derive the corresponding convergence guarantees of FedAvg with these two clipping techniques. The authors also discuss how these two results can be utilized in the differentially private federated learning.

**Strengths:**

The strengths of the paper are as follows:
1. The authors provide the convergence rate of FedAvg with two different clipping techniques.
2. The authors show that how their results can be applied in the privacy protection setting.

**Weaknesses:**

The weaknesses of the current paper are as follows:
1. It is unclear how the results are appropriate for privacy setting.
2. The results cannot recover the unclipped results when the clipping parameters goes to infinity.

**Questions:**

The problem studied in this paper is very interesting and can be very useful in other related problems, such as differentially private federated learning. However, I have several questions about the current paper:
1. It seems that when we choose $c$ as infinity, the results (e.g., Theorem I) cannot reduce to the unclipped results (e.g., LocalSGD  Koloskova et al. 2020), and I'm wondering what steps cause this discrepancy?
2. I'm not sure how the results can be applied to the differentially private setting. The authors consider the stochastic setting, and thus the authors need to specify what is the dataset you want to protect when you apply Corollary I and Corollary II. From my understanding, it would be more meaningful if the authors can provide the finite sum results with bounded stochastic gradient assumption instead of the stochastic setting with bounded variance assumption for the application of differentially private setting.
3. On page 5, comparison to the previous works, why you can claim that the established results can recover the rate of the centralized clipped mini-batch SGD?
4. I'm wondering when you consider the finite sum with bounded stochastic gradient assumption, how the lower bound result will look like in terms of the clipping parameter $c$ and the bounded gradient norm?
5. According to Corollary I and Corollary II, it seems to me that there is no need to use any local update. If this is the case, why don't you just use the private variant of the Minibatch SGD?

---

> ### Author Response · Authors · 2023-11-20
>
> We thank the reviewer for the constructive and helpful comments. Your every comment is important to us. You can find our replies below:
>
> > W2 and Q1. It seems that when we choose $c\rightarrow\infty$, the results (e.g., Theorem I) cannot be reduced to the unclipped results. The results cannot recover the unclipped results when the clipping parameters goes to infinity.
>
> We apologize for the confusion. The complete convergence results of two algorithms, including the case where $c$ is extremely large, are presented in **Theorem III and Theorem IV in the Appendix**. Theorem III and IV recover the unclipped FedAvg when $c\to \infty$ (please see the explanation below Corollary III and Theorem IV).
>
> Please let us know if there is anything that is unclear about recovering the standard unclipped results.
>
> > W1 and Q2. I'm not sure how the results can be applied to the differentially private setting. The authors consider the stochastic setting, and thus the authors need to specify what is the dataset you want to protect when you apply Corollary I and Corollary II. From my understanding, it would be more meaningful if the authors can provide the finite sum results with bounded stochastic gradient assumption instead of the stochastic setting with bounded variance assumption for the application of differentially private setting.
>
> Thank you for your question. We first refer to paragraph **Privacy discussion** under Corollary I and II, together with more detailed information in **Appendix C.6 and D.5** for a comprehensive discussion in private settings.
>
> - We totally agree that finite-sum is ideal for the DP-training settings. We hope the study in Appendix C.6 and D.5 has addressed this concern.
> - We, unfortunately, do not understand why the bounded gradient assumption is better than the bounded variance assumption. We agree that, in practice, knowing the gradient norm may give some insights into choosing the clipping thresholds. However, as far as we understand, the bounded gradient assumption is a more restricted and less practical assumption theoretically, which can exclude simple quadratics. We refer to Table 1 for analysis that uses bounded gradient assumption.
>
> We hope we have addressed your concern. If not, we would like to engage in further discussion if the reviewer could elaborate more on this question.
>
> > Q3. On page 5, comparison to the previous works, why you can claim that the established results can recover the rate of the centralized clipped mini-batch SGD?
>
> We apologize for the confusion. Please see **Section E in the Appendix** for a more detailed explanation for recovering the centralised clipped mini-batch SGD by discussing the relation between $\sigma$ and $c$.
>
> > Q4. I'm wondering when you consider the finite sum with bounded stochastic gradient assumption, how the lower bound result will look like in terms of the clipping parameter and the bounded gradient norm?
>
> We hope our response in Q2 has addressed this question.
>
> Under the bounded gradient norm assumption, there is no clipping bias if we pick $c$ to be larger than the bounded norm. See Zhang et al. 2022 for more detailed information.
>
> > Q5. According to Corollary I and Corollary II, it seems to me that there is no need to use any local update. If this is the case, why don't you just use the private variant of the Minibatch SGD?
>
> Thanks for the interesting question! Yes, this is one of the main weaknesses of most distributed optimization algorithms for addressing non-convex problems.
>
> Let us consider the problem that minimizes $f(\mathbf{x}):=\frac{1}{n}\sum_{i=1}^n f_i(\mathbf{x})$, where $\{f_i\}$ have non-zero heterogeneity. The current distributed algorithms with/without variance reduction can best achieve the same convergence guarantee as gradient descent (gradient norm squared decays as $\frac{LF_0}{R}$ [Karimireddy et al. 2019]), which shows no benefit in doing multiple local steps.
>
> However, doing multiple local steps in practice might still be better due to the potential similarities among the individual functions. The same story applies here. Although we do not observe the benefit of using multiple local steps theoretically, we can possibly reduce communication costs when we use multiple local steps in some scenarios in practice (e.g., see Fig.1 (b)).
>
> > General
>
> If you agree that we addressed all issues, please consider raising your score--this would boost our chances. If you believe this is not the case, please let us know, and we will try our best to answer your further questions.

---

> > ### Comment · Reviewer_6Kwh · 2023-12-04
> >
> > Thank you so much for your response. However, I would like to keep my original scores since I'm still concerning about the practical implications of the results, especially when we have finite sum problems and multiple local updates. The latter one may be more important since for non-private setting, several works have already shown the benefits of local updates in strongly convex and nonconvex settings, and I think it is important to study the effect of local updates in the private settings.

---

### Official Review · Reviewer_6LCG · 2023-10-31

**Soundness:** 2 fair
**Presentation:** 2 fair
**Contribution:** 2 fair
**Rating:** 5
**Confidence:** 5

**Summary:**

This paper studies two different clipping methods, per-sample clipping and per-update clipping, in distributed DP-SGD. Per-sample clipping captures a case where the user clips the update in each local iteration; per-update clipping is more similar to a local SGD scenario where users do local updates for multiple rounds before clipping the update and communicate with the central server. This paper demonstrates that when the clipping threshold is large enough or learning rate small enough, for second-moment bounded gradient, the bias due to clipping can go to zero.

**Strengths:**

This paper generalizes the convergence analysis in SCAFFOLD to study the clipped DP-SGD. The proof seems solid and the claims on the clipping bias make sense to me.

**Weaknesses:**

1. My main concern is that the authors do not make the implication of the theoretical results presented clear. I am puzzled by the motivation of studying Algorithm 1 with a fixed learning rate. It seems that Algorithm 1 can also apply both an inner and an outer learning rate and they should produce the similar bias control as what is claimed in Algorithm 2. So, this make the comparison between Algorithm 1 and 2 in Section 3.3 very confusing, given that after incorporated with the inner learning rate, both of them can achieve arbitrary accuracy. So, what do the analysis want to tell? Which clipping method should we apply in practice? Overall, this makes the practical impact of the theoretical results weak.

2. The privacy-utility tradeoff is not well studied in this paper. In Appendix D.4, the authors briefly discuss the iteration number T required to achieve the balanced point between convergence progress and utility loss resulting from noise. I would suggest plugging the expression of v with $\epsilon$ and $\delta$ into the bound, and compare with existing works, such as "Differentially private empirical risk minimization revisited: Faster and more general". It is not clear to me whether the proposed analysis brings any improvement either in terms of the optimal utility-privacy tradeoff or the efficiency, i.e., the convergence time. The clipping bias needs to be carefully controlled such that the three terms: convergence advancement, utility loss by noise, and the bias should be all in the same degree. From the complicated expression, I find it hard to figure out in practice, do we want a large bias but slow convergence with large DP noise or the converse way.

3. For an ICLR paper, I feel the experiments need to be strengthened. The experiments on both MNIST and CIFAR10 do not report the test accuracy but only the loss. Still, we do not know **under the corresponding  optimal parameter selection** of the two clipping methods, which one performs better in practice. There is no comparison presented with existing empirical works on DP-SGD, such as "Unlocking High-Accuracy Differentially Private Image Classification through Scale".

4. The code is not released.

**Questions:**

1. What is the optimal utility-privacy tradeoff/ utility bound under the optimal parameter selection of Algorithm 1 and 2 (in particular, Algorithm 1 with both inner and outer leaning rate)? In particular, what is the tradeoff considering the bias caused.

2. Which clipping method we should use in practice?

3. Can the different clipping methods produce better performance compared to existing works?

---

> ### Author Response · Authors · 2023-11-20
>
> We thank the reviewer for the evaluation of our paper. Your every comment is important to us. We did our best to understand and reply to your constructive feedback. Most updates are made into **privacy discussion** and **section 3.3** in the revised manuscript. We provide answers to your concerns below:
>
> > W1. This make the comparison between Algorithm 1 and 2 in Section 3.3 very confusing, given that after incorporated with the inner learning rate, both of them can achieve arbitrary accuracy. So, what do the analysis want to tell? Which clipping method should we apply in practice? Overall, this makes the practical impact of the theoretical results weak.
>
> We sincerely apologize for making you feel puzzled about the story and the motivation. Here is the clarification.
>
> Algorithms 1 and 2 with DP noise are commonly used to protect different levels of privacy. We study Algorithm 1 because it is a widely used algorithm [Liu et al. 2022, Yang et al. 2022, Koloskova et al. 2023] for achieving example-level local DP. Our rigorous analysis on per-update clipping (Algorithm 2) opens directions for designing a new per-sample clipping algorithm by incorporating an inner stepsize. However, this strategy is new and non-trivial. We will explore this in our future work.
>
> Nevertheless, these two algorithms still share some similarities from a pure algorithmic point of view. Comparing these two algorithms using our analysis is still meaningful. Please see **section 3.3** in the updated manuscript.
>
> > W2. optimal utility-privacy tradeoff
>
> We thank the reviewer for the reference and appreciate the insightful questions and suggestions. We refer the reviewer to the paragraphs **Privacy discussion** under Corollary I and II, **section 3.3**, for discussing the optimal privacy-utility trade-off and practical implications (see more detailed information in Appendix C.6 and D.5) in our updated manuscript. Please let us know if there is anything still unclear from your side.
>
> > W3. Test accuracy. We do not know under the corresponding optimal parameter selection which one performs better in practice.
>
> Thank you for the suggestion. We have provided the test accuracy in sections H and I in the appendix.
>
> We agree with the reviewer that practical performance can vary depending on the choices of many other factors, as demonstrated in N. Ponomareva, 2023. We have clarified this at the beginning of **section 3.4** in the revised manuscript.
>
> > Q1. What is the optimal utility-privacy tradeoff/ utility bound under the optimal parameter selection of Algorithm 1 and 2 (in particular, Algorithm 1 with both inner and outer leaning rate)? In particular, what is the tradeoff considering the bias caused.
>
> We refer the reviewer to paragraph **Privacy discussion** under Corollary I and II, section 3.3, and more detailed information in Appendix **C.6.3 and D.5.3** for all the discussions.
>
> Note that when $\tau=1$, Algorithm 2 (the same as Algorithm 1 but with both inner and outer stepsize) achieves the same optimal privacy-utility trade-off as Algorithm 1 (single stepsize). Therefore, there is no obvious benefit in terms of optimal privacy-utility bound in using two stepsizes for DP training.
>
> > Q2. Which clipping method we should use in practice?
>
> Both algorithms are commonly used for protecting different levels of privacy. We should choose the algorithm according to the specific purpose, e.g., based on the definition of the neighbouring datasets.
>
> > Q3. Can the different clipping methods produce better performance compared to existing works?
>
> We hope our answer in W1 has addressed your concern. We theoretically study the same clipping algorithms as the existing works [Liu et al. 2022, Zhang et al. 2022, Yang et al. 2022]. Therefore, we do not expect to achieve a better practical performance than them.
>
> > General
>
> If you agree that we addressed all issues, please consider raising your score--this would boost our chances. If you believe this is not the case, please let us know, and we will try our best to answer your every question.

---

> ### Comment · Reviewer_6LCG · 2023-11-23
>
> Thanks so much for your response and additional experiments. But I would like to keep my score since my practice concerns still remain. On one one hand, I still did not see any concrete comparisons between the two algorithms with their **respective** optimal parameter selections in general. Currently, the authors seem to claim that when $\tau=1$, their asymptotic  performance is the same. So, still, I did not see insights from the theory which can be used to guide the practical hyper-parameter selection. On the other hand, I did not see the concrete experimental improvements. I did not see the $\epsilon, \delta$ parameters in the experiments or certain proper learning rate selections (the inner-outer one suggested by the author) can improve the practical DP-SGD performance. Hopefully, those issues can be addressed in your future revision.

---

### Author Response · Authors · 2023-11-20

We thank all the reviewers for their valuable and insightful feedback. We are glad that the reviewers found the problem we study interesting and can be useful in other related problem ($\color{red} 6Kwh$, $\color{green} EwA6$), the comparison between per-sample and per-update clipping interesting ($\color{blue} CsYx$), the proof is solid ($\color{purple} 6LCG$, $\color{orange} VZxT$), and the bound we have provided is appealing and more interpretable ($\color{orange} VZxT$).

We found the constructive feedback of the reviewers very helpful and have prepared an updated version of our manuscript. We have mainly **updated the privacy discussion, section 3.3, and 3.4 in the revised manuscript**. We reply to each reviewer in more detail in individual responses. We summarize the changes below:

- We have improved our discussion of the optimal privacy-utility trade-off under optimal parameters in the **Privacy discussion** paragraph under Corollary I and II, section 3.3, with more detailed information in sections C.6 and D.5 in the Appendix. (Reviewer $\color{purple} 6LCG$, $\color{red} 6Kwh$, $\color{blue} CsYx$, $\color{green} EwA6$)

- We have improved the comparison between Algorithm 1 and Algorithm 2 in section 3.3 (Reviewer $\color{purple} 6LCG$, $\color{red} 6Kwh$, $\color{blue} CsYx$)

- We have clarified the differences between our work and Koloskova et al. 2023 in section 3.3 (Reviewer $\color{blue} CsYx$, $\color{orange} VZxT$)

- We have clarified that we can recover the unclipped FedAvg when $c\rightarrow\infty$ and per-sample clipping can recover the clipped mini-batch SGD in special cases in section E in appendix (Reviewer $\color{red} 6Kwh$)

- We have improved the presentation of the optimal hyperparameter choices when there is no DP-noise in section C.5 and D.4 in Appendix and when there is DP-noise in section C.6 and D.5 in Appendix (Reviewer $\color{blue} CsYx$, $\color{orange} VZxT$)

- Based on Reviewer $\color{purple} 6LCG$'s request, we have provided the test accuracy on the MNIST and CIFAR10 datasets in section H and I.

---

### Meta-Review · Area_Chair_GSdZ · 2023-12-05

**Metareview:**

The paper analyzes the convergence guarantees of two commonly used clipping algorithms for federated learning. The analysis holds for arbitrary choice of clipping threshold (similar to a recent result by Koloskova 2023 - which is done in a non-federated learning setting), while prior works generally either assume a large enough clipping threshold or assume homogeneous data. Some reviews highlight the overlap with the proof technique of Koloskova 2023 but I do not think this can be seen as a trivial extension as the federated learning setting adds significant complexity to the proof.

The reviewers also made the comment that the analysis doesn't have real practical applications and I do agree with this assessment. The paper mostly makes a theoretical contribution.

Some other concerns raised by the reviewers appear more minor to me, including comments about the privacy setting which does not appear to be well-studied. Perhaps the authors could de-emphasize this part of the paper.

Overall, I think the technical contribution of the paper is still good and I recommend acceptance.

**Justification For Why Not Higher Score:**

Mostly a technical contribution, probably only of interest to a small subset of the community.

**Justification For Why Not Lower Score:**

I think the technical contribution is good. Perhaps this paper is not the best fit for ICLR which is usually more deep learning focused. I am happy to discuss this with the SAC if necessary.

---

### Decision · Program_Chairs · 2024-01-16

Accept (poster)